# TRIM28 promotes HIV-1 latency by SUMOylating CDK9 and inhibiting P-TEFb

Xiancai Ma[1,2], Tao Yang[1,2], Yuewen Luo[1,2], Liyang Wu[1,2], Yawen Jiang[1,2], Zheng Song[1,2], Ting Pan[1,2], Bingfeng Liu[1,2], Guangyan Liu[3], Jun Liu[1,2], Fei Yu[2], Zhangping He[2], Wanying Zhang[2], Jinyu Yang[4], Liting Liang[2], Yuanjun Guan[5], Xu Zhang[2], Linghua Li[6], Weiping Cai[6], Xiaoping Tang[6], Song Gao[4], Kai Deng[2], Hui Zhang[2]*

[1]Institute of Human Virology, Sun Yat-sen University, Guangzhou, China; [2]Key Laboratory of Tropical Disease Control of Ministry of Education, Zhongshan School of Medicine, Sun Yat-sen University, Guangzhou, China; [3]College of Basic Medical Sciences, Shenyang Medical College, Shenyang, China; [4]State Key Laboratory of Oncology in South China, Collaborative Innovation Center for Cancer Medicine, Sun Yat-sen University Cancer Center, Guangzhou, China; [5]Core Laboratory Platform for Medical Science, Zhongshan School of Medicine, Sun Yat-sen University, Guangzhou, China; [6]Department of Infectious Diseases, Guangzhou Eighth People's Hospital, Guangzhou, China

*For correspondence:
Correspondence: zhangh92@mail.sysu.edu.cn

Competing interests: The authors declare that no competing interests exist.

**Abstract** Comprehensively elucidating the molecular mechanisms of human immunodeficiency virus type 1 (HIV-1) latency is a priority to achieve a functional cure. As current 'shock' agents failed to efficiently reactivate the latent reservoir, it is important to discover new targets for developing more efficient latency-reversing agents (LRAs). Here, we found that TRIM28 potently suppresses HIV-1 expression by utilizing both SUMO E3 ligase activity and epigenetic adaptor function. Through global site-specific SUMO-MS study and serial SUMOylation assays, we identified that P-TEFb catalytic subunit CDK9 is significantly SUMOylated by TRIM28 with SUMO4. The Lys44, Lys56 and Lys68 residues on CDK9 are SUMOylated by TRIM28, which inhibits CDK9 kinase activity or prevents P-TEFb assembly by directly blocking the interaction between CDK9 and Cyclin T1, subsequently inhibits viral transcription and contributes to HIV-1 latency. The manipulation of TRIM28 and its consequent SUMOylation pathway could be the target for developing LRAs.
DOI: https://doi.org/10.7554/eLife.42426.001

## Introduction

Despite the suppressive combined antiretroviral therapy (cART), the persistence of HIV-1 in the latent reservoirs is the major obstacle to achieve a cure (*Chun et al., 1997*; *Finzi et al., 1997*; *Wong et al., 1997*). To completely eradicate the reservoir, it needs almost 73.4 years of cART due to its long half-life in resting CD4[+] T cells (*Siliciano et al., 2003*). Although over 200/10[6] resting CD4[+] T cells contain proviruses, only 1/10[6] resting CD4[+] T cells (or 1/200 of them) contain inducible replication-competent proviruses and 40/10[6] resting CD4[+] T cells contain intact non-inducible proviruses (*Eriksson et al., 2013*; *Ho et al., 2013*). Most of the proviruses are defective, some of which can be induced to produce functional viral proteins and exposed to immunosurveillance (*Ho et al., 2013*; *Pollack et al., 2017*). Most of the integration sites locate in the intron of actively transcribed genes (*Schröder et al., 2002*). Some integration hotspots were found in latently infected clonally expanded CD4[+] T cells in HIV-1 patients on cART (*Cohn et al., 2015*; *Maldarelli et al., 2014*; *Wagner et al., 2014*). To decrease the latent reservoirs, several functional cure strategies which are

**eLife digest** The human immunodeficiency virus-1, or HIV-1, infects certain human cells, including white blood cells. One reason the infection is incurable is because the virus can integrate its genetic information into its host, and essentially 'sleep' within the host cell, a process called latency. This helps to hide HIV-1 from the immune system and stops it getting destroyed.

Latency represents a critical challenge in treating and curing HIV-1. One proposed cure for HIV-1 involves 'shocking' the viruses out of latency so that they can be eliminated. Applying this so-called shock and kill approach means scientists need to understand more about how latency is maintained. Previous evidence shows that latency requires proteins known as histone deacetylases and histone methyltransferases. Certain gene-silencing proteins called transcription suppressors are also involved.

Ma et al. have now examined latent HIV-1 in several kinds of human cells grown in the laboratory. The cells were modified to make certain proteins at much lower levels than normal. The experiments showed that the loss of a protein called TRIM28 'wakes up' latent HIV-1. TRIM28 attaches chemical marks called SUMOylations to gene regulators in the cell. These SUMOylations restrict the activity of HIV-1's genes, which is important to maintain latency. Specifically, TRIM28 adds SUMOylations to a protein named CDK9 at three key positions.

Reducing the levels of TRIM28 made it easier to shock many HIV-1 in infected cells out of latency. With further investigation, targeting TRIM28 may one day be used to treat HIV-1 infection through a shock and kill method.

DOI: https://doi.org/10.7554/eLife.42426.002

defined as a long-term control of HIV-1 replication and remission of the symptoms of HIV-1 infection without cART, have been proposed (*Katlama et al., 2013*). The latently infected resting CD4 +T cells do not produce sufficient viral antigens which are recognized by immune system. Thus, the infected cells can hardly be eradicated. To this end, the 'shock and kill' strategy, which is one of the functional cure strategies, has been introduced and extensively performed these years. (*Deeks, 2012*; *Geng et al., 2016b*; *Liu et al., 2016*; *Liu et al., 2015*). Based on the 'shock and kill' strategy, the inducible proviruses are 'shocked' out by latency reversing agents (LRAs). Then the immune surveillance system recognizes and 'kills' these HIV-1-expressing cells utilizing various ways which include CTL response and antibody-dependent cell-mediated cytotoxicity (ADCC). However, some infected cells harbor non-inducible proviruses which can hardly be reactivated by LRAs. Permanent silence of proviruses, accompanied by potent anti-HIV-1 immune surveillance, have been proposed as another strategy to inactivate proviruses in infected cells (*Gallo, 2016*; *Kessing et al., 2017*; *Liu et al., 2015*; *Mousseau et al., 2012*; *Mousseau et al., 2015*; *Shan et al., 2012*). Further elucidating the mechanisms of HIV-1 latency will help us to better understand the formation and maintenance of viral reservoirs and develop new therapeutic interventions.

Epigenetic regulations contribute to the establishment and maintenance of HIV-1 latency. Both histone deacetylases including HDAC1 and HDAC2, and histone methyltransferases including G9a, Suv39H1, GLP, EZH2 and SMYD2, have been found to be responsible for ''writing' repressive marks on HIV-1 long terminal repeat (LTR) (*Boehm et al., 2017*; *Ding et al., 2013*; *du Chéné et al., 2007*; *Friedman et al., 2011*; *Imai et al., 2010*; *Marban et al., 2007*; *Ruelas and Greene, 2013*). Suppressive epigenetic marks are further maintained by 'reader' proteins HP1γ and L3MBTL1 (*Boehm et al., 2017*; *du Chéné et al., 2007*). In addition, multiple miRNAs including miR-28, miR-125b, miR-150, miR-223 and miR-382, and lncRNAs including NEAT1 and NRON, were also found to target viral RNAs and viral proteins to mediate transcriptional or posttranscriptional regulations of HIV-1 latency (*Huang et al., 2007*; *Li et al., 2016*; *Zhang et al., 2013*).

Apart from the above epigenetic mechanisms of HIV-1 latency, another barrier to successfully reactivate latent HIV-1 depends upon transcriptional control (*Mbonye and Karn, 2014*). In transcription initiation level, HIV-1 latency is contributed by both the insufficiency of transcription factors including NF-κB, Sp1, AP-1, NFAT and TFIIH, and the accumulation of transcription suppressors including LSF, YY1 and CTIP2 (*Mbonye and Karn, 2017*). For the escaped RNA Polymerase II (RNAP II) which passed through initiation, the absence of HIV-1 Tat and the presence of negative elongation

factors NELF and DSIF facilitate promoter-proximal pausing of RNAP II on HIV-1 LTR (*Ping and Rana, 2001*; *Razooky et al., 2015*). To further escape from promoter-proximal pausing and turn to transcriptional elongation, RNAP II must be extensively phosphorylated at Ser2 residues by positive transcription elongation factor b (P-TEFb), which consists of cyclin-dependent kinase 9 (CDK9) and Cyclin T1 (*Ott et al., 2011*). However, the expression of Cyclin T1 is downregulated in latently infected cells (*Budhiraja et al., 2013*). CDK9 is also inactive because of the dephosphoryla-tion of its T-loop at Thr186 and sequestered in the 7SK small nuclear ribonucleoprotein (snRNP) complex by HEXIM1 or HEXIM2 (*Budhiraja et al., 2013*; *Nguyen et al., 2001*; *Yang et al., 2001*). Another two studies indicate that CDK9 is acetylated at Lys44 by p300 to fully perform its kinase activity (*Cho et al., 2010*; *Fu et al., 2007*). Acetylation of Lys48 by GCN5 negatively regulates CDK9 activity (*Sabò et al., 2008*).

Although many work have unveiled the epigenetic and transcriptional mechanisms of HIV-1 latency, some important questions remain. For instance, there could be a versatile factor responsible for both mechanisms. The mechanism of promoter-proximal pausing has not been fully elucidated. In addition, how the P-TEFb is appropriately sequestered, released and targeted to HIV-1 promoter. More realistically, we have not yet found a powerful LRA which can efficiently reactivate the latent HIV-1 (*Spivak and Planelles, 2018*). To find more cellular factors as potential targets for LRAs, we designed and screened a custom siRNA library targeting multiple cellular epigenetic and non-epige-netic modification pathways in the nucleus. We found that a SUMOylation E3 ligase tripartite motif-containing protein 28 (TRIM28), also known as transcriptional intermediary factor 1β (TIF1β) and KAP1 (KRAB-associated protein-1), binds to CDK9 and mediates the SUMOylation of CDK9, result-ing in the disassociation of CDK9 with Cyclin T1 and the inhibition of CDK9 kinase activity. Conse-quently, its depletion significantly reactivates HIV-1 transcription and reverses HIV-1 latency.

## Results

### TRIM28 suppresses HIV-1 expression and contributes to HIV-1 latency

To identify cellular targets which may contribute to HIV-1 suppression and latency, we started from the design and high-throughput screening of a custom siRNA library which targeted several cellular pathways within the nucleus including chromatin binding, epigenetic modification, chromatin remod-eling, ubiquitination, SUMOylation, and chromosome organization (*Supplementary file 1*). We knocked down each gene in a TZM-bl cell line which harbors an integrated copy of *luciferase* under the control of HIV-1 promoter (*Platt et al., 1998*). We found that many proteins restricted the activ-ity of HIV-1 promoter based on the expression level of luciferase upon knockdown each target (*Figure 1A*). The top hit proteins included HP1α, GLP, SUZ12 and CYLD, which have been identified to inhibit HIV-1 transcription (*Ding et al., 2013*; *Khan et al., 2018*; *Manganaro et al., 2014*). Intrigu-ingly, we found that knockdown of two less-defined SUMOylation pathway genes TRIM28 and SUMO4 significantly upregulated HIV-1 promoter activity (*Figure 1A*, *Figure 1—figure supplement 1A–B*). The overexpression of TRIM28 inhibited the basal level of HIV-1 promoter activity and res-cued HIV-1 repression in dose-dependent manner (*Figure 1—figure supplement 1C*). The upregula-tion was more significant when combined with HIV-1 Tat and TNFα (*Figure 1—figure supplement 1D*). We measured the expression of TRIM28 in different cells and found that TRIM28 is ubiquitously overexpressed in multiple cell lines and primary cells (*Figure 1—figure supplement 1E*). As a com-plemental experiment to search for latency contributors, we compared gene expression in unstimu-lated and PHA-stimulated primary CD4$^+$ T cells utilizing RNA-Seq (*Figure 1—figure supplement 1F*). We found that TRIM28 was highly expressed in unstimulated primary CD4$^+$ T cells and down regulated upon activation by PHA (*Figure 1—figure supplement 1G*). The expression of TRIM28 was upregulated again when the activated CD4$^+$ T cells returned to resting status (*Figure 1—figure supplement 1H*, *Figure 1—figure supplement 2*). To test whether TRIM28 contributes to HIV-1 latency, we knocked down TRIM28 in HIV-1 latency cell line J-Lat 10.6 and found that the depletion of TRIM28 upregulated HIV-1 expression (*Figure 1B–C*) (*Jordan et al., 2003*). Besides, HIV-1 reacti-vation was enhanced much higher when supplemented with histone deacetylase (HDAC) inhibitor suberoylanilide hydroxamic acid (SAHA, vorinostat) or Bromodomain and Extra-Terminal (BET) domain inhibitor JQ-1, both of which have been widely described as LRAs (*Spivak and Planelles,*

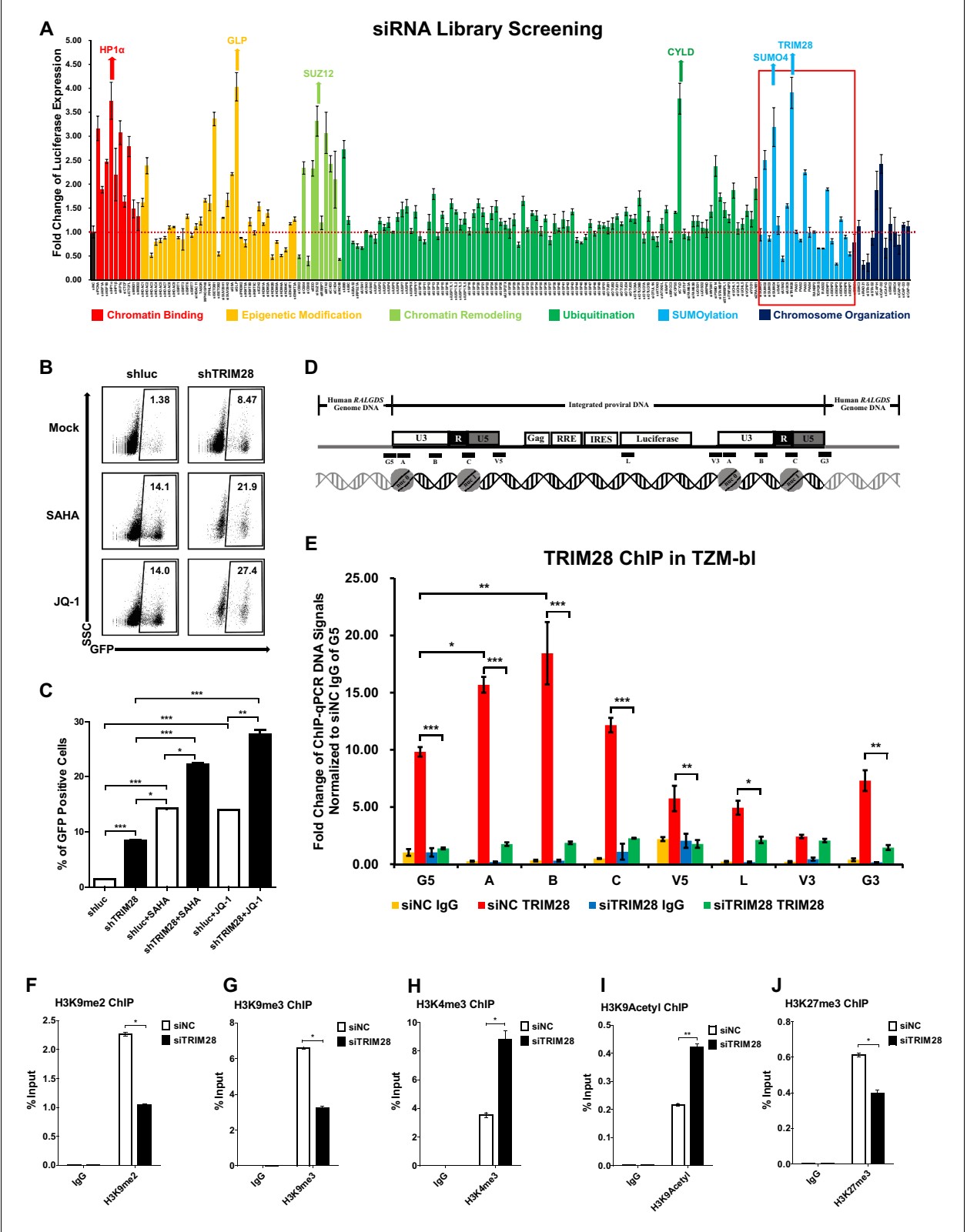

**Figure 1.** TRIM28 suppresses HIV-1 expression and contributes to HIV-1 latency. (**A**) A siRNA library targeting 182 human genes was transfected into TZM-bl cell line, respectively. Three distinct siRNAs targeting each gene were transfected as a mixture. Forty-eight hours post-transfection, cells were harvested and the activity of luciferase from cell lysates was measured. Fold changes were calculated for each gene compared to negative control siRNA (siNC). (**B–C**) shRNA constructs were packaged into recombinant lentiviruses and infected J-Lat 10.6. The reactivation efficiency was measured by

*Figure 1 continued on next page*

*Figure 1 continued*

the GFP-positive percentage which was shown in the top right corner. SAHA and JQ-1 were used as positive controls. (D) Eight ChIP-qPCR primers targeting HIV-1 reporter provirus were designed. G5: Cellular DNA and viral 5′LTR junction; A: Nucleosome 0 assembly site; B: Nucleosome-free region; C: Nucleosome one assembly site; V5: Viral 5′LTR and gag leader sequence junction; L: Luciferase region; V3: Viral poly purine tract and 3′LTR junction; G3: Viral 3′LTR and cellular DNA junction. (E) ChIP assay with antibody against TRIM28 was performed in TZM-bl cell line. All the ChIP-qPCR DNA signals were normalized to siNC IgG of G5. (F–J) ChIP assays with antibodies against H3K9me2, H3K9me3, H3K4me3, H3K9Acetyl and H3K27me3 were performed in TZM-bl cell lines. Data represents mean ±SEM in triplicates. p-Values were calculated by Student's *t*-test. *p<0.05, **p<0.01, ***p<0.001.

DOI: https://doi.org/10.7554/eLife.42426.003

The following figure supplements are available for figure 1:

**Figure supplement 1.** TRIM28 suppresses HIV-1 expression and is upregulated upon activation by PHA.

DOI: https://doi.org/10.7554/eLife.42426.004

**Figure supplement 2.** Primary CD4$^+$T cells populations' identities.

DOI: https://doi.org/10.7554/eLife.42426.005

**Figure supplement 3.** TRIM28 contributes to HIV-1 latency and is enriched on HIV-1 LTR.

DOI: https://doi.org/10.7554/eLife.42426.006

*2018*). These results were well repeated in other latency model cell lines including J-Lat 6.3, 8.4, 9.2, and 15.4 (*Figure 1—figure supplement 3A–E*).

TRIM28 was previously identified to inhibit HIV-1 integration by recruiting HDAC1 to deacetylate HIV-1 integrase (*Allouch et al., 2011*). However, its roles in the expression of integrated HIV-1 and HIV-1 latency have not been clearly elucidated. To this end, we performed chromatin immunoprecipitation (ChIP) assay of TRIM28 in TZM-bl and J-Lat 10.6 cell lines to examine its possible association with integrated HIV-1 DNA (*Supplementary file 2*). We found that TRIM28 was significantly enriched on HIV-1 LTR compared to the regions of host-provirus junction and viral coding region (*Figure 1D– E* and *Figure 1—figure supplement 3F*). The enrichment of TRIM28 on HIV-1 LTR was not influenced by TNFα signaling (*Figure 1—figure supplement 3G*). Because TRIM28 was identified as an epigenetic adaptor recruiting HP1, SETDB1 and NuRD complex to maintain suppressive epigenetic environment, we then tested whether the depletion of TRIM28 would influence the epigenetic status of HIV-1 LTR (*Iyengar and Farnham, 2011*). We observed significant decrease of H3K9me2 and H3K9me3, as well as significant increase of H3K4me3 and H3K9Ac after knocking down TRIM28 (*Figure 1F–I*, *Figure 1—figure supplement 3H–J*). The depletion of TRIM28 also induced slight H3K27me3 downregulation (*Figure 1J*, *Figure 1—figure supplement 3K*). These results indicate that TRIM28 suppresses HIV-1 expression and contributes to HIV-1 latency by manipulating suppressive epigenetic modifications.

## TRIM28 SUMOylates many transcription factors and transferases

Having identified the suppressive epigenetic adaptor role of TRIM28 on HIV-1 latency, we next attempted to search for new mechanism(s) of TRIM28 by function-based mutation. TRIM28 is a mutifunctional protein containing seven different domains (*Ivanov et al., 2007*). The C-terminal bromodomain (BR), which is SUMOylated by the adjacent plant homeodomain (PHD), recruits SETDB1 and NuRD complex in a SUMOylation-dependent manner. The N-terminal tripartite motif RBCC region is composed of a RING finger domain (RING), two B-box domians (BB), and a coiled-coil domain (CC). The RING of TRIM28 functions as an intermolecular SUMO E3 ligase, while PHD is important for the intramolecular SUMO E3 ligase activity (*Ivanov et al., 2007*; *Liang et al., 2011*; *Neo et al., 2015*).

We constructed different TRIM28 mutants by depleting each of the seven domains (*Figure 2A*). Then we knocked down the endogenous TRIM28 with siRNA targeting 3′UTR of *TRIM28* mRNA and supplied with the wild-type TRIM28 construct and the mutants, respectively. Reactivation of HIV-1 expression by the knockdown of endogenous TRIM28 was re-suppressed to the basal level by the wild-type TRIM28 overexpression (*Figure 2B*). Theoretically, none of the HP1BD, NHD, or BR mutants, especially the mutant of PHD which harbor the intramolecular SUMO E3 ligase activity, was able to significantly rescue the suppression, but the results showed they did. Nevertheless, the mutant without RING or RBCC domains totally aborted the re-suppression, which might be due to the loss of the Krüppel-associated box domain zinc fingers (KRAB-ZNFs) binding ability. We tested a mutant containing only RBCC. Interestingly, it still resumed the suppression. We also tested whether

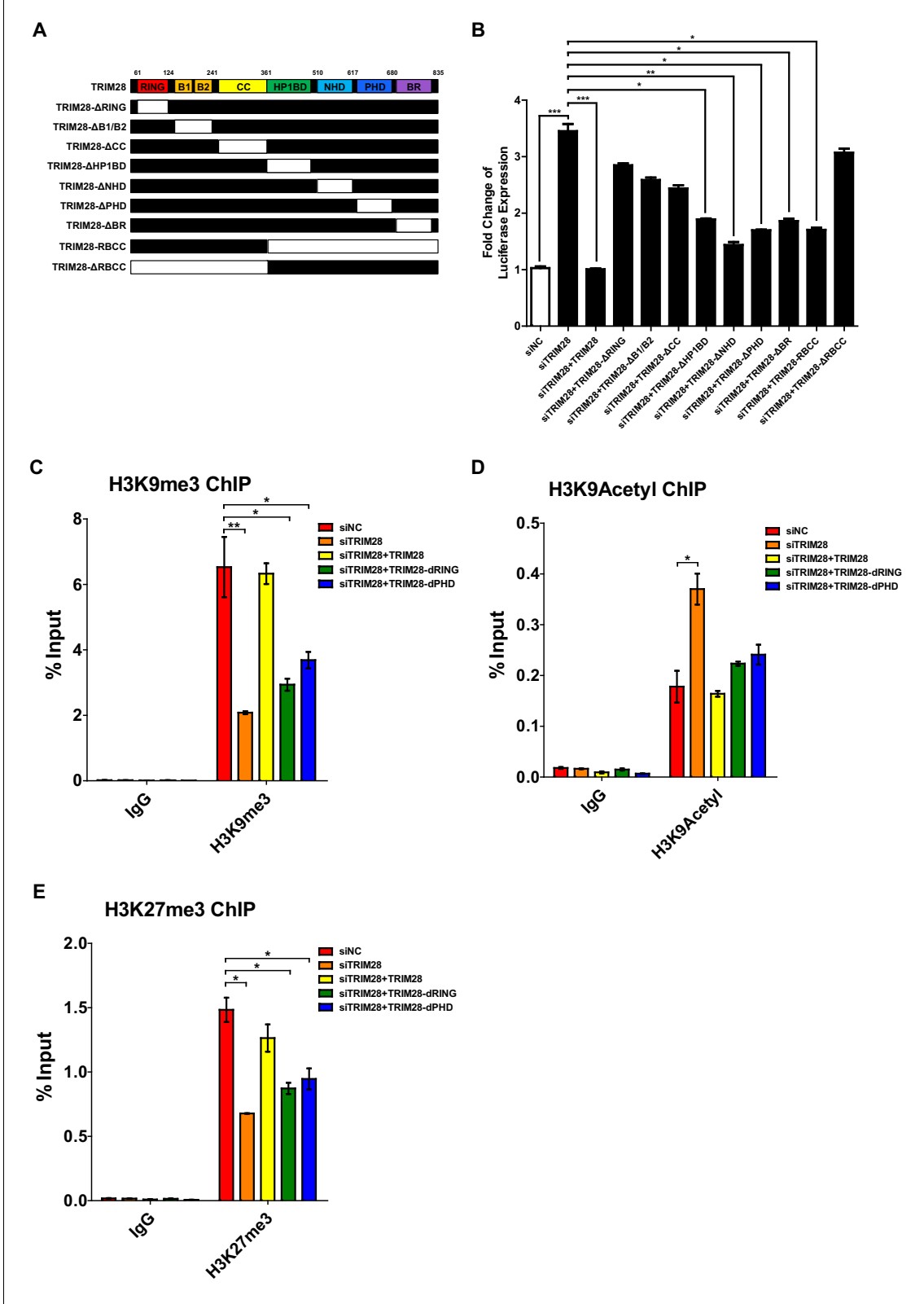

**Figure 2.** Both RING and PHD domains E3 ligase activities are important for repressive epigenetic modifications. (**A**) Schematic of wild-type TRIM28 and nine TRIM28 mutants. (**B**) Endogenous TRIM28 was knocked down by siRNA targeting 3'UTR in TZM-bl cells and re-expressed with wild type and different TRIM28 mutants. The luciferase activity was measured. Data represents mean ±SEM in triplicates. p-Values were calculated by Student's *t*-test. *p<0.05, **p<0.01, ***p<0.001. (**C–E**) Endogenous TRIM28 in TZM-bl cells was knocked down by siRNA targeting 3'UTR of *TRIM28* mRNA. Another

*Figure 2 continued on next page*

*Figure 2 continued*

three groups whose endogenous TRIM28 was knocked down were overexpressed with wild type TRIM28 construct or TRIM28 mutants without RING or PHD domain, respectively. ChIP assays with antibodies against H3K9me3, H3K9Acetyl and H3K27me3 were performed for each group. Data represents mean ±SEM in triplicates. p-Values were calculated by Student's *t*-test. *p<0.05, **p<0.01.

DOI: https://doi.org/10.7554/eLife.42426.007

The following figure supplement is available for figure 2:

**Figure supplement 1.** Positive controls for ChIP.

DOI: https://doi.org/10.7554/eLife.42426.008

the two E3 ligase domains contributed to the epigenetic suppression of HIV-1 promoter by knocking down endogenous TRIM28, followed by the overexpression of wild type or mutated TRIM28. The results showed that the wild-type TRIM28 was able to rescue the suppressive epigenetic marks H3K9me3 and H3K27me3 and suppress the active epigenetic mark H3K9Acetyl, however, the mutant without RING or PHD domain was only able to rescue partial of the suppressive marks (*Figure 2C–E*, *Figure 2—figure supplement 1*). As the RING within RBCC domain plays a key role for the intermolecular SUMO E3 ligase activity of TRIM28, we therefore hypothesize that TRIM28 may utilize the RING domain to SUMOylate cellular protein (s) which is (are) vital for HIV-1 expression (*Liang et al., 2011*).

To identify candidate substrates SUMOylated by TRIM28, we conducted a modified global site-specific SUMOylation Mass Spectrometry (SUMO-MS) (*Figure 3A*). We generated SUMO1-Q92R, SUMO2-Q88R and SUMO4-Q88R mutants mimicking yeast SUMO Smt3 to enable efficient identification of SUMO-acceptor lysines by MS (*Supplementary file 3*) and co-expressed the SUMO mutants with TRIM28 and SUMO E2 UBC9 followed by the enrichment of SUMO conjugated substrates (*Hendriks et al., 2014*). To increase the coverage and mapping possibility of targeted proteins, we used SDS-PAGE to separate the enriched proteins and excised the entire gel lane into 16 slices which were subjected to separate in-gel digestions. The digested peptides were analyzed by nanoscale LC-MS/MS. Finally, we identified 1,329 SUMOyalted proteins at significance threshold below $10^{-7}$ (*Supplementary file 4*). Based on the STRING network analysis, the SUMOylated proteins exerted a large complex network at the interaction confidence of 0.7 (*Figure 3B*). We further performed MCODE analysis on SUMOylated proteins and found that the STRING core network could be clustered into 12 subclusters with interconnectivity scores ranging from 14 to 96 (*Figure 3B*, *Figure 3—figure supplement 1A* and *Supplementary file 5*). Through Gene Ontology (GO) analysis, we found that cellular and metabolic processes were the top two biological processes which the SUMOylated proteins could be involved in (*Figure 3—figure supplement 1B* and *Supplementary file 6*). Most SUMOylated targets have the catalytic activity and DNA binding function. Many transferases and transcription factors were also among the SUMOylated candidates. We specifically clustered the transferases and transcription factors by *k*-means clustering and visualized with STRING analysis. Interestingly, we found that many candidates were pivotal for HIV-1 expression, such as JUN, JUNB, JUND, mTOR, STAT3, Cyclin T1 (CCNT1) and CDK9 (*Figure 3C*). Especially, CDK9 and CCNT1 were also found in MCODE Cluster 8 (*Figure 3—figure supplement 1A*). Recently, it has been identified that the SUMOylation of transcription factor STAT5 was inactivated by benzotriazoles, resulting in the reactivation of latent HIV-1 (*Bosque et al., 2017*). SUMOylation may participate in transcription more generally. We further narrowed down the significance threshold below $10^{-8}$ to find the more extensively SUMOylated targets. CDK9 was still among the top protein candidates (*Supplementary file 7*). Then, we co-overexpressed SUMO system proteins (SUMO1, SUMO2, SUMO4, UBC9 and TRIM28) with 10 transcription factor candidates, respectively. Several transcription factors were SUMOylated, such as NFKB1A, RelA, CCNT1, CDK9, SKIP, MEN1 and JUN, which verified the reliability of our global site-specific SUMO-MS (*Figure 3D*). Nevertheless, the SUMOylation signals were much more significant for CDK9, which merited being further studied.

## CDK9 is SUMOylated by TRIM28

To further verify that CDK9 is SUMOylated by TRIM28, we conducted several in vivo and in vitro SUMOyaltion assays. In vertebrates, there are four well-studied SUMO paralogs, SUMO1, SUMO2,

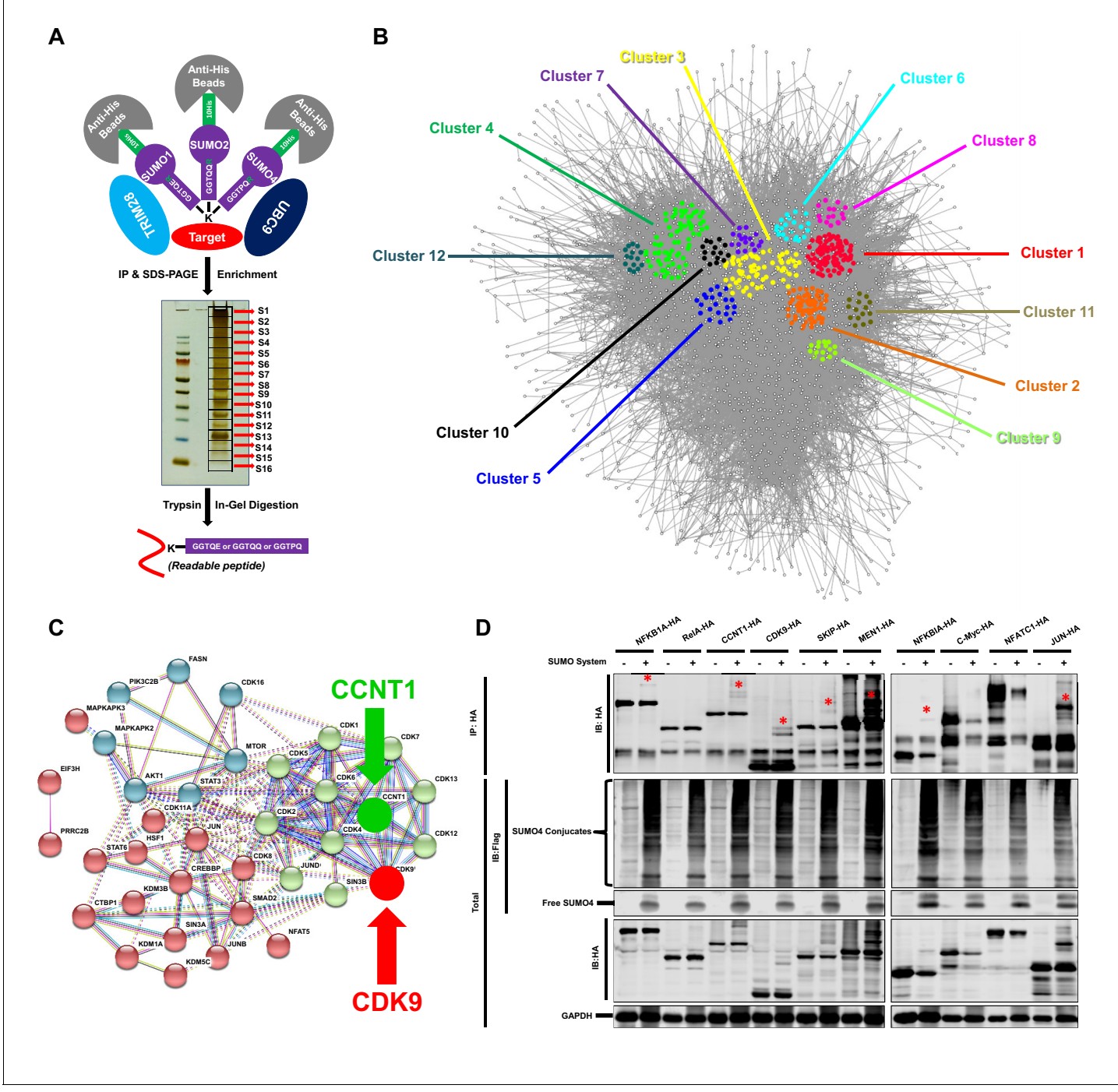

**Figure 3.** TRIM28 SUMOylates many transcription factors and transferases. (**A**) Schematic of global site-specific SUMO-MS. His-tagged SUMO mutants were co-overexpressed with UBC9 and TRIM28. The SUMOylated proteins were enriched by His-tag beads and separated by SDS-PAGE. Gel fragments were excised and subjected to separate in-gel digestions. The digested peptides were desalted and analyzed by nanoscale LC-MS/MS. (**B**) SUMOylated proteins were analyzed with STRING. The network were further analyzed by MCODE. Twelve highly interconnected functional subclusters were extracted and shown in different colors. (**C**) Transferases and transcription factors were clustered by *k*-means clustering and visualized with STRING analysis. (**D**) Ten HA-tagged various transcriptional factors were overexpressed with Flag-tagged SUMO proteins, UBC9 and TRIM28. The targeted proteins were immunoprecipitated (IP) by anti-HA-tag beads followed by immunoblotting (IB) with anti-HA and –Flag antibodies. Asterisk (*) indicated the SUMOylated bands.

DOI: https://doi.org/10.7554/eLife.42426.009

The following figure supplement is available for figure 3:

**Figure supplement 1.** STRING, MCODE and GO analysis of proteins SUMOylated by TRIM28.

*Figure 3 continued on next page*

*Figure 3 continued*

DOI: https://doi.org/10.7554/eLife.42426.010

SUMO3, and SUMO4. Because SUMO2 and SUMO3 share highly sequence-homolog and have similar functions, they are often referred to as SUMO2/3 (*Cubeñas-Potts and Matunis, 2013*). It is worthy to note that the depletion of SUMO4 was able to upregulate the HIV-1 promoter activity more significantly than the depletion of the other SUMO paralogs in our siRNA library screening (*Figure 1A*). The upregulation was more significant when combined with HIV-1 Tat, the phenomenon of which was similar as we observed for TRIM28 (*Figure 4A–B*, *Figure 4—figure supplement 1A*). The knockdown or knockout of SUMO4 was able to reactivate latent pseudotyped HIV-1 in J-Lat 10.6 as well (*Figure 4C*). SUMO4 is also ubiquitously overexpressed in multiple cell lines and primary CD4$^+$ T cells (*Figure 4—figure supplement 1B*). After PHA stimulation in primary CD4$^+$ T cells, the expression of SUMO4 was downregulated (*Figure 1—figure supplement 1G*, *Figure 4—figure supplement 1C*). The expression SUMO4 returned to basal level when activated primary CD4$^+$ T cells re-entered to resting status (*Figure 4—figure supplement 1C*). As the SUMOylation of TRIM28 and associated epigenetic modifiers participates in the regulation of epigenetic patterns, we next testified whether SUMO4 could influence the function of TRIM28 and the epigenetic status of HIV-1 promoter (*Iyengar and Farnham, 2011*). We found that more than half of TRIM28 was lost from HIV-1 LTR upon SUMO4 knockdown, which indicated that the enrichment of TRIM28 on HIV-1 LTR may be partially SUMOylation-dependent apart from the Krüppel-associated box domain zinc fingers (KRAB-ZNFs)–dependent binding (*Figure 4D*, *Figure 4—figure supplement 1D–H*). We also found that H3K9me, H3K9me2 and H3K9me3 were significantly decreased on HIV-1 LTR in the absence of SUMO4, as well as the H3K9 methylation 'writer' SETDB1 and 'reader' HP1α (*Figure 4E–G*, *Figure 4K–L*). Moreover, we observed significant upregulation of H3K9acetyl and H3K4me3 and downregulation of HDAC1, which was consistent with previous reports that TRIM28 recruited SETDB1, HP1α and HDAC1 in a SUMOylation-dependent manner (*Figure 4H–I*, *Figure 4M*) (*Iyengar and Farnham, 2011*). Besides, we found that the H3K27me3 was also decreased on HIV-1 LTR upon SUMO4 knockdown (*Figure 4J*). It is possible that some polycomb repressive complex 2 (PRC2) components such as EZH2 and SUZ12, the major 'writers' of H3K27me3, may be SUMOylated by SUMO4, resulting in the enhancement of modifier function.

As SUMO4 was able to mediate HIV-1 suppression and latency, possibly through the epigenetic control of HIV-1 promoter, we next attempted to identify the underlying mechanism by investigating its role in TRIM28-mediated CDK9 SUMOylation. We co-overexpressed CDK9 with SUMO1, SUMO2 and SUMO4, respectively. We found that CDK9 was mainly SUMOylated with SUMO1 and SUMO4 (*Figure 4—figure supplement 1I*). The SUMO4-CDK9 amount was much more abundant than the SUMO1-CDK9 amount. Besides, SUMO E3 ligase TRIM28 utilized more SUMO4 compared with SUMO1 and SUMO2 (*Figure 4—figure supplement 1J*). After the supplement of TRIM28, the SUMO-CDK9 amount turned to be more abundant. However, the SUMOyaltion did not increase if we only co-overexpressed CDK9 with TRIM28 but without SUMO E2 UBC9, which indicated that TRIM28-mediated SUMOylation was UBC9-dependent (*Figure 5A*). The SUMO-CDK9 amount was increased dose-dependently when the TRIM28 increased gradually (*Figure 5B*). We then conducted in vitro SUMOylation assay. Only when SUMO4, E1 SAE1/UBA2, E2 UBC9 and TRIM28 were supplied into the SUMO conjugation reaction buffer together, was SUMO4 conjugated to CDK9 (*Figure 5C*). After knocking down TRIM28 in HeLa cells, the SUMOylated CDK9 decreased (*Figure 5D*). In our previous siRNA screening, we noticed that the absence of several SUMO-specific isopeptidases (SENPs), which deSUMOylated substrates, prevented the expression of HIV-1, especially SENP3 (*Figure 5—figure supplement 1A–B*). We then co-overexpressed SENP3 with TRIM28 and found that SENP3 prevented TRIM28-mediated CDK9 SUMOylation (*Figure 5E*). To investigate whether TRIM28-mediated SUMOylation of CDK9 by SUMO4 exist in primary CD4$^+$ T cells, we firstly confirmed that the conjugation of SUMO4 to cellular proteins frequently occurs (*Figure 5—figure supplement 1C*). We also immunoblotted the endogenous CDK9 in primary CD4$^+$ T cells and found that a small portion of CDK9 was SUMOylated by SUMO4 (*Figure 5—figure supplement 1D*). The SUMO4-SUMOylated endogenous CDK9 increased significantly after the overexpression of SUMOylation components including SUMO4, UBC9 and TRIM28 (*Figure 5—figure supplement 1E*). Taken

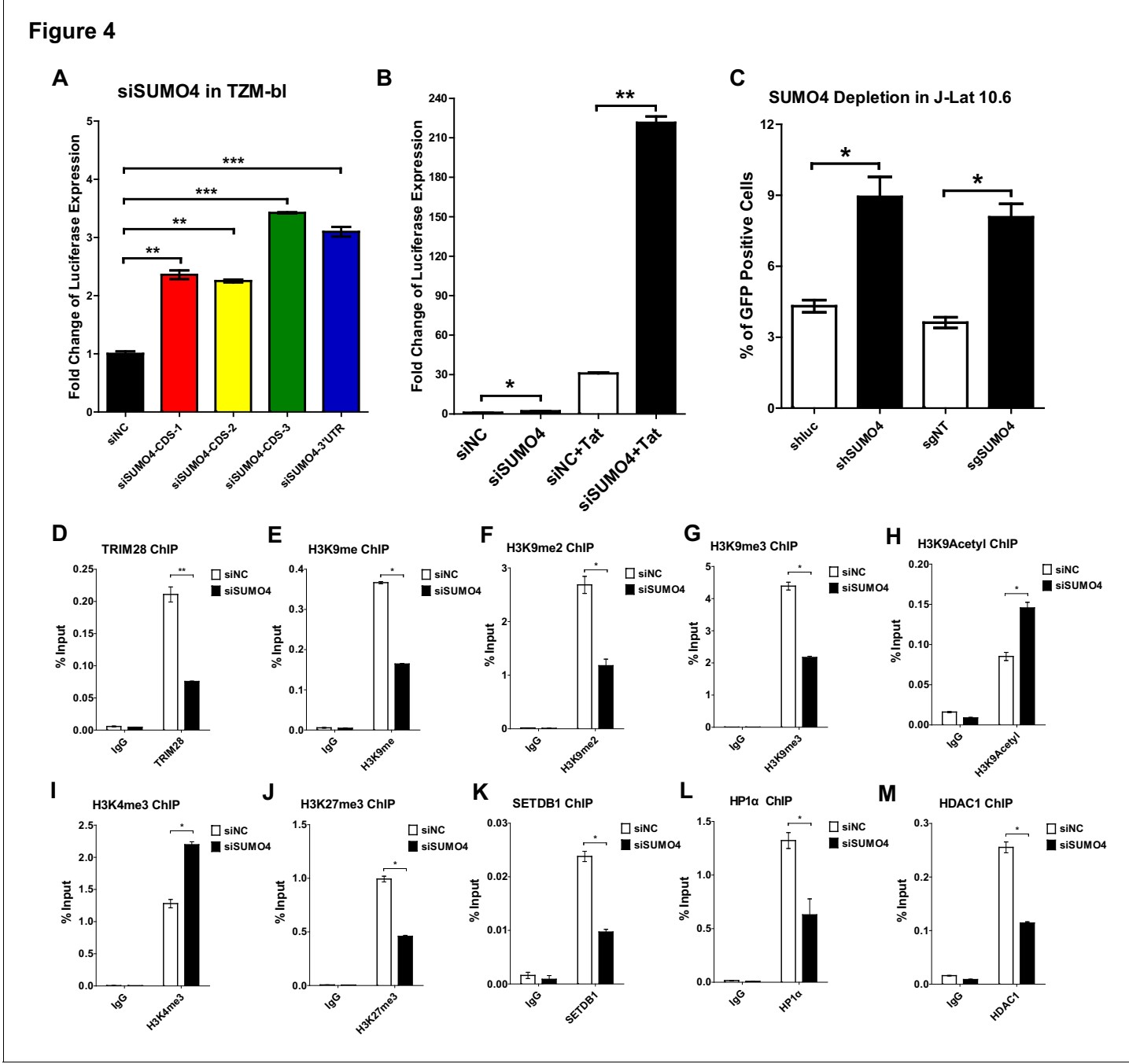

**Figure 4.** SUMO4 suppresses HIV-1 expression and contributes to HIV-1 latency. (**A**) SUMO4 in TZM-bl cells was knocked down by siRNAs targeting the coding sequence and 3'UTR of *SUMO4* mRNA. The luciferase from clarified lysates was quantitated and normalized to siNC. Data represents mean ± SEM in triplicates. p-Values were calculated by Student's *t*-test. **p<0.01, ***p<0.001. (**B**) SUMO4 in TZM-bl cells was knocked down by siRNAs or treated with siNC. HIV-1 Tat construct was co-treated with siRNAs. The luciferase from clarified lysates was quantitated and normalized to the siNC which had no additive. Data represents mean ±SEM in triplicates. p-Values were calculated by Student's *t*-test. *p<0.05, **p<0.01. (**C**) shRNA or sgRNA constructs targeting luciferase (shluc), non-target (sgNT) and SUMO4 (shSUMO4 and sgSUMO4) were packaged into recombinant lentiviruses and infected J-Lat 10.6. The reactivation efficiency was measured by the GFP positive percentage. Data represents mean ±SEM in triplicates. p-Values were calculated by Student's *t*-test. *p<0.05. (**D–M**) SUMO4 in TZM-bl cells was knocked down by siRNA targeting *SUMO4* mRNA. ChIP assays with antibodies against TRIM28, H3K9me, H3K9me2, H3K9me3, H3K9Acetyl, H3K4me3, H3K27me3, SETDB1, HP1α and HDAC1 were performed for each group. Data represents mean ±SEM in triplicates. p-Values were calculated by Student's *t*-test. *p<0.05, **p<0.01.
DOI: https://doi.org/10.7554/eLife.42426.011

The following figure supplement is available for figure 4:

*Figure 4 continued on next page*

*Figure 4 continued*

**Figure supplement 1.** SUMO4 is uregulated upon activation by PHA and is the major paralog used by CDK9 and TRIM28.

DOI: https://doi.org/10.7554/eLife.42426.012

together, our data indicates that TRIM28 mediates the conjugation of SUMO4 to CDK9, which is reversed by SENP3.

## The RING domain of TRIM28 plays a key role in binding to and SUMOylating CDK9

To identify whether TRIM28 binds to CDK9, we used the super-resolution continuous STochastic Optical Reconstruction Microscopy (cSTORM) to investigate the three dimensional (3D) co-localization in the resolution of 20 nm. We found that TRIM28 existed in many small clusters and large bodies in the nucleus and co-localized with dotted SUMO4 (*Figure 6A*, first panel). From amplified view and 3D-cSTORM, we found that SUMO4 proteins were enriched by TRIM28 and shaped big spots (*Figure 6A*, second and third panels; *Video 1*). Although CDK9 existed in dispersed dots all within the nucleus, we still found that CDK9 co-localized with TRIM28 (*Figure 6B*, first panel). Similarly to SUMO4, CDK9 proteins were enriched by and surrounded TRIM28 bodies (*Figure 6B*, second and third panels; *Video 2*). The lateral resolution of cSTORM imaging can be up to 20 nm and the axial resolution is 50 nm, which is within the range to distinguish protein complexes, even single protein molecules (*Lagache et al., 2015*). Thus, we transformed the cSTORM-imaged protein molecules and complexes into small or large spots based on their diameter (*Figure 6C–D*, *Figure 6—figure supplement 1A–B*, left panel; *Video 3*, *Video 4*). The direct interaction between spots and spots was measured in compliance with the criterion of maximal distance of 10 nm (*Figure 6C–D*, *Figure 6—figure supplement 1A–B*, middle panel). The indirect interaction between complexes and spots was measured in compliance with the criterion of maximal distance of 100 nm (*Figure 6C–D*, *Figure 6—figure supplement 1A–B*, right panel). Finally, we found that nearly 80% of TRIM28 spots or complexes were co-localized with 94% of SUMO4 spots (*Figure 6E*). Similarly, 88% of TRIM28 spots or complexes were co-localized with 76% of CDK9 spots (*Figure 6E*).

Through co-immunoprecipitation (Co-IP) assay, we found that CDK9 bound to TRIM28, even in the presence of RNase (*Figure 7—figure supplement 1A*). To identify which region of TRIM28 bound to CDK9, we examined various TRIM28 deletion mutants to enrich CDK9. The depletion of RING aborted the binding of CDK9 as well as the SUMOylation of CDK9 (*Figure 7A–B*). Further, we co-transfected GFP-TRIM28 and several GFP-TRIM28 mutants with RFP-CDK9 in HEK293T cells and utilized the super-resolution Structured Illumination Microscopy (SIM) to investigate the co-localization. Exogenously expressed TRIM28 also co-localized with CDK9 with Pearson's coefficient of 0.7336 and thresholded Mander's coefficient of 0.5846, which indicated a highly co-localization. However, the mutant of RING domain deletion was not capable (*Figure 7C–D*). We also inspected the SUMOylation status of each TRIM28 mutants and found that all the mutants was SUMOylated, which coincided with previous reports that both the RING and PHD had the E3 ligase activity and enriched UBC9 (*Figure 7—figure supplement 1B*) (*Ivanov et al., 2007*; *Liang et al., 2011*). Collectively, our results indicate that TRIM28 binds to CDK9 and SUMOylates CDK9 through its RING domain.

## CDK9 function is inhibited when SUMOylated by TRIM28

After confirming CDK9 is indeed SUMOylated by TRIM28, and also because the RBCC domain contributes to HIV-1 suppression, we next tried to examine whether the function of CDK9 is influenced by TRIM28-mediated SUMOylation. We firstly utilized ATAC-Seq to probe the chromatin accessibility of HIV-1 promoter upon TRIM28 elimination. We found that most of the increased accessible regions across the genome lied on promoters and distal intergenic regions upon the depletion of TRIM28 in J-Lat 10.6 or TZM-bl cell lines (*Figure 8—figure supplement 1A–B*). Through GO analysis and Clusters of Orthologous Groups of proteins (COGs) analysis, we found that the chromatin accessibility variation happened in genes related to various biological processes and cellular components upon TRIM28 depletion (*Figure 8—figure supplement 1C–D*). Most of the influenced general functional genes had the DNA or protein-binding abilities and catalytic activities (*Figure 8—figure*

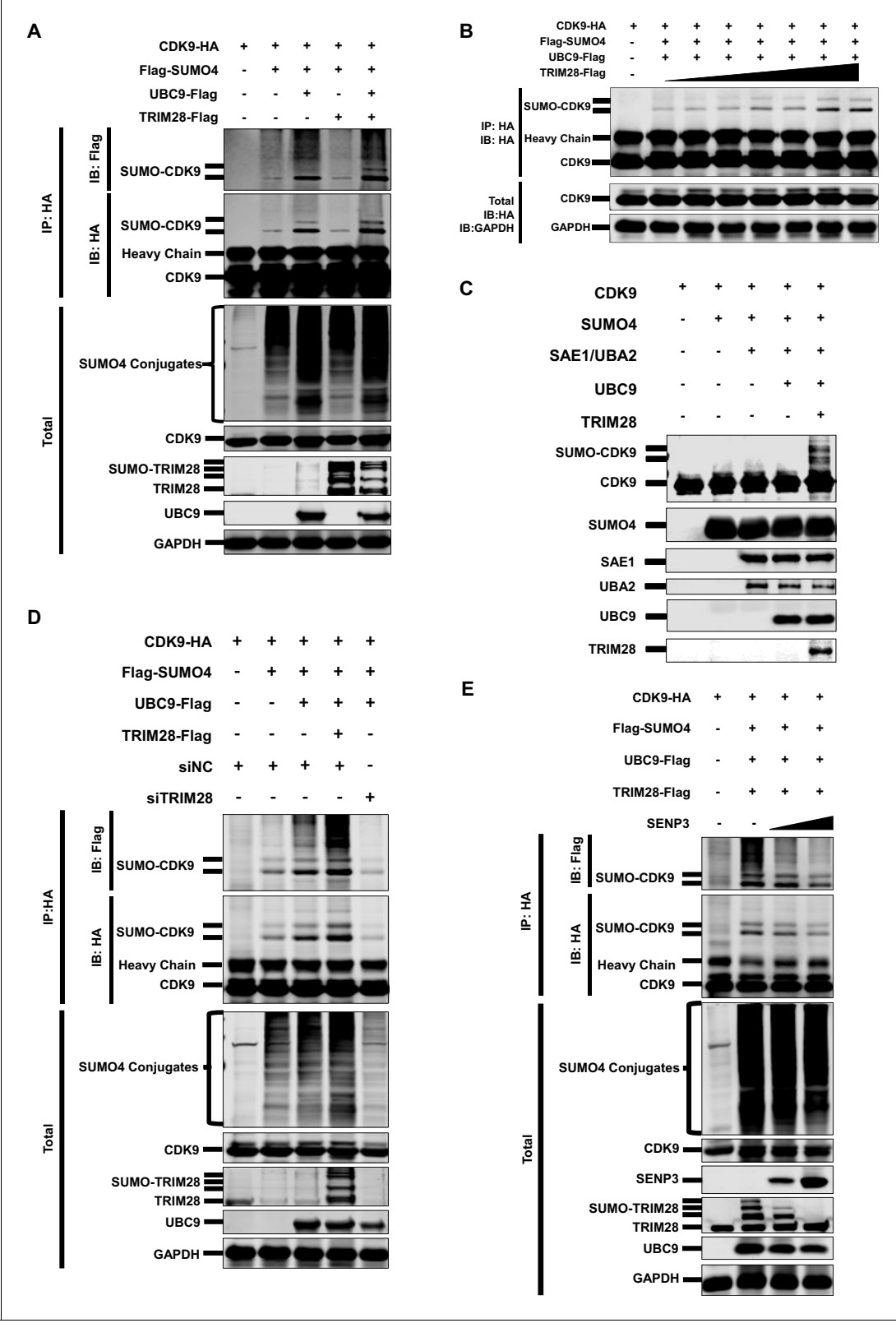

**Figure 5.** CDK9 is SUMOylated by TRIM28. (**A**) HA-tagged CDK9 was co-overexpressed with Flag-tagged SUMO4, UBC9 or TRIM28. CDK9 was IP with anti-HA-tag beads, followed by IB with anti-HA and –Flag antibodies. TRIM28, UBC9 and GAPDH in total samples were IB with specific antibodies targeting each proteins. (**B**) HA-tagged CDK9 was co-overexpressed with Flag-tagged SUMO4, Flag-tagged UBC9 and different amount of Flag-tagged TRIM28. Target proteins were IB as in (**A**). (**C**) In vitro purified CDK9, SUMO4, SAE1, UBA2, UBC9 and TRIM28 were co-cultured in SUMO

*Figure 5 continued on next page*

eLIFE Research article

*Figure 5 continued*

conjugation reaction buffer. Proteins including SUMOylated CDK9 were IB with antibodies against each targets. (**D**) HA-tagged CDK9 was co-overexpressed with Flag-tagged SUMO4, Flag-tagged UBC9 or Flag-tagged TRIM28, and siNC. In the last group, CDK9 was co-overexpressed with SUMO4, UBC9 and siRNA against TRIM28. Target proteins were IB as in (**A**). (**E**) HA-tagged CDK9 was co-overexpressed with Flag-tagged SUMO4, Flag-tagged UBC9, Flag-tagged TRIM28 or two gradients of SENP3. Target proteins were IB as in (**A**).

DOI: https://doi.org/10.7554/eLife.42426.013

The following figure supplement is available for figure 5:

**Figure supplement 1.** CDK9 is deSUMOylated by SENP3 and CDK9 SUMOylation occurs in primary CD4[+]T cells.

DOI: https://doi.org/10.7554/eLife.42426.014

*supplement 1C–D*, *Figure 8—figure supplement 2A–B*). To inspect whether the chromatin accessibility of HIV-1 genome was influenced upon TRIM28 depletion, we separately aligned the sequencing reads to HIV-1 reference genome. We found that the accessible region indicated by transposable tag density increased on HIV-1 LTR when TRIM28 was knocked out from J-lat 10.6 cell lines, as well as when TRIM28 was knocked down in TZM-bl cell lines, which indicated significantly enhanced promoter activity (*Figure 8A–B*). The promoters of genes within which the integrated pseudotyped HIV-1 or HIV-1 reporter provirus located and housekeeping gene GAPDH were not influenced (*Figure 8—figure supplement 2C–F*). Alternatively, we also observed significant enrichment of CDK9 and Ser2 super-phosphorylated RNAP II on HIV-1 LTR upon the knockdown of either TRIM28 or SUMO4, which was in agreement with the results that the depletion of TRIM28 or SUMO4 reactivated HIV-1 expression (*Figure 8C–D*).

Interestingly, through Co-IP assay, we found that Cyclin T1 only bound to wild-type CDK9 to form P-TEFb complex, not the SUMOylated CDK9 (*Figure 8E*). To investigate whether TRIM28-mediated SUMOylation of CDK9 affects the kinase activity of CDK9, we conducted in vitro CDK9 SUMOylation assay followed by CDK9 kinase assay (*Figure 8—figure supplement 3A*). We found that the kinase activity of CDK9 significantly decreased when SUMOylated by TRIM28. However, the kinase activity of CDK9 was not influenced without the addition of TRIM28, although the other SUMOylation components have been added (*Figure 8F*). Collectively, TRIM28-mediated SUMOylation impairs both the binding ability of CDK9 to Cyclin T1 and the kinase activity of CDK9 to RNAP II, resulting in the dysfunction of transcription elongation.

## The Lys44, Lys56 and Lys68 residues of CDK9 are SUMOylated with SUMO4

To elucidate the mechanisms that SUMOylation weakens the interaction between CDK9 and Cyclin T and the CDK9 kinase activity, we next attempted to identify the CDK9 SUMOylation sites which should occur on lysine residues. In order to narrow down the search scope, we equally grouped the sequence of CDK9 into three parts. Each part was given a mutant version that all the lysines were mutated to arginines. Then, we combined these six sequences and obtained eight constructs including the wild type CDK9 (*Figure 9—figure supplement 1A* and *Supplementary file 3*). The construct named CDK9-K0R, which contained the mutation that all the lysines were changed to arginines, totally aborted the capability of CDK9 to be SUMOylated (*Figure 9—figure supplement 1B*). However, the other CDK9 mutants still were able to be SUMOylated by TRIM28, which indicated that multiple SUMOylation sites might exist across the whole CDK9 sequence. To locate all the suspicious SUMOylation sites, we adopted reversing mutation strategy based on CDK9-K0R construct. Each of the 29 arginines of CDK9-K0R was mutated back to lysine separately (*Supplementary file 3*). Finally, we found that several lysines on CDK9 were significantly SUMOylated (*Figure 9A*). Among them, multiple SUMOylation sites were adjacent to CDK9 C-terminal autophosphorylation sites which have been reported to be required for high-affinity binding of Tat–P-TEFb to TAR RNA (*Baumli et al., 2008*; *Garber et al., 2000*). SUMOylation may decrease the binding ability through preventing the neighboring phosphorylation.

It was notable that, although the remove of endogenous TRIM28 significant downregulated SUMO-CDK9, slightly residual SUMO-CDK9 still occurred, implying that other CDK9 SUMOylation E3 ligases may exists and some of the SUMOylation sites are not the TRIM28 targets (*Figure 5D*). To further identify which sites are indeed SUMOylated by TRIM28 only, we knocked down the endogenous TRIM28 and tested the SUMOylation potential of the candidate sites identified above.

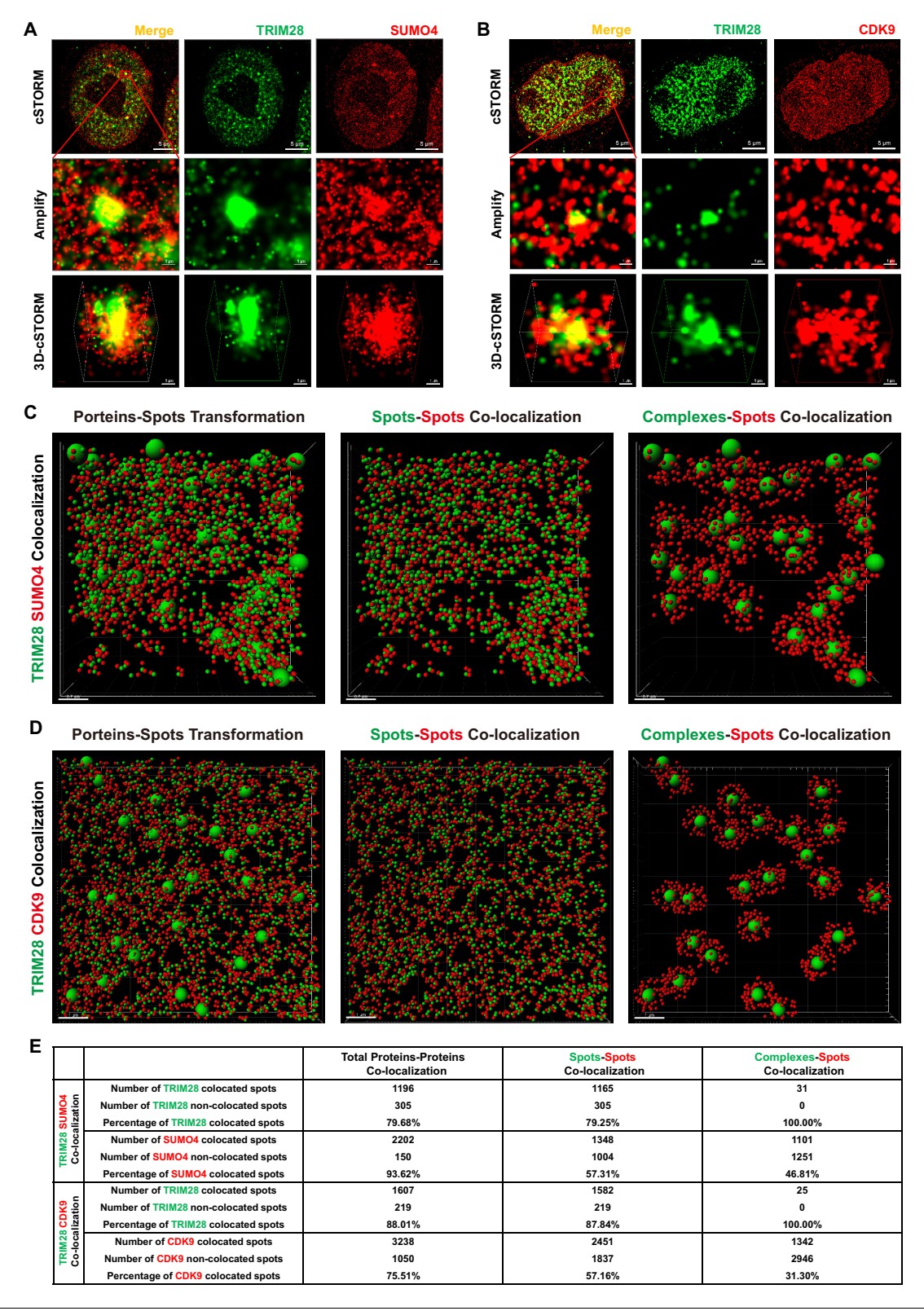

**Figure 6.** TRIM28 co-localizes with SUMO4 and CDK9. (**A**) cSTORM image of endogenous TRIM28 and SUMO4 in HEK293T cells. The first row: the original whole nucleus; the second row: one of the amplified region of the nucleus; the third row: the 3D-cSTORM image of the amplified region. Merged views of TRIM28 and SUMO4 were shown on the left column. Endogenous TRIM28 was shown in the middle column and colored green. Endogenous SUMO4 was shown in the right column and colored red. Of note, DAPI and Hoechst were not allowed to dye DNA according to cSTORM

*Figure 6 continued on next page*

*Figure 6 continued*

protocol. (**B**) cSTORM image of endogenous TRIM28 and CDK9 in HEK293T cells. Each row was shown as in (**A**). First column: merged view of TRIM28 and CDK9, yellow indicating co-localization; second column: endogenous TRIM28 which was colored green; third column: endogenous CDK9 which was colored red. (**C–D**) cSTORM-imaged protein molecules and complexes were transformed into small or large spots based on their diameter. The left panel of each figure showed the original transformation. The middle panel showed spots-spots co-localization in compliance with the criterion of maximal distance of 10 nm. The right panel showed complexes-spots co-localization in compliance with the criterion of maximal distance of 100 nm. Green spots indicated TRIM28 molecules. Red spots indicated SUMO4 or CDK9 molecules. (**E**) Quantitation of co-localization of TRIM28 with SUMO4 or CDK9. Both of total proteins-proteins, spots-spots and complexes-spots co-localizations were measured.

DOI: https://doi.org/10.7554/eLife.42426.015

The following figure supplement is available for figure 6:

**Figure supplement 1.** Amplified views of transformed co-localization.

DOI: https://doi.org/10.7554/eLife.42426.016

We found that the SUMOylation signals of Lys44, Lys56 and Lys68 totally disappeared in the absence of endogenous TRIM28, further supporting that these sites are specifically SUMOylated by TRIM28 (*Figure 9B*). The target-specific SUMO-MS for directly analyzing the enriched SUMO-CDK9 also confirmed this result (*Figure 9—figure supplement 1C–E*). As the acetylation of Lys44 is required for its kinase activity, it is not surprising that the kinase activity of CDK9 was weakened when CDK9 was SUMOylated (*Cho et al., 2010*; *Fu et al., 2007*). Interestingly, other two SUMOylated sites Lys56 and Lys68 are within the interaction region of CDK9 and Cyclin T1 based on the co-crystal structure (PDB ID: 4EC8) (*Baumli et al., 2012*) (*Figure 9C*). Because SUMO protein is a polypeptide macromolecule, its presence can form steric hindrance which prevents the formation of P-TEFb complex.

## TRIM28 depletion reactivates latent HIV-1 in cells from HIV-1-infected individuals

To verify whether TRIM28 could be a safe target for developing new LRAs, we firstly evaluated the possible toxicities associated with depleting TRIM28 in Hela cells, Jurkat cells as well as resting CD4⁺ T cells isolated from aviremic participants. We conducted several experiments which included cytotoxicity assay, cell viability assay, cell number counting and cell proliferation assay. The results showed that the depletion of TRIM28 was non-toxic to cell viability and proliferation (*Figure 10—figure supplement 1*, *Figure 10—figure supplement 2*). Afterwards, we tried to determine whether the knockdown of TRIM28 reactivated latent HIV-1 in resting CD4⁺ T cells from HIV-1-infected individuals who received suppressive cART for at least 6 months. Stimulation with αCD3/αCD28 significantly induced the expression of HIV-1 based on the quantitation of intracellular HIV-1 RNAs (*Figure 10A* and *Figure 10—figure supplement 3A–B*). The depletion of TRIM28 reactivated similar amount of HIV-1 RNA as suberanilohydroxamic acid (SAHA). After we combined the knockdown of TRIM28 with SAHA, the reactivation was more significant (*Figure 10A*). To provide evidence that SUMO4-mediated modification of CDK9 by TRIM28 is one of the mechanisms used by TRIM28 to contribute to HIV-1 latency in cells isolated from aviremic participants, we also tested whether the depletion of SUMO4 could reactivate latent HIV-1 in resting CD4⁺ T cells isolated from HIV-1-infected individuals. The results showed that the depletion of SUMO4 reactivated substantial productions of HIV-1 RNAs which were even slightly higher than those activated by SAHA. The combination use of SUMO4 knockdown and SAHA addition could reactivate more HIV-1 RNAs than those reactivated by them separately (*Figure 10—figure supplement 4*). We next examined whether the knockdown of TRIM28 reactivated more

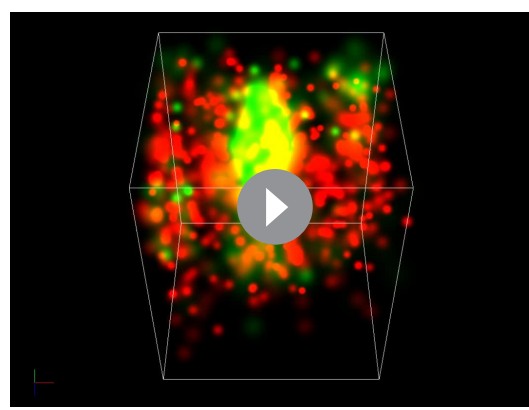

**Video 1.** 3D-cSTORM movie of the 3D co-localization of TRIM28 with SUMO4. Green spots indicate TRIM28. Red spots indicate SUMO4.

DOI: https://doi.org/10.7554/eLife.42426.017

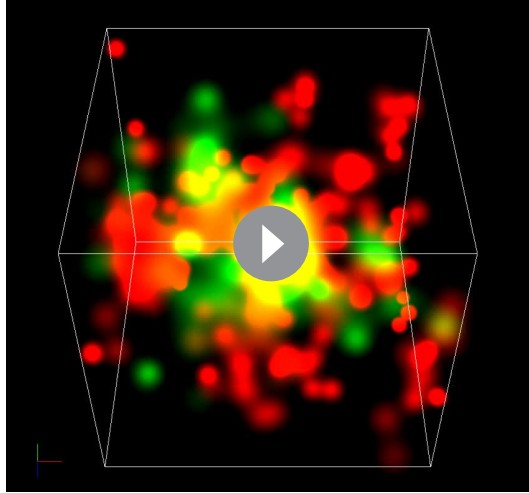

**Video 2.** 3D-cSTORM movie of the 3D co-localization of TRIM28 with CDK9. Green spots indicate TRIM28. Red spots indicate CDK9.
DOI: https://doi.org/10.7554/eLife.42426.018

**Video 3.** Transformed 3D-cSTORM movie of the 3D co-localization of TRIM28 with SUMO4. Green spots indicate TRIM28. Red spots indicate SUMO4.
DOI: https://doi.org/10.7554/eLife.42426.019

genetically-diversified HIV-1, as we described previously (*Figure 10—figure supplement 3A*) (*Geng et al., 2016b*). Although TRIM28 depletion alone reactivated similar amount of genetically diversified HIV-1 with SAHA, the combination of TRIM28 knockdown and SAHA reactivated much more genetically-diversified HIV-1 (*Figure 10B*). To determine whether the reactivated HIV-1 was replication-competent, we co-cultured the PHA-stimulated, SAHA-induced, or TRIM28-deficient resting CD4 +T cells from HIV-1-infected individuals, with PHA-activated CD4 +T cells from heathy donors (*Figure 10—figure supplement 3A*). The accumulating production of p24 antigen indicates the reactivated HIV-1 viral particles were replication-competent. The knockdown of TRIM28 reactivated replication-competent viruses in all the three samples (*Figure 10C* and *Figure 10—figure supplement 3C*). Similarly, the combination of SAHA with TRIM28 knockdown reactivated more replication-competent viral particles. These results indicate that TRIM28 contributes to HIV-1 latency in HIV-1-infected individuals. Targeting TRIM28 is well-tolerated for HIV-1-infected CD4[+] T cells.

## Discussion

TRIM28 has been found as an epigenetic adaptor which recruits multiple suppressive epigenetic modifiers to the LTRs of endogenous retroviruses (*Rowe et al., 2010*; *Wolf and Goff, 2007*). It is also identified to stabilize promoter-proximal pausing of RNAP II with some unsolved functions (*Bunch et al., 2014*). Furthermore, TRIM28 is also a SUMO E3 ligase which can mediate intramolecular SUMOylation of its bromodomain and intermolecular SUMOylation of IFN regulatory factor 7 (IRF7), resulting in the recruitment of epigenetic modifiers and the inhibition of IRF7 function, respectively (*Ivanov et al., 2007*; *Liang et al., 2011*) In this

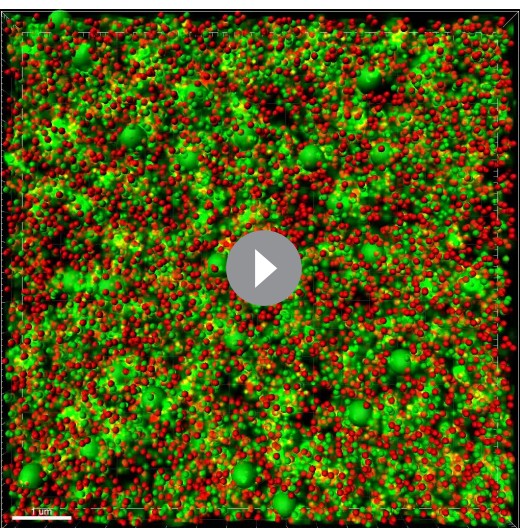

**Video 4.** Transformed 3D-cSTORM movie of the 3D co-localization of TRIM28 with CDK9. Green spots indicate TRIM28. Red spots indicate CDK9.
DOI: https://doi.org/10.7554/eLife.42426.020

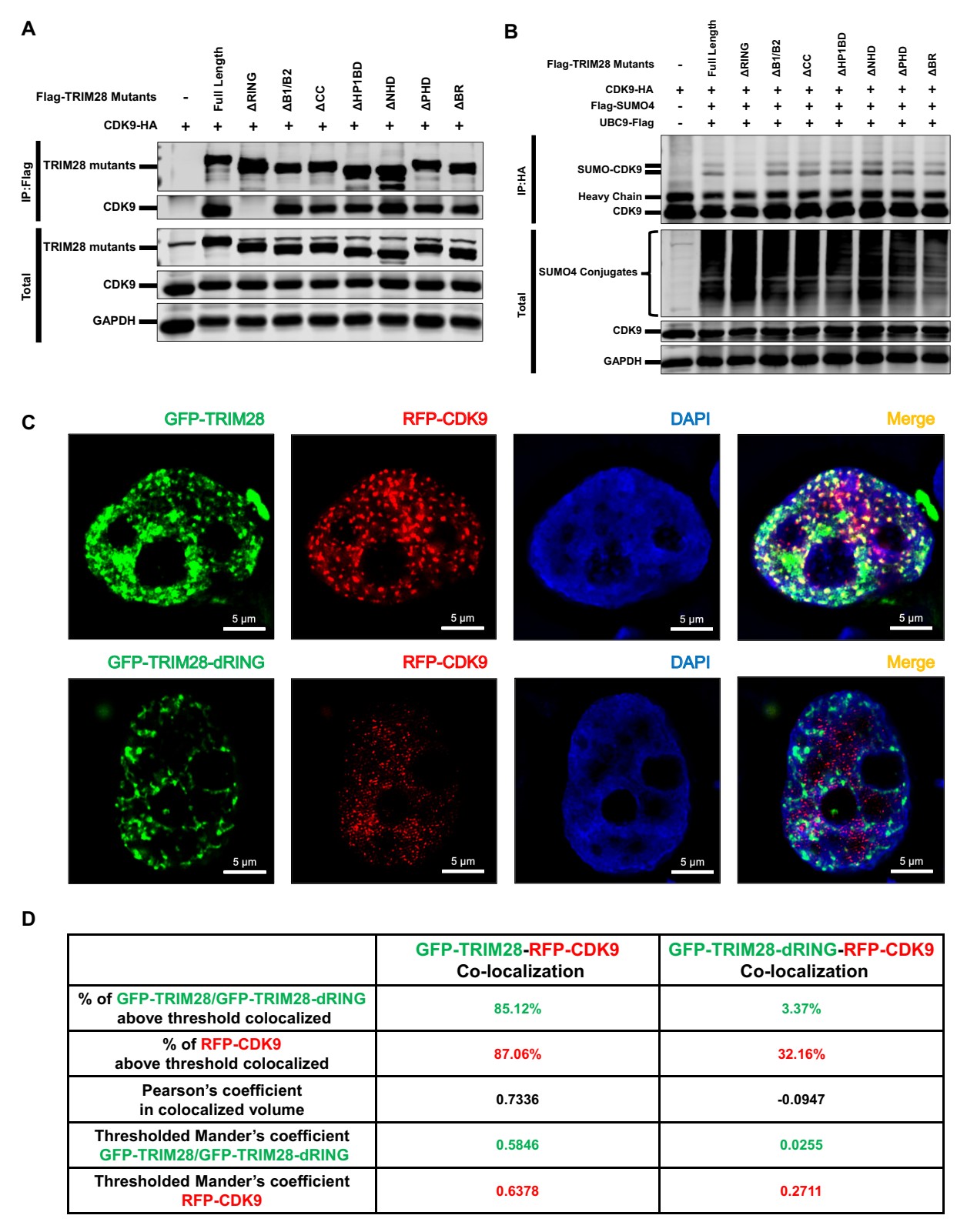

**Figure 7.** The RING domain of TRIM28 plays a key role in binding to and SUMOylating CDK9. (**A**) HA-tagged CDK9 was co-overexpressed with Flag-tagged full length TRIM28 or domain-truncated TRIM28 mutants. Flag-tagged proteins were IP, followed by IB with antibodies against HA-tag, Flag-tag and GAPDH. (**B**) HA-tagged CDK9 was co-overexpressed with Flag-tagged SUMO4, Flag-tagged UBC9, Flag-tagged full length TRIM28 or Flag-tagged domain-truncated TRIM28 mutants. CDK9 was IP with anti-HA-tag beads, followed by IB with antibodies against HA-tag, Flag-tag and GAPDH. (**C**)

*Figure 7 continued on next page*

*Figure 7 continued*

GFP-tagged TRIM28 or TRIM28-dRING mutant was co-overexpressed with RFP-tagged CDK9 in HEK293T cells. The samples were fixed and dyed according to the immunofluorescence procedure, then visualized in Nikon A1 N-SIM. DAPI was used to dye DNA which was colored into blue. (D) Quantitation of co-localization of TRIM28 or TRIM28-dRING with CDK9. The percentage of co-localization was indicated by percentage of target protein voxels above threshold co-localized voxels. Both Pearson's coefficient and thresholded Mander's coefficient were used to evaluate co-localization. For Pearson's coefficient, a value of 1 represents perfect co-localization, 0 no co-localization, and −1 perfect inverse co-localization. For thresholded Mander's coefficient, a value of 1 represents perfect co-localization and 0 no co-localization.

DOI: https://doi.org/10.7554/eLife.42426.021

The following figure supplement is available for figure 7:

**Figure supplement 1.** TRIM28 enriches CDK9 in the presence of RNase and SUMOylation status of TRIM28 mutants.

DOI: https://doi.org/10.7554/eLife.42426.022

report, we identified that TRIM28 functions not only as a well-defined epigenetic adaptor but also as a SUMO E3 ligase to SUMOylate P-TEFb complex to significantly repress HIV-1 expression and contributes to HIV-1 latency. Based on our data, we propose a model of TRIM28-mediated HIV-1 latency (*Figure 10D*). In active status, P-TEFb complex is recruited by HIV-1 Tat to the partly transcribed HIV-1 RNA trans-activation response element (TAR). P-TEFb catalytic subunit CDK9 super-phosphorylates the Ser2 residues of RNAP II, facilitating the processivity of RNAP II on the transcribing HIV-1 RNA. In latent status, TRIM28 is recruited to HIV-1 LTR and SUMOylates CDK9 in Lys44, Lys56 and Lys68, resulting in the inhibition of CDK9 kinase activity and its disconnecting with Cyclin T1. Without the super-phosphorylation on Ser2, RNAP II promoter-proximal paused at LTR. Therefore, the latent status is maintained by both TRIM28-mediated CDK9 dysfunction and TRIM28-mediated suppressive epigenetic modification on nucleosome nuc-1 which lies precisely downstream of HIV-1 promoter (*Verdin et al., 1993*).

Nevertheless, previous works reported that CDK9 and P-TEFb regulatory subunit Cyclin T1 were recruited to HIV-1 LTR by TRIM28 through 7SK snRNP bridging, although some debates existed (*D'Orso and Frankel, 2010*; *D'Orso et al., 2012*; *Mbonye and Karn, 2014*; *Mbonye and Karn, 2017*; *McNamara et al., 2016*; *Ott et al., 2011*). In contrast, we found that TRIM28 was still able to enrich CDK9 in the presence of RNase (*Figure 7—figure supplement 1A*). Instead, our results showed that TRIM28 bound to CDK9 through its RING domain. Besides, our findings regarding the effect of TRIM28 upon HIV-1 transcription are inconsistent with the observations from D'Orso's group. They found that TRIM28 facilitates RNAP II elongation by manipulating 'on-site' P-TEFb activation, resulting in quick response to stimulation and facilitating HIV-1 transcribing (*McNamara et al., 2016*). However, in a simple HIV-1 expression model, several HIV-1 latency models, and resting CD4$^+$ T cells isolated from HIV-1-infected individuals, we all found that the depletion of TRIM28 results in HIV-1 transcriptional activation and TRIM28 functions as a latency contributor rather than a stimulator in our model. We also found that the enrichment of TRIM28 on HIV-1 promoter was unchanged upon TNFα stimulation, which indicated that TRIM28 might not be controlled by TNFα signaling (*Figure 1—figure supplement 3G*). Whether these controversies are caused by cell lines, cellular conditions, or various HIV-1 integration sites as hypothesized by them, still needs to be further confirmed.

## SUMO enigma of TRIM28

Post-translational modifications of CDK9 have been studied extensively, most of which focus on phosphorylation and acetylation (*Cho et al., 2010*). Interestingly, many CDK9 SUMOylation sites which we identified here are highly related to phosphorylation and acetylation. The acetylation of Lys44 is vital for CDK9 phosphorylation activity on RNAP II. The SUMOylation of Lys44 masks the kinase activity. The acetylated Lys44 can also be deacetylated by NuRD complex which recruited by TRIM28.

Although multiple sites on CDK9 can be SUMOylated by TRIM28, the percentage of SUMOylated CDK9 is only a small proportion (less than 5%). This phenomenon has been observed for most of the identified SUMOylation targets (*Gareau and Lima, 2010*; *Impens et al., 2014*). How the small portion triggers extensive effect on target substrate remains a mystery. Two models have been proposed to explain the small fraction of SUMOylation mediated transcriptional suppression, respectively (*Hay, 2005*; *Johnson, 2004*). Both models suggest transcriptional suppression is

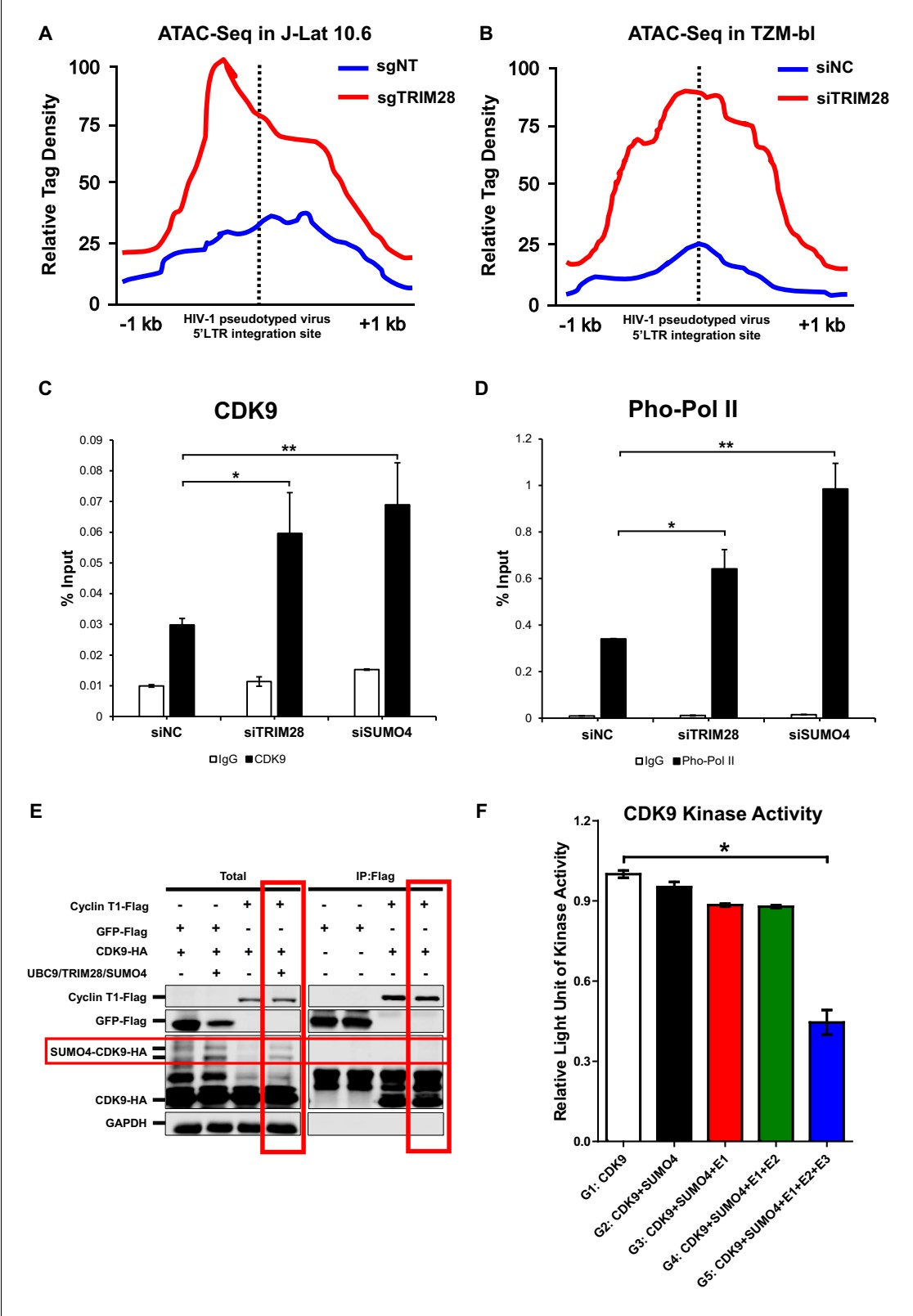

**Figure 8.** CDK9 function is reduced when SUMOylated by TRIM28. (**A–B**) TRIM28-defective (sgTRIM28) J-Lat 10.6 cell line was generated by CRISPR-CAS9 technique. ATAC-Seq was conducted with sgNT and sgTRIM28 J-Lat 10.6 cell lines, as well as siNC and siTRIM28 TZM-bl cell lines. The tag reads of the HIV-1 pseudotyped virus/minigenome 5'LTR integration sites were counted and normalized to the total mapped reads, and represented as relative tag density. The highest tag density was set as 100. Figures showed 2 kb range centered the 5'LTR integration sites. (**C–D**) ChIP assays with

*Figure 8 continued on next page*

*Figure 8 continued*

antibodies against CDK9 and Ser2 Pho-Pol II were performed in TZM-bl cell lines which were treated with siNC, siSUMO4 and siTRIM28, respectively. (E) Cyclin T1 or GFP was co-overexpressed with CDK9 in the absence or presence of SUMO4, UBC9 and TRIM28. Cyclin T1 and GFP were IP followed by IB. (F) Fold change of kinase activity when CDK9 was SUMOylated. Data represents mean ±SEM in triplicates. p-Values were calculated by Student's *t*-test. *p<0.05, **p<0.01.

DOI: https://doi.org/10.7554/eLife.42426.023

The following figure supplements are available for figure 8:

**Figure supplement 1.** The distribution and GO analysis of increased accessible regions upon TRIM28 depletion.
DOI: https://doi.org/10.7554/eLife.42426.024
**Figure supplement 2.** The COG analysis of increased accessible regions and the chromatin accessibility variations on target genes.
DOI: https://doi.org/10.7554/eLife.42426.025
**Figure supplement 3.** Schematic of in vitro SUMOylation assay and CDK9 kinase assay.
DOI: https://doi.org/10.7554/eLife.42426.026

initiated by SUMOylation. However, the maintenance of suppression is SUMOylation-independent. In our co-localization experiment, we found that CDK9 is extensively recruited to the sub-compartment shaped by TRIM28, although the SUMOylated CDK9 is only a small proportion based on the western blotting data. We propose that SUMOylation is a transient signal for CDK9 to enter to silent status or silent complex. The SUMOylated CDK9 may recruit other suppressive modifiers to stabilize the suppressive complex. After the remove of SUMO peptide by ubiquitous SENPs, CDK9 might be still sequestered in the suppressive complex. In recent years, TRIM28 was identified to form a large repressive complex with other epigenetic silencing complex such as the human silencing hub (HUSH) complex which also recruits SETDB1 to HIV-1 LTR to maintain H3K9me3 (*Robbez-Masson et al., 2018*; *Tchasovnikarova et al., 2015*). In rapid growing cells, 90% of P-TEFb is sequestered in suppressive complex 7SK snRNP (*Zhou et al., 2012*). Whether TRIM28 is part of 7SK snRNP and whether TRIM28 complex shares overlap with 7SK snRNP or other CDK9 suppressive complexes in primary CD4$^+$ T cells need to be further elucidated.

## TRIM28-mediated transcriptional-pausing

TRIM28 has previously been found to stabilize the RNAP II promoter-proximal pausing (*Bunch et al., 2014*). However, the detailed mechanism is largely unknown. Our findings here could potentially explain this phenomenon. The largest barrier for RNAP II to escape from transcriptional-pausing to effective elongation is the recruitment of P-TEFb to super-phosphorylate RNAP II. TRIM28 is bound to upstream of transcription start sites (TSSs) and SUMOylates the invaded CDK9, resulting in the disconnection of CDK9 with Cyclin T1 and inhibition of CDK9 kinase activity. This hypothesis is also consistent with our finding that the depletion of TRIM28 or SUMO4 induces more significant HIV-1 expression when combining the use of HIV-1 Tat. Without the constraint of TRIM28-mediated CDK9 SUMO4-SUMOylation, HIV-1 Tat utilizes more functional CDK9 to facilitate RNAP II on transcribing HIV-1 RNA. Another mechanisms which TRIM28 may manipulate is TRIM28-mediated suppressive epigenetic modifications on nucleosomes downstream of RNAP II pausing sites, which further stabilizes transcriptional-pausing. One report showed that SENP3 deSUMOylates RbBP5, one of the subunits of MLL1/MLL2 complexes, resulting in the complexes stabilization, H3K4me3 accumulation and RNAP II recruitment (*Nayak et al., 2014*). We found that SENP3 prevents TRIM28-mediated CDK9 SUMOylation, which facilitates the transcriptional-pausing release of recruited RNAP II. More work needs to further identify the upstream signaling pathway which determines when to release TRIM28-mediated transcriptional-pausing of RNAP II on HIV-1 LTR.

## Future development of LRAs targeting both epigenetics and transcription

Until now, nearly all the shock agents have failed to decrease the latent HIV-1 reservoir based on several clinical trials (*Spivak and Planelles, 2018*). The only effective LRAs across multiple latency model cell lines and ex vivo patient cells are protein kinase C (PKC) agonists (*Bullen et al., 2014*). However, PKC agonists induce some degree of T cell activation which is toxic to global T cells. Several lines of evidence have shown that both epigenetic regulation and transcriptional control are two barriers which we need to overcome when we develop novel LRAs (*Mbonye and Karn, 2017*).

Interestingly, we found that TRIM28 bridges both suppressive epigenetic modifications and RNAP II transcriptional-pausing to contribute to HIV-1 latency. Besides, LRAs which target the SUMOylation of transcription factor result in the reactivation of latent HIV-1 (*Bosque et al., 2017*). TRIM28-mediated RNAP II transcriptional-pausing on HIV-1 promoter is also SUMOylation-dependent as we have elucidated extensively above. Developing next-generation LRAs targeting TRIM28 may release both epigenetic and transcriptional restrictions, which also provides a new direction to search dual-function candidates.

# Materials and methods

## Key resources table

| Reagent type (species) or resource | Designation | Source or reference | Identifiers | Additional information |
|---|---|---|---|---|
| Strain, strain background (*Escherichia coli*) | *E.coli* DH5α: F⁻, φ 80dlacZ ΔM15, Δ(lacZYA -argF )U169, deoR , recA1 , endA1 , hsdR17 (rK⁻, mK⁺), phoA, supE44 , λ⁻, thi −1, gyrA96 , relA1 | Takara | Cat#9057 | |
| Strain, strain background (*Escherichia coli*) | *E. coli* HB101: F-, hsdS20(rB-, mB-), recA13, ara-14, proA2, lacY1, galK2, rpsL20 (str), xyl-5, mtl-1,supE44, leuB6, thi-1. | Takara | Cat#9051 | |
| Strain, strain background (*Escherichia coli*) | *E.coli* BL21: F⁻, ompT, hsdSB (rB⁻mB⁻), gal, dcm | Takara | Cat#9126 | |
| Strain, strain background (*Escherichia coli*) | *E.coli* Stbl3: F-, mcrB, mrr, hsdS20 (rB-, mB-), recA13, supE44, ara-14, galK2, lacY1, proA2, rpsL20 (StrR), xyl-5, λ- leu, mtl-1 | ThermoFisher | Cat#C7381201 | |
| Cell line (*Homo sapiens*) | HEK293T | ATCC | CRL-3216; RRID: CVCL_0063 | female |
| Cell line (*Homo sapiens*) | HeLa | ATCC | CCL-2; RRID: CVCL_0030 | female |
| Cell line (*Homo sapiens*) | TZM-bl | NIH AIDS Reagent Program | Cat#8129 | female |
| Cell line (*Homo sapiens*) | J-Lat 6.3 | PMID: 12682019 | NIH AIDS Reagent Program Cat#9846 | Dr. Eric Verdin (The Buck Institute for Research on Aging, Novato, CA, USA) |
| Cell line (*Homo sapiens*) | J-Lat 8.4 | PMID: 12682019 | NIH AIDS Reagent Program Cat#9847 | Dr. Eric Verdin (The Buck Institute for Research on Aging, Novato, CA, USA) |
| Cell line (*Homo sapiens*) | J-Lat 9.2 | PMID: 12682019 | NIH AIDS Reagent Program Cat#9848 | Dr. Eric Verdin (The Buck Institute for Research on Aging, Novato, CA, USA) |
| Cell line (*Homo sapiens*) | J-Lat 10.6 | PMID: 12682019 | NIH AIDS Reagent Program Cat#9849 | Dr. Eric Verdin (The Buck Institute for Research on Aging, Novato, CA, USA) |
| Cell line (*Homo sapiens*) | J-Lat 15.4 | PMID: 12682019 | NIH AIDS Reagent Program Cat#9850 | Dr. Eric Verdin (The Buck Institute for Research on Aging, Novato, CA, USA) |

*Continued on next page*

*Continued*

| Reagent type (species) or resource | Designation | Source or reference | Identifiers | Additional information |
|---|---|---|---|---|
| Biological sample (*Homo sapiens*) | Blood samples from healthy individuals | Guangzhou Blood Center, Guangzhou | http://www.gzbc.org/ | |
| Biological sample (*Homo sapiens*) | Blood samples from HIV-1-infected individuals | Department of Infectious Diseases, Guangzhou 8th People's Hospital, Guangzhou | http://gz8h.com.cn/ | |
| Antibody | Mouse Monoclonal anti-TRIM28 Antibody | Proteintech | Cat#66630–1-Ig; RRID: AB_2732886; Lot#10006062 | (1:1000) |
| Antibody | Rabbit Polyclonal anti-TRIM28 Antibody | Proteintech | Cat#15202–1-AP; RRID: AB_2209890; Lot#00051172 | (1:1000) |
| Antibody | Rabbit Polyclonal Anti-Histone H3 (tri methyl K4) Antibody | Abcam | Cat#ab8580; RRID: AB_306649; Lot#GR273043-3 | Use 2 µg for 25 µg of chromatin |
| Antibody | Rabbit Polyclonal Anti-Histone H3 (acetyl K9) Antibody | Abcam | Cat#ab4441; RRID: AB_2118292; Lot#GR270585-1 | Use 2 µg for 25 µg of chromatin |
| Antibody | Mouse Monoclonal Anti-Histone H3 (tri methyl K27) Antibody | Abcam | Cat#ab6002; Lot#GR275911-3 | Use 5 µg for 25 µg of chromatin |
| Antibody | Normal Rabbit Anti-IgG Antibody | CST | Cat#2729; RRID: AB_1031062 | Use 1 µg for 25 µg of chromatin |
| Antibody | Rabbit Polyclonal Anti-UBE2I Antibody | Abclonal | Cat#A2193; Lot#45473 | (1:1000) |
| Antibody | Rabbit Polyclonal Anti-UBA2 Antibody | Abclonal | Cat#A4363 | (1:1000) |
| Antibody | Rabbit Polyclonal Anti-SAE1 Antibody | Proteintech | Cat#10229–1-AP; RRID: AB_2182917; Lot#00040591 | (1:1000) |
| Antibody | Rabbit Monoclonal Anti-SUMO4 Antibody | Abcam | Cat#ab126606; RRID: AB_11128131; Lot#GR851138-12 | (1:1000) |
| Antibody | Rabbit Monoclonal Anti-CDK9 (C12F7) Antibody | CST | Cat#2316; Lot#6 | (1:1000) |
| Antibody | Rabbit Polyclonal Anti-SENP3 Antibody | Proteintech | Cat#17659–1-AP; RRID: AB_2301618; Lot#00025621 | (1:1000) |
| Antibody | Rabbit Polyclonal Anti-RNA polymerase II CTD repeat YSPTSPS (phosphor-Ser2) Antibody | Abcam | Cat#ab5095; RRID: AB_304749; Lot#GR278215-1 | Use 2 µg for 25 µg of chromatin |
| Antibody | Mouse Monoclonal Anti-Histone H3 (di methyl K9) Antibody | Abcam | Cat#ab1220; RRID: AB_449854 | Use 4 µg for 25 µg of chromatin |
| Antibody | Rabbit Polyclonal Anti-Histone H3 (tri methyl K9) Antibody | Abcam | Cat#ab8898; RRID: AB_306848 | Use 4 µg for 25 µg of chromatin |
| Antibody | Donkey Anti-Mouse IgG H and L (Alexa Fluor 647) Antibody | Abcam | Cat#ab150107; Lot#GR311164-3 | (1:200) |
| Antibody | Donkey Anti-Rabbit IgG H and L (Alexa Fluor 647) Antibody | Abcam | Cat#ab150075; Lot#GR3174006-4 | (1:200) |

*Continued on next page*

*Continued*

| Reagent type (species) or resource | Designation | Source or reference | Identifiers | Additional information |
|---|---|---|---|---|
| Antibody | Donkey Anti-Rabbit IgG (H + L), Highly Cross-Adsorbed, CF 568 Dye Conjugates, Single Label for STORM | Biotium | Cat#20803–500 µl; Lot#17C0626 | (1:200) |
| Antibody | Donkey Anti-Mouse IgG (H + L), Highly Cross-Adsorbed, CF 568 Dye Conjugates, Single Label for STORM | Biotium | Cat#20802–500 µl; Lot#17C1004 | (1:200) |
| Antibody | Rabbit Anti-DDDDK Tag Polyclonal Antibody, Unconjugated | MBL | Cat#PM020; RRID: AB_591224; Lot#026 | (1:1000) |
| Antibody | Mouse Monoclonal Anti-HA-Tag Antibody | MBL | Cat#M180-3; RRID: AB_10951811; Lot#008 | (1:10000) |
| Antibody | Mouse Monoclonal Anti-His-Tag Antibody | Proteintech | Cat#66005–1-Ig; RRID: AB_11232599; Lot#00083246 | (1:1000) |
| Antibody | Rabbit Polyclonal Anti-GAPDH Antibody | Proteintech | Cat#10494–1-AP; RRID: AB_2263076; Lot#00039889 | (1:10000) |
| Antibody | IRDye 680RD Goat anti-Mouse IgG (H + L), 0.5 mg Antibody | LI-COR Biosciences | Cat#926–68070; RRID: AB_10956588; Lot#C70613-15 | (1:10000) |
| Antibody | IRDye 800CW Goat Anti-Rabbit IgG, Conjugated Antibody | LI-COR Biosciences | Cat#926–32211; RRID: AB_621843; Lot#C70620-05 | (1:10000) |
| Antibody | PerCP-Cy 5.5 Mouse Anti-Human CD45RO | BD Biosciences | Cat#560607; RRID: AB_1727500; Lot#5338941 | (1:1000) |
| Antibody | APC/Cy7 anti-human CD45RA | BioLegend | Cat#304127; RRID: AB_10708419; Lot#B164612 | (1:1000) |
| Antibody | Anti-Human CD69 PE-Cy7 | ThermoFisher | Cat#25-0699-42; RRID: AB_1548714; Lot#E10154-1635 | (1:1000) |
| Antibody | Anti-Human CD62L PE-Cyanine7 | ThermoFisher | Cat#25-0629-42; RRID: AB_1257142; Lot#4291471 | (1:1000) |
| Antibody | Anti-Human CD4 FITC | ThermoFisher | Cat#11-0048-42; RRID: AB_1633390; Lot#E10526-1631 | (1:1000) |
| Antibody | PE-Cy5 Conjugated Amti-human CD25 (IL-2R) | ThermoFisher | Cat#15-0259-42; RRID: AB_1944361; Lot#E11289-102 | (1:1000) |
| Recombinant DNA reagent | VSV-G glycoprotein-expression vector | PMID: 9306402 | Addgene Plasmid #12259 | Dr. Didier Trono (School of Life Sciences, Ecole Polytechnique Fédérale de Lausanne, Lausanne, Switzerland) |
| Recombinant DNA reagent | Lentiviral packaging construct pCMVΔR8.2 | PMID: 9306402 | Addgene Plasmid #12263 | Dr. Didier Trono (School of Life Sciences, Ecole Polytechnique Fédérale de Lausanne, Lausanne, Switzerland) |

*Continued on next page*

*Continued*

| Reagent type (species) or resource | Designation | Source or reference | Identifiers | Additional information |
|---|---|---|---|---|
| Recombinant DNA reagent | Lentiviral construct vector pLKO.3G-RFP | This paper | N/A | Progenitor: pLKO.3G |
| Recombinant DNA reagent | Lentiviral construct vector lentiCRISPRv2 | PMID: 25075903 | Addgene Plasmid #52961 | Dr. Feng Zhang (Broad Institute of MIT and Harvard) |
| Recombinant DNA reagent | Plasmid: 10His-SUMO1-Q92R | This paper | *Supplementary file 3* | Progenitor: pcDNA3.1(+) |
| Recombinant DNA reagent | Plasmid: 10His-SUMO2-Q88R | This paper | *Supplementary file 3* | Progenitor: pcDNA3.1(+) |
| Recombinant DNA reagent | Plasmid: 10His-SUMO4-Q88R | This paper | *Supplementary file 3* | Progenitor: pcDNA3.1(+) |
| Recombinant DNA reagent | Plasmid: 3HA-CDK9-KKR | This paper | *Supplementary file 3* | Progenitor: pcDNA3.1(+) |
| Recombinant DNA reagent | Plasmid: 3HA-CDK9-RRK | This paper | *Supplementary file 3* | Progenitor: pcDNA3.1(+) |
| Recombinant DNA reagent | Plasmid: 3HA-CDK9-RKK | This paper | *Supplementary file 3* | Progenitor: pcDNA3.1(+) |
| Recombinant DNA reagent | Plasmid: 3HA-CDK9-KRR | This paper | *Supplementary file 3* | Progenitor: pcDNA3.1(+) |
| Recombinant DNA reagent | Plasmid: 3HA-CDK9-KRK | This paper | *Supplementary file 3* | Progenitor: pcDNA3.1(+) |
| Recombinant DNA reagent | Plasmid: 3HA-CDK9-RKR | This paper | *Supplementary file 3* | Progenitor: pcDNA3.1(+) |
| Recombinant DNA reagent | Plasmid: 3HA-CDK9-K0R | This paper | *Supplementary file 3* | Progenitor: pcDNA3.1(+) |
| Recombinant DNA reagent | Plasmids: 3HA-CDK9-K0R-RXK (X represent mutation position) | This paper | *Supplementary file 3* | Progenitor: pcDNA3.1(+) |
| Sequence-based reagent | siRNA Library | RiboBio | *Supplementary file 1*; http://www.ribobio.com/ | |
| Sequence-based reagent | ChIP-qPCR Primers | This paper | *Supplementary file 2* | |
| Sequence-based reagent | siRNA targeting TRIM28 3'UTR:5'-GCTCTGTTCTCTGTCCTGT-3' | RiboBio | http://www.ribobio.com/ | |
| Sequence-based reagent | shRNA targeting Luciferase:5'-ACCGCCTGAAGTCTCTGATTAA-3' | PMID: 29863470 | N/A | |
| Sequence-based reagent | shRNA targeting TRIM28 CDS:5'-CCAGCCAACCAGCGGAAATGTGA-3' | PMID: 18082607 | N/A | |
| Sequence-based reagent | sgRNA targeting Dummyguide (sgNT):5'-ACGGAGGCTAAGCGTCGCAA-3' | PMID: 25075903 | N/A | Dr. Feng Zhang (Broad Institute of MIT and Harvard) |
| Sequence-based reagent | sgRNA targeting TRIM28 CDS:5'-CACCGATTGAGCTGGCAGTCTCGGC-3' | PMID: 25075903 | N/A | Dr. Feng Zhang (Broad Institute of MIT and Harvard) |

*Continued on next page*

*Continued*

| Reagent type (species) or resource | Designation | Source or reference | Identifiers | Additional information |
|---|---|---|---|---|
| Sequence-based reagent | β-Actin qPCR Forward Primer:5'-GCATGGAGTCCTGTGGCA-3' | PMID: 27291871 | N/A | |
| Sequence-based reagent | β-Actin qPCR Reverse Primer:5'-CAGGAGGAGCAAT GATCTTGA-3' | PMID: 27291871 | N/A | |
| Sequence-based reagent | TRIM28 qPCR Forward Primer:5'-CTACTCAAGTG CAGAGCCCC-3' | This paper | N/A | |
| Sequence-based reagent | TRIM28 qPCR Reverse Primer:5'-GGGAAGACCTT GAAGACGGG-3' | This paper | N/A | |
| Sequence-based reagent | HIVTotRNA Forward Primer:5'-CTGGCTAACTAGG GAACCCACTGCT-3' | PMID: 27291871 | N/A | |
| Sequence-based reagent | HIVTotRNA Reverse Primer:5'-GCTTCAGCAAGCC GAGTCCTGCGTC-3' | PMID: 27535056 | N/A | |
| Sequence-based reagent | 1 st round Nest PCR Forward Primer (E00):5'-TAGAAAGAGCAGA AGACAGTGGCAATGA-3' | PMID: 27434587 | N/A | |
| Sequence-based reagent | 1 st round Nest PCR Reverse Primer (ES8B):5'-CACTTCTCCA ATTGTCCCTCA-3' | PMID: 27434587 | N/A | |
| Sequence-based reagent | 2nd round Nest PCR Forward Primer (E20):5'-GGGCCACACATGC CTGTGTACCCACAG-3' | PMID: 27434587 | N/A | |
| Sequence-based reagent | 2nd round Nest PCR Reverse Primer (E115):5'-AGAAAAATTCCCC TCCACAATTAA-3' | PMID: 27434587 | N/A | |
| Chemical compound, drug | (+)-JQ-1 | Selleckchem | Cat#S7110 | |
| Chemical compound, drug | Vorinostat (SAHA) | Selleckchem | Cat#S1047 | |
| Chemical compound, drug | Formaldehyde solution | Sigma-Aldrich | Cat#F8775-25ML | |
| Chemical compound, drug | TRIzol Reagent | ThermoFisher | Cat#15596018 | |
| Chemical compound, drug | 4',6-Diamidino-2-Phenylindole, Dihydrochloride (DAPI) | ThermoFisher | Cat#D1306 | |
| Chemical compound, drug | Cysteamine (MEA) | Sigma-Aldrich | Cat#30070–10G | |
| Chemical compound, drug | Glucose Oxidase from Aspergillus niger, Type VII, lyophilized powder, ≥100,000 units/g solid | Sigma-Aldrich | Cat#G2133-250KU | |

*Continued on next page*

*Continued*

| Reagent type (species) or resource | Designation | Source or reference | Identifiers | Additional information |
|---|---|---|---|---|
| Chemical compound, drug | Catalase from bovine liver , lyophilized powder, $\geq$10,000 units/mg protein | Sigma-Aldrich | Cat#C40-1G | |
| Chemical compound, drug | Sodium borohydride (NaBH4) | Sigma-Aldrich | Cat#213462–25G | |
| Chemical compound, drug | 16% Paraformaldehyde (formaldehyde) Aqueous Solution | Electron Microscopy Sciences | Cat#15710 | |
| Chemical compound, drug | 8% Glutaraldehyde Aqueous Solution | Electron Microscopy Sciences | Cat#16019 | |
| Chemical compound, drug | Normal Donkey Serum (NDS) | Jackson ImmunoResearch | Cat#017-000-121 | |
| Chemical compound, drug | Triton X-100 | Sigma-Aldrich | Cat#T8787-50ML | |
| Chemical compound, drug | Protease Inhibitor Cocktail (PIC) | Sigma-Aldrich | Cat#P8340-1ML | |
| Chemical compound, drug | N-Ethylmaleimide (NEM) | Selleckchem | Cat#S3692 | |
| Chemical compound, drug | EZview Red Anti-HA Affinity Gel | Sigma-Aldrich | Cat#E6779-1ML | |
| Chemical compound, drug | EZview Red Anti-FLAG M2 Affinity Gel | Sigma-Aldrich | Cat#F2426-1ML | |
| Chemical compound, drug | Anti-His-tag Agarose | Abcam | Cat#ab1231 | |
| Chemical compound, drug | Penicillin-Streptomycin, Liquid | ThermoFisher | Cat#15140122 | |
| Chemical compound, drug | L-Glutamine, 200 mM Solution | ThermoFisher | Cat#25030081 | |
| Chemical compound, drug | Fetal Bovine Serum (FBS) | ThermoFisher | Cat#10270–106 | |
| Chemical compound, drug | Phytohemagglutinin -M (PHA-M) | Sigma-Aldrich | Cat#11082132001 | |
| Peptide, recombinant protein | Recombinant Human TNF-$\alpha$ | PeproTech | Cat#300-01A | |
| Peptide, recombinant protein | Recombinant Human IL-2 | R&D Systems | Cat#202-IL-500 | |
| Peptide, recombinant protein | Recombinant Human SUMO Activating Enzyme E1 (SAE1/UBA2) | R&D Systems | Cat#E-315 | |
| Peptide, recombinant protein | Recombinant Human UBE2I/Ubc9 | R&D Systems | Cat#E2-645-100 | |
| Peptide, recombinant protein | Recombinant Human CDK9 | Abcam | Cat#ab85603 | |
| Peptide, recombinant protein | Recombinant Human SUMO4 | This paper | N/A | |
| Peptide, recombinant protein | Recombinant Human TRIM28 | Abcam | Cat#ab131899 | |
| Commercial assay or kit | SUMO Conjugation Reaction Buffer Kit | R&D Systems | Cat#SK-15 | |
| Commercial assay or kit | Human Lymphocyte Separation Kit | TBDsciences | Cat#LTS1077 | |

*Continued on next page*

*Continued*

| Reagent type (species) or resource | Designation | Source or reference | Identifiers | Additional information |
|---|---|---|---|---|
| Commercial assay or kit | BD IMag Human CD4 + T Lymphocyte Enrichment Set-DM | BD Biosciences | Cat#557939 | |
| Commercial assay or kit | Luciferase Assay System | Promega | Cat#E4550 | |
| Commercial assay or kit | SimpleChIP Enzymatic Chromatin IP Kit (Magnetic Beads) | CST | Cat#9003S | |
| Commercial assay or kit | TruePrep DNA Library Prep Kit V2 for Illumina | Vazyme | Cat#TD501 | |
| Commercial assay or kit | HIV-1 p24 ELISA Kit | Abcam | Cat#ab218268 | |
| Commercial assay or kit | ProteoSilver Plus Silver Stain Kit | Sigma-Aldrich | Cat#PROTSIL2 -1KT | |
| Commercial assay or kit | CDK9/CyclinK Kinase Enzyme System | Promega | Cat#V4104 | |
| Commercial assay or kit | ADP-GloTM Kinase Assay | Promega | Cat#V6903 | |
| Commercial assay or kit | Cell Counting Kit-8 | Dojindo | Cat#CK04; Lot#KT793 | |
| Commercial assay or kit | Zombie Violet Fixable Viability Kit | BioLegend | Cat#423113; Lot#B256957 | |
| Commercial assay or kit | CellTrace CFSE Cell Proliferation Kit - For Flow Cytometry | ThermoFisher | Cat#C34554 | |
| Software, algorithm | Prism 5 | GraphPad | https://www.graphpad.com/scientific-software/prism/ | |
| Software, algorithm | MEGA 7 | MEGA | https://www.megasoftware.net/ | |
| Software, algorithm | Cytoscape (3.6.1) | Cytoscape Consortium | RRID:SCR_015784 | |
| Software, algorithm | STRING | Cytoscape Consortium | RRID:SCR_005223 | |
| Software, algorithm | MCODE | Cytoscape Consortium | RRID:SCR_015828 | |
| Software, algorithm | BD LSRFortessa cell analyzer | BD Biosciences | http://www.bdbiosciences.com/in/instruments/lsr/index.jsp | |
| Software, algorithm | FlowJo V10 | Tree Star | https://www.flowjo.com/ | |
| Software, algorithm | Odyssey CLX Imager | LI-COR Biosciences | https://www.licor.com/bio/products/imaging_systems/odyssey/ | |
| Software, algorithm | Image Studio Lite Ver 4.0 | LI-COR Biosciences | https://www.licor.com/bio/products/software/image_studio_lite/ | |
| Software, algorithm | CFX Manager | BIO-RAD | http://www.bio-rad.com/ | |
| Software, algorithm | GloMax 96 Microplate Luminometer Software (version 1.9.3) | Promega | https://www.promega.com/resources/software-firmware/detection-instruments-software/promega-branded-instruments/glomax-96-microplate-luminometer/ | |

*Continued on next page*

*Continued*

| Reagent type (species) or resource | Designation | Source or reference | Identifiers | Additional information |
|---|---|---|---|---|
| Software, algorithm | SkanIt SW for Microplate Readers | ThermoFisher | https://www.thermofisher.com/order/catalog/product/5187139?SID=srch-srp-5187139 | |
| Software, algorithm | NIS-Elements Advanced Research microscope imaging software | Nikon | https://www.nikoninstruments.com/Products/Software | |
| Software, algorithm | PyMOL | Schrödinger | RRID:SCR_000305 | |
| Software, algorithm | FastQC | Babraham Institute | RRID:SCR_014583 | |
| Software, algorithm | Hisat2 | PMID: 25751142 | RRID:SCR_015530 | |
| Software, algorithm | DEGseq | Bioconductor | RRID:SCR_008480 | |
| Software, algorithm | gplots | R Foundation | https://www.rdocumentation.org/packages/gplots/versions/3.0.1 | |
| Software, algorithm | Bowtie2 | PMID: 22388286 | RRID:SCR_016368 | |
| Software, algorithm | Samtools | PMID: 19505943 | RRID:SCR_002105 | |
| Software, algorithm | igvtools | Broad Institute | https://software.broadinstitute.org/software/igv/igvtools | |
| Software, algorithm | Imaris (Version 9.2) | BITPLANE | RRID:SCR_007370 | |

## Study participants

Chronically HIV-1-infected participants sampled by this study were recruited from Department of Infectious Diseases in Guangzhou 8th People's Hospital, Guangzhou. The Ethics Review Board of Sun Yat-Sen University and the Ethics Review Board of Guangzhou 8th People's Hospital approved this study. All the participants were given written informed consent with approval of the Ethics Committees. The enrollment of HIV-1-infected individuals was based on the criteria of prolonged suppression of plasma HIV-1 viremia on cART, which is undetectable plasma HIV-1 RNA levels (less than 50 copies/ml) for a minimum of 6 months, and having high $CD4^+$ T cell count (at least 350 cells/$mm^3$). Blood samples from healthy individuals were obtained from Guangzhou Blood Center. We did not have any interaction with the healthy individuals or protected information, and therefore no informed consent was required.

## Cell lines

HEK293T (CVCL_0063) and HeLa (CVCL_0030) cells which were obtained from ATCC, and TZM-bl (8129) cells, which were obtained from NIH AIDS Reagent Program, were cultured in DMEM supplemented with 1% penicillin-streptomycin (ThermoFisher), 1% L-glutamine (ThermoFisher), and 10% FBS (ThermoFisher). J-Lat 6.3, 8.4, 9.2, 10.6 and 15.4 cell lines, which were originally generated from Dr. Eric Verdin (The Buck Institute for Research on Aging, Novato, CA) Laboratory, were obtained from Dr. Robert F. Siliciano (Department of Medicine, Johns Hopkins University School of Medicine, Baltimore, MD) Laboratory. All the J-Lat cell lines were cultured in RPMI 1640 supplemented with 1% penicillin-streptomycin, 1% L-glutamine, and 10% FBS. Peripheral blood mononuclear cells (PBMCs) and primary $CD4^+$ T cells, which were isolated and purified from study participants, were cultured in RPMI 1640 supplemented with 1% penicillin-streptomycin, 1% L-glutamine, and 10% FBS. 1/1000 Recombinant human interleukin 2 (IL-2) (R and D) was supplied for primary $CD4^+$ T cells to

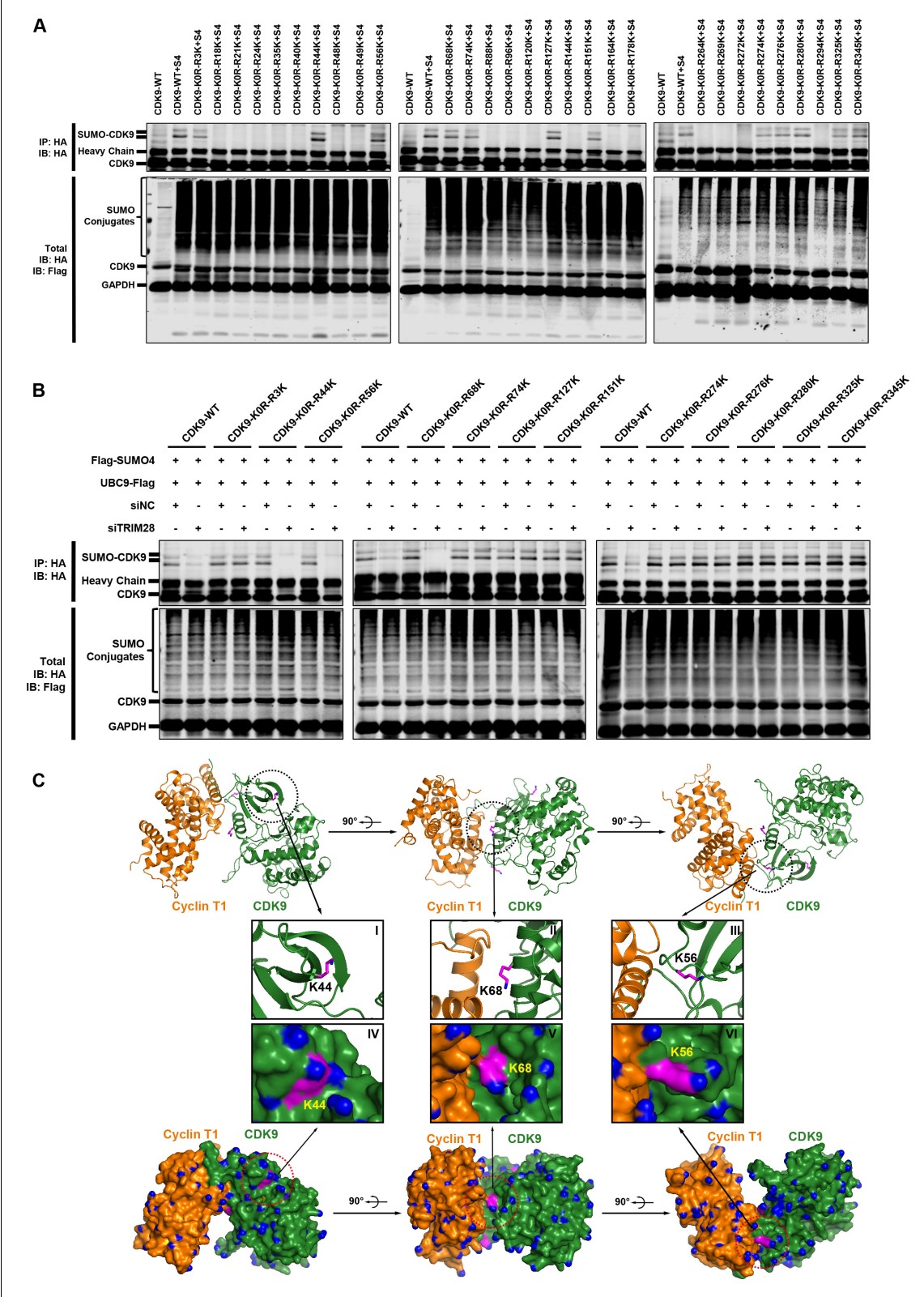

**Figure 9.** The Lys44, Lys56 and Lys68 residues of CDK9 are SUMOylated with SUMO4. (**A**) Different HA-tagged CDK9 reversing mutation constructs or wild type CDK9 were co-overexpressed with SUMO4, UBC9 and TRIM28, respectively. CDK9 and CDK9 mutants were IP with anti-HA-tag beads followed by IB. S4: SUMO4. (**B**) HA-tagged wild type CDK9 and 12 identified SUMOylation site reversing mutation constructs were co-overexpressed with Flag-tagged SUMO4 and Flag-tagged UBC9. The endogenous TRIM28 was knocked down with siRNAs. CDK9 and CDK9 mutants were IP with

*Figure 9 continued on next page*

Figure 9 continued

anti-HA-tag beads followed by IB. Asterisks represented the constructs whose SUMOylation bands disappeared upon TRIM28 knockdown. (C) Three angles of co-crystal structure of Cyclin T1 and CDK9 (PDB ID: 4EC8). Three SUMOylation sites Lys44, Lys56 and Lys68 were shown in ball-and-stick models. The two upper panels showed the ribbon models, while two lower panels showed the surface models. The inner six framed figures which numbered from I to VI represented the amplification views of Lys44, Lys56 and Lys68 sites.

DOI: https://doi.org/10.7554/eLife.42426.027

The following figure supplement is available for figure 9:

**Figure supplement 1.** The Lys44, Lys56 and Lys68 residues of CDK9 are SUMOylated with SUMO4.

DOI: https://doi.org/10.7554/eLife.42426.028

maintain proliferation. All cells have been tested for mycoplasma using a PCR assay and confirmed to be mycoplasma-free. All cells cultured in sterile incubator at 37°C and 5% $CO_2$.

## SiRNA library screening

SiRNA library targeting 182 human genes, negative control siRNA (siNC) and siRNA targeting TRIM28 3'UTR (5'-GCTCTGTTCTCTGTCCTGT-3') were purchased from RiboBio (Guangzhou, China) (**Supplementary file 1**). Three siRNAs were synthesized for each gene. The siRNAs targeting each gene were transfected as a mixture and have been validated by company to insure that at least one siRNA was able to knock down target gene mRNA up to 70%. The siRNA library covered six cellular pathways within the nucleus, which were chromatin binding, epigenetic modification, chromatin remodeling, ubiquitination, SUMOylation, and chromosome organization. Evenly mixed TZM-bl cell suspension was added into each well of 96-well plates with a Tecan Freedom EVO150 (Tecan, Män-nedorf, Schweiz) to insure that the cell confluency was 60% when the cells were transfected. Twelve hours post-seeding, cells from each well were transfected with siRNAs targeting each gene using Lipofectamine RNAiMAX (ThermoFisher) according to the manufacturer's instruction. Each gene was set three biological replicates. Forty-eight hours post-transfection, cell samples from each well were removed culture medium and washed twice with PBS. Fifty microliter passive lysis buffer (Promega) was added into each well and lysed for 30 min with shaking. The cell lysates were clarified with centrifugation at 12,000 *g* for 3 min. Luciferase in the cell lysates was measured with luciferase-reporter assay (Promega) using a multiwell plate luminometer with an auto-injector (Promega) and analyzed by GloMax 96 Microplate Luminometer Software (Promega). Fold changes were calculated for each gene compared with siNC according to the light units.

## ShRNA-mediated knockdown and CRISPR-CAS9-sgRNA-mediated knockout

ShRNA targeting luciferase (shluc: 5'-ACCGCCTGAAGTCTCTGATTAA-3') was set as negative control (*Rousseaux et al., 2018*). The shRNA target sequence against TRIM28 CDS was 5'-CCAGC-CAACCAGCGGAAATGTGA-3' (*Ivanov et al., 2007*). Target sequences were cloned into pLKO.3G-RFP which was derived from pLKO.3G. The GFP-tag was replaced with RFP-tag in pLKO.3G-RFP. Pseudotyped viral stocks were produced in HEK293T cells by co-transfecting 3 µg of VSV-G glyco-protein-expression vector, 6 µg of lentiviral packaging construct pCMVΔR8.2, and 6 µg shRNA-expression lentiviral construct using Lipofectamine 2000 (ThermoFisher) according to the manufac-turer's instruction. VSV-G glycoprotein-expression vector was abtained from Addgene (Addgene plasmid # 12259). pCMVΔR8.2 was a kindly gift from Dr. Didier Trono (School of Life Sciences, Ecole Polytechnique Fédérale de Lausanne, Lausanne, Switzerland) (*Zufferey et al., 1997*). Virus superna-tants from each 10 cm dish were concentrated into 1 ml RPMI 1640 by PEG 6000. J-Lat 6.3, 8.4, 9.2, 10.6 and 15.4 cell lines were spin-infected with shRNA virus. Forty-eight hours later, infected cells were treated with 500 nM SAHA (Selleckchem) or 1 µM JQ-1 (Selleckchem). Another 24 hr later, the percentages of GFP positive cells from each group were determined by BD LSRFortessa cell analyzer (BD Biosciences) and analyzed by FlowJo V10 (Tree Star). The infection efficiency was measured based on the percentage of RFP-positive cells using flow cytometry. The knockdown efficiency was confirmed by both qPCR and western blot.

For knocking out TRIM28, CRISPR-CAS9 system was used. SgRNA targeting dummyguide (sgNT: 5'-ACGGAGGCTAAGCGTCGCAA-3') was set as negative control (*Sanjana et al., 2014*). The sgRNA

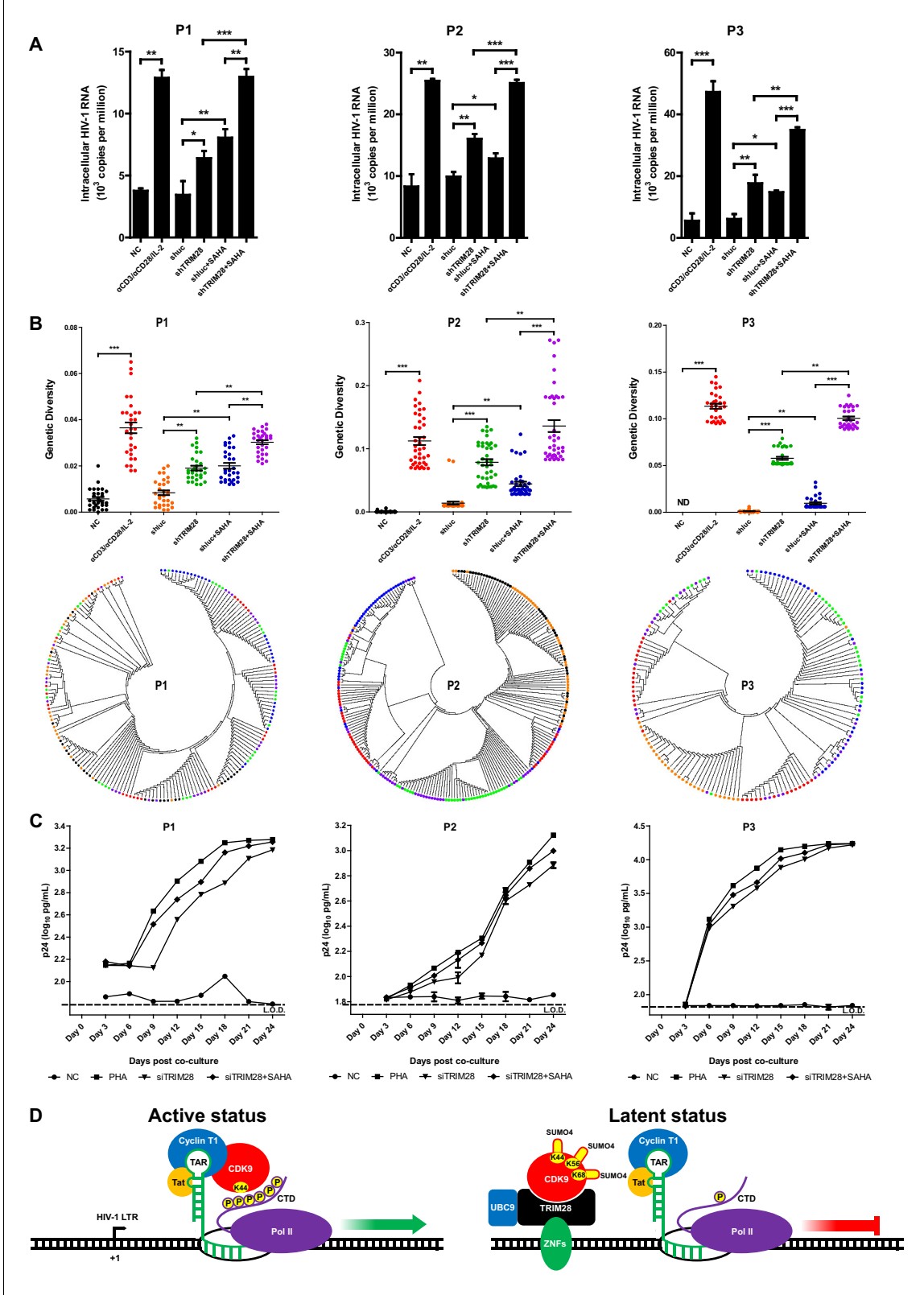

**Figure 10.** TRIM28 depletion reactivates latent HIV-1 in cells from HIV-1-infected individuals. (**A**) shRNAs targeting luciferase and TRIM28 were packaged into lentiviruses and infected CD4+ T cells from HIV-1-infected individuals. Unstimulated CD4 +T cells were used as negative control (NC). Stimulation with αCD3/αCD28/IL-2 was used as positive control. Intracellular HIV-1 RNA was isolated and quantitated by qPCR. Experiments were conducted in three HIV-1-infected individuals. (**B**) The experiment setting was as in (**A**). Envelope V1 to V3 region from intracellular HIV-1 RNAs was

*Figure 10 continued on next page*

*Figure 10 continued*

reverse-transcribed and PCR-amplified. The PCR products were TA-ligated in pMD-18 T vector. At least 60 single clones were picked from each group and sequenced. The sequences from each group were aligned and the genetic diversity index was calculated and analyzed by Mann-Whitney *U*-test. The upper panel showed the statistical analysis results. The lower panel indicated the bootstrap consensus trees which were generated based on HIV-1 sequences. *p<0.05, **p<0.01, ***p<0.001. (C) Resting CD4[+] T cells from HIV-1-infected individuals were isolated and nucleofected with siRNAs targeting negative control or TRIM28. Seventy-two hours later, PHA-stimulated uninfected CD4[+] T cells were added into each group and co-cultured for another 27 days. The supernatants were collected and half-changed every 3 days. P24 antigens in supernatants were measured with ELISA and plotted in $\log_{10}$ scale. Dashed lines indicated the limit of detection (L.O.D.) of 50 pg/ml. Triplicates were represented by mean ±SEM. (D) Schematic of TRIM28-mediated HIV-1 latency.

DOI: https://doi.org/10.7554/eLife.42426.029

The following figure supplements are available for figure 10:

**Figure supplement 1.** Cytotoxicity assay, cell viability assay and cell number counting used to evaluate the toxicity of targeting TRIM28.

DOI: https://doi.org/10.7554/eLife.42426.030

**Figure supplement 2.** Cell proliferation assay used to evaluate the toxicity of targeting TRIM28.

DOI: https://doi.org/10.7554/eLife.42426.031

**Figure supplement 3.** TRIM28 depletion reactivates latent HIV-1 in cells from HIV-1-infected individuals.

DOI: https://doi.org/10.7554/eLife.42426.032

**Figure supplement 4.** SUMO4 depletion reactivates latent HIV-1 in cells from HIV-1-infected individuals.

DOI: https://doi.org/10.7554/eLife.42426.033

target sequence against TRIM28 CDS was 5'-CACCGATTGAGCTGGCAGTCTCGGC-3' (*Sanjana et al., 2014*). Target sequences were cloned into lentiCRISPRv2 (*Sanjana et al., 2014*). Pseudotyped viruses were produced and concentrated as shRNA viruses. J-Lat 10.6 cells were spin-infected with sgRNA virus and cultured for 48 hr followed by puromycin (Sigma-Aldrich) selection. Three days post-selection, the supernatant of infected cells was replaced with fresh RPMI 1640 and infected cells were went on culturing for 2 to 7 days. The knockout efficiency was confirmed both western blot. The percentages of GFP-positive cells were determined by flow cytometry.

## ChIP-qPCR

Chromatin immunoprecipitation (ChIP) was performed according to the manufacturer's instruction (CST). Approximately $4 \times 10^6$ cells were prepared for each immunoprecipitation (IP). Briefly, TZM-bl cells were treated with siNC, siTRIM28 or TNFα (PeproTech) for 48 hr followed by crosslinking proteins to DNA with 1% formaldehyde (Sigma-Aldrich) for 10 min at room temperature. The fixation was quenched with 125 mM glycine for 5 min at room temperature followed by centrifuging at 1,500 rpm for 5 min at 4°C. The supernatants were removed immediately. Cell pellets were resuspended in ice-cold Buffer A (CST) supplemented with DTT and Protease inhibitor cocktail (PIC) and incubated on ice for 10 min. The nuclei were enriched by centrifugation at 3000 rpm for 5 min at 4°C and resuspended in ice-cold Buffer B (CST) supplemented with DTT. Nuclei pellets were centrifuged again, removed supernatants and resuspended in 100 µl Buffer B supplemented with DTT and 0.5 µl micrococcal nuclease (CST) per IP preparation. The digestion was conducted at 37°C for 20 min. Incubation tubes were inverted several times per 5 min. After digestion, the reaction was stopped by adding 50 mM EDTA followed by centrifugation at 13,000 rpm for 1 min at 4°C. Nuclei pellet was resuspended in 100 µl ChIP Buffer (CST) supplemented with PIC per IP preparation and incubated for 10 min on ice. The nuclei pellet was further lysed by sonication with 3 sets of 20 s pulses at 40% amplitude. Pellet was incubated on ice for 30 s between pulses. The lysates were clarified by centrifugation at 10,000 rpm for 10 min at 4°C. The supernatants which contained digested chromatin were transferred into new tube. One-tenth of the chromatin sample was proceeded to analyze the size and concentration. Briefly, 50 µl chromatin sample was removed RNA by RNase A (CST) and reversed cross-linking by 200 mM NaCl and Proteinase K (CST). DNA from samples were purified by DNA purification spin columns (CST). Concentration was determined by measuring $OD_{260}$. The size range was analyzed by electrophoresis on a 1% agarose gel, which should be between 150 and 900 bp.

For each IP preparation, approximately 10 µg chromatin was diluted into ChIP Buffer. Ten microliter diluted chromatin, which was 2% input sample, was transferred to a new tube and stored at −20°C. Immunoprecipitation antibodies normal rabbit IgG (CST, 2729), anti-TRIM28 antibody

(Proteintech, 15202–1-AP), anti-H3K9me2 antibody (Abcam, ab1220), anti-H3K9me3 antibody (Abcam, ab8898), anti-H3K4me3 antibody (Abcam, ab8580), anti-H3K27me3 antibody (Abcam, ab6002), anti-H3K9Acetyl antibody (Abcam, ab4441), anti-CDK9 antibody (CST, 2316), and anti-RNA polymerase II CTD repeat YSPTSPS (phospho Ser2) antibody (Abcam, ab5095) were separately added to siNC and siTRIM28 groups, respectively. The immunoprecipitation was carried out overnight at 4°C while rotating. ChIP-Grade Protein G Magnetic Beads (CST) were added to the each IP reaction and incubated with IP samples for another 2 hr at 4°Cwhile rotating. The protein G magnetic beads were pelleted by placing the IP tubes in a magnetic separation rack and washed with 3 times low-salt washes and one time high-salt wash. Each wash was conducted at 4°C for 5 min while rotating. DNA enriched by protein G magnetic beads was eluted by ChIP Elution Buffer (CST). All the DNA samples including 2% input samples were reversed cross-linking with 200 mM NaCl and Proteinase K and purified as above.

ChIP primers targeting the HIV-1 mini-model in TZM-bl cell line were used to quantitate each target by Real-Time Quantitative PCR. The quantitation regions were shown below. G5: Cellular DNA and viral 5'LTR junction; A: Nucleosome 0 assembly site; B: Nucleosome free region; C: Nucleosome one assembly site; V5: Viral 5'LTR and gag leader sequence junction; L: Luciferase region; V3: Viral poly purine tract and 3'LTR junction; G3: Viral 3'LTR and cellular DNA junction. Primers which amplified each region were shown in *Supplementary file 2*. All the ChIP-qPCR DNA signals were normalized to siNC IgG of G5. ChIP-qPCR in J-Lat 10.6 cell line was conducted as in TZM-bl cell line. In J-Lat 10.6, G5' represented cellular DNA and viral 5'LTR junction; E represented envelop; G3' represented viral 3'LTR and cellular DNA junction; A, B, C, V5 and V3 represented as in *Figure 1D*.

## RNA isolation, reverse transcription and qPCR

The identities of unstimulated primary CD4$^+$ T cells, PHA-stimulated primary CD4$^+$ T cells and resting CD4$^+$ T cells were confirmed by flow cytometry with antibodies against human CD4 (Thermo-Fisher, 11-0048-42), CD45RA (BioLegend, 304127), CD45RO (BD Biosciences, 560607), CD62L (ThermoFisher, 25-0629-42), CD69 (ThermoFisher, 25-0699-42) and CD25 (ThermoFisher, 15-0259-42). RNAs from indicated numbers of cells were isolated with TRIzol reagent (ThermoFisher) and proceeded to cDNA synthesis with PrimeScript RT reagent Kit (Takara). For the samples which quantitated the expression of TRIM28, Real-time PCR was performed with SYBR Ex-taq premix (Takara) in a CFX96 Real-time PCR Detection System (Bio-Rad). Human *β-actin* mRNA was measured as internal control (*Li et al., 2016*). Primer pairs were shown as below: *β-actin* qPCR Forward Primer: 5'-GCA TGGAGTCCTGTGGCA-3', *β-actin* qPCR Reverse Primer: 5'-CAGGAGGAGCAATGATCTTGA-3'; *TRIM28* qPCR Forward Primer: 5'-CTACTCAAGTGCAGAGCCCC-3', *TRIM28* qPCR Reverse Primer: 5'-GGGAAGACCTTGAAGACGGG-3'. The relative expression of each gene was calculated as $2^{[Ct(Control-TRIM28)-Ct(Control-\beta-Actin)]-[Ct(Treatment-TRIM28)-Ct(Treatment-\beta-Actin)]}$. For the quantitation of HIV-1 expression, a specific reverse primer was used to reversely transcribe HIV-1 RNA: 5'- GCTTCAG-CAAGCCGAGTCCTGCGTC-3'. QPCR was performed for specific reverse-transcribed HIV-1 cDNA with primer pairs: HIVTotRNA Forward Primer: 5'-CTGGCTAACTAGGGAACCCACTGCT-3' and HIV-TotRNA Reverse Primer: 5'-GCTTCAGCAAGCCGAGTCCTGCGTC-3' (*Liu et al., 2016*). After quantitation, an in vitro transcribed HIV-1 RNA was used as the external control for measuring cell-associated viral RNAs. The Ct of each group was converted to mass and further converted to copies. The final expression of intracellular HIV-1 RNA was represented as $10^3$ copies viral RNA per million CD4 +T cells.

## Global site-specific SUMO-MS

His-tagged SUMO mutants SUMO1-Q92R, SUMO2-Q88R and SUMO4-Q88R were co-overexpressed with E2 UBC9 and E3 TRIM28 in HeLa cells. Forty-eight hours post-transfection, cell pellets were lysed by guanidine lysis buffer (6 M guanidine-HCl, 100 mM sodium phosphate, and 10 mM Tris, buffered at pH 8.0). Lysates were sonicated for 15 s with 5 s pulse at a power of 30 W. Subsequently, prewashed anti-His Ni-NTA agarose beads (QIAGEN), 50 mM imidazole and 5 mM β-mercaptoethanol were added into the lysates and tumbled overnight at 4C. After overnight incubation, beads were centrifuged at 500 r.c.f. and washed for 30 min at 4C with the following wash buffers in order: wash buffer A (6 M guanidine-HCl, 0.1% Triton X-100, 10 mM imidazole, 5 mM β-mercaptoethanol, 100 mM sodium phosphate, and 10 mM Tris, buffered at pH 8.0), wash buffer B (8 M urea, 0.1%

Triton X-100, 10 mM imidazole, 5 mM β-mercaptoethanol, 100 mM sodium phosphate, and 10 mM Tris, buffered at pH 8.0), wash buffer C (8 M urea, 10 mM imidazole, 5 mM β-mercaptoethanol, 100 mM sodium phosphate, and 10 mM Tris, buffered at pH 6.3), wash buffer D (8 M urea, 5 mM β-mercaptoethanol, 100 mM sodium phosphate, and 10 mM Tris, buffered at pH 6.3), and wash buffer E (same as wash buffer D). After washing, proteins were eluted three times from beads with elution buffer (7 M urea, 500 mM imidazole, 100 mM sodium phosphate, and 10 mM Tris, buffered at pH 7.0) for 30 min at 4C. All the eluates were combined together and filtered with 0.45 μm filter (Millipore). The clarified proteins were concentrated with a 10 kDa-cutoff filter (Millipore) and washed with PBS for three times. Concentrated proteins were transferred to new tubes and boiled with 4 × protein SDS-PAGE loading buffer (Takara) at 100C for 15 min. Samples were separated with 4–12% protein gel (ThermoFisher). The gel was dyed with silver stain kit (Sigma-Aldrich). Sixteen gel slices were cut out and proceeded to in-gel digestion.

Briefly, gel slices were destained and treated with 10 mM DTT followed by the treatment of 55 mM iodoacetamide. The gels were washed with 25 mM $NH_4HCO_3$ and 25 mM $NH_4HCO_3$ in 50% ACN followed by desiccation with vacuum. One hundred nanogram trypsin (ThermoFisher) which was dissolved in 25 mM $NH_4HCO_3$ was added to each gel and incubated overnight at 37C. Twenty four hours later, digested peptides were extracted with the following extraction solutions in order: 50% ACN containing 5%TFA, 75% ACN containing 0.1% TFA, and 100% ACN. The extracts were subjected to vacuum for 3 hr to remove the solvent. The peptides were desalted and enriched by C18 ZipTip (Millipore), and redissolved in 50% ACN containing 0.1% TFA, followed by vacuum to remove the solvent. Twelve microliter of 0.01% formic acid was used to resolve the peptides and proceeded to nanoscale LC-MS/MS with an EASY-nLC system (ThermoFisher) connected to a Q-Exactive (ThermoFisher) with higher collisional dissociation (HCD) fragmentation. Peptide were separated by 20-cm-long analytical columns (ID 75 μm, Polymicro Avantes) packed in house with Luna 3.0u C18 (2) 100A (Phenomenex) with a 90-min gradient from 3% to 90% acetonitrile in 0.1% formic acid and a flow rate of 300 nL/min. Data-dependent acquisition mode with a top-ten method was used to operate the mass spectrometer. Full-scan MS spectra were obtained with a target value of 3E6, a resolution of 70,000, with a scan range from 300 to 1,800 m/z. HCD tandem MS/MS spectra were obtained with a target value of 1E6, a resolution of 17,500, and a normalized collision energy of 25%. Unknown charges, or charges lower than two and higher than eight were rejected.

## Target-specific SUMO-MS

To confirm the SUMOylation sites on CDK9 by SUMO-MS, two different tagged SUMO4 mutants were used to co-overexpressed with HA-tagged CDK9, respectively, which were Flag-tagged SUMO4-Q88R and His-tagged SUMO4-Q88R. Anti-HA-tag beads (Sigma-Aldrich) were used to immunoprecipitate CDK9 and corresponding SUMO-CDK9. Enriched target proteins were eluted from beads by boiling with 4 × protein SDS PAGE loading buffer at 100°C for 15 min. The supernatants containing target proteins were transferred to new tubes after centrifugation at 12,000 rpm for 3 min. One part of the samples was proceeded to western blot with antibodies against HA-tag, Flag-tag and His-tag to determine the SUMOylation efficiency. The left samples were separated with 4–12% SDS-PAGE protein gel and developed with silver staining. Stained bands which indicated the SUMOylated CDK9 were cut out and proceeded to in-gel digestion as above. LC-MS/MS was used to analyze the SUMOylated peptides as we have described in Global site-specific SUMO-MS.

## Co-immunoprecipitation and western blot

For all the SUMOylation-related co-immunoprecipitation (Co-IP), different tagged protein-expression constructs were transfected into Hela cells which were cultured in 6 cm dishes. Forty-eight hours post-transfection, cells were washed twice with PBS and lysed with NP-40 lysis buffer (10 mM Tris-HCl buffered at pH 7.5, 150 mM NaCl, 0.5% NP-40, 1% Triton X-100, 10% Glycerol, 2 mM EDTA, 1 mM NaF, 1 Mm $Na_3VO_4$) supplemented with 1/100 protease inhibitor cocktail (PIC) (Sigma-Aldrich) and 2 M N-Ethylmaleimide (NEM) (Selleckchem) for 30 min on ice. Every 10 min, the incubation tubes were inverted several times. The lysates were clarified by centrifugation at 12,000 rpm for 10 min at 4°C, followed by incubating with anti-HA-tag beads (Sigma-Aldrich), anti-Flag-tag beads (Sigma-Aldrich) or anti-His-tag beads (Abcam) for 4 hr to overnight at 4°C while rotating. The next day, proteins which were enriched by beads were washed for five times with ice-cold STN IP wash

buffer (10 mM Tri-HCl buffered at pH 7.5, 150 mM NaCl, 0.5% NP-40, 0.5% Triton X-100) and eluted by boiling with 4 × protein SDS-PAGE loading buffer at 100°C for 15 min. The supernatants containing target proteins were transferred to new tubes after centrifugation at 12,000 rpm for 3 min, followed by western blot with antibodies against HA-tag (MBL, PM020), Flag-tag (MBL, M180-3), His-tag (Proteintech, 66005–1-Ig) or other indicated antibodies. GAPDH (Proteintech, 10494–1-AP) was set as internal reference. 680RD goat anti-mouse IgG antibody (LI-COR Biosciences, 926–68070) and 800CW goat anti-rabbit IgG antibody (LI-COR Biosciences, 926–32211) were used as secondary antibodies. The western blot membranes were developed with Odyssey CLX Imager (LI-COR Biosciences) and analyzed by Image Studio Lite Ver 4.0 (LI-COR Biosciences).

## SUMOylation and in vitro SUMOylation assay

For a given protein, the SUMOylated form is only a small proportion. To enhance the SUMOylation signals, we conducted several SUMOylation assay by co-overexpression target proteins with SUMOylation system components which were SUMOs, E1 SAE1/UBA2, E2 UBC9, and E3 TRIM28. In vertebrates, there are four well-studied SUMO paralogs, SUMO1, SUMO2, SUMO3, and SUMO4. Because SUMO2 and SUMO3 share highly sequence identity and have similar functions, they are referred to as SUMO2/3. In preliminary data, we found the overexpression of E1 had little influence on the SUMOylation due to the high expression of endogenous E1. Therefore, we omitted E1 in the following SUMOylation assays. Besides, there are lots SUMO-specific isopeptidases (SENPs) which deSUMOylate substrates. Thus we used mature SUMO polypeptides instead of immature ones. For CDK9 SUMOylation assay, 2 μg HA-tagged wild type or mutated CDK9-expression plasmids, 4 μg Flag-tagged SUMO4-expression plasmids, 500 ng Flag-tagged UBC9 and 500 ng Flag-tagged TRIM28 were co-transfected into Hela cells which cultured in 6 cm dishes. Forty-eight hours post-transfection, cells were harvested in NP-40 lysis buffer containing 2 M NEM which was used to prevent deSUMOylation. Co-IP and western blot against HA-tagged CDK9 was performed according to the procedure which we mentioned above. For SENP3-mediated deSUMOylation assay, 500 ng or 1 μg SENP3-expression plasmids were additionally co-overexpressed with indicated amount of CDK9, SUMO4, UBC9 and TRIM28. Specific antibodies against SENP3 (Proteintech, 17659–1-AP) was used in western blot to confirm the expression.

In vitro SUMOylation assay was performed by co-culturing in vitro-purified 1 μg CDK9 (Abcam) with in vitro-purified 4 μg SUMO4 (This paper), 500 ng E1 (SAE1/UBA2) (R and D), 500 ng UBC9 (R and D) or 500 ng TRIM28 (Abcam) in SUMO conjugation reaction buffer (R and D). The reaction was initiated by adding 1 mM Mg-ATP solution and incubated for 3 hr at 30°C, followed by adding stop buffer to terminate the reaction. Samples were boiled with SDS-PAGE loading buffer supplemented with 1 M DTT for 15 min at 100°C and proceeded to western blot with specific antibodies against CDK9 (CST, 2316), SUMO4 (Abcam, ab126606), SAE1 (Proteintech, 10229–1-AP), UBA2 (Abclonal, A4363), UBC9 (Abclonal, A2193), and TRIM28 (Proteintech, 15202–1-AP).

## SIM and STORM imaging

For samples used for super-resolution Structured Illumination Microscopy (SIM) imaging, HEK293T cells were plated into Lab-Tek II chambered coverglass (ThermoFisher) which was pretreated with poly-lysine (Sigma-Aldrich). Twelve hours later, cells were transfected with GFP-tagged TRIM28 or GFP-tagged TRIM28-dRING with RFP-tagged CDK9. Twenty-four hours post-transfection, cells were washed with PBS once and fixed with 3% paraformaldehyde (Electron Microscopy Sciences)/0.1% glutaraldehyde (Electron Microscopy Sciences) for 10 min at room temperature (RT). Fixed samples were reduced with 0.1% NaBH$_4$ (Sigma-Aldrich) for 7 min at room temperature while shaking, followed by washing with PBS for 3 times at room temperature, 5 min per wash. Cells were further permeabilized with 0.2% Triton X-100 (Sigma-Aldrich) for 15 min and blocked with 10% normal donkey serum (NDS) (Jackson ImmunoResearch)/0.05% Triton X-100 for 90 min at RT. After blocking, samples were washed with 1% NDS/0.05% Triton X-100 for 15 min at RT for five times. Then, samples were wash with PBS once for 5 min, followed by post-fixation for 10 min with 3% paraformaldehyde/ 0.1% glutaraldehyde. After post-fixation, samples were washed with PBS for three times, 5 min per wash. 4', 6-Diamidino-2-Phenylindole, Dihydrochloride (DAPI) (ThermoFisher) solution was added into samples to dye DNA for 10 min while shaking. Finally, samples were washed with PBS for three times and imaged on an Eclipse Ti inverted microscope equipped with a CFI Apo TIRF objective (NA

1.49, oil immersion) and NIS-Elements AR software, an sCMOS camera (Hamamatsu Flash 4.0, 6.5 µm × 6.5 µm pixel size), and four lasers named SIM 405, SIM 488, SIM 561 and SIM 647. The original images were acquired with 512 × 512 resolution and reconstructed to form the SIM images with 1024 × 1024 resolution. The lateral resolution of the SIM image is 115 nm and the axial resolution is 300 nm. Z-step size was set to 0.20 µm. For each focal plane, 15 images (five phases, three angles, 3D-SIM mode) were captured with the NIS-Elements software. SIM images were reconstructed and analyzed with the N-SIM module of the NIS-Elements Advanced Research software (Nikon). For the quantitation of co-localization, SIM images were further analyzed with Imaris software (Version 9.2) (BITPLANE) using Coloc toolbar. Percentages of each channel voxels above threshold co-localized were calculated. Both Pearson's coefficient and thresholded Mander's coefficient were calculated to indicate the qualities of co-localization. For Pearson's coefficient, a value of 1 represents perfect co-localization, 0 no co-localization, and −1 perfect inverse co-localization. For thresholded Mander's coefficient, a value of 1 represents perfect co-localization and 0 no co-localization.

For samples used for super-resolution continuous STochastic Optical Reconstruction Microscopy (cSTORM) imaging, cells were plated, fixed, reduced, permeabilized, blocked and washed as in SIM samples preparation. After blocking, primary antibodies against TRIM28 (Proteintech, 66630–1-Ig), SUMO4 (Abcam, ab126606) and CDK9 (CST, 2316) were incubated with cells for 60 min at RT in 5% NDS/0.05% Triton X-100. Samples were washed for five times with 1% NDS/0.05% Triton X-100 at RT, 15 min per wash. Then, cells were incubated with secondary antibodies diluted in 5% NDS/0.05% Triton X-100 for 30 min at RT while shaking. Two sets of secondary antibody pairs were used to confirm the specificity, which were: Donkey Anti-Mouse IgG H and L (Alexa Fluor 647) Antibody (Abcam, ab150107) combining with Donkey Anti-Rabbit IgG H and L (CF 568) Antibody (Biotium, 20803–500 µl), Donkey Anti-Rabbit IgG H and L (Alexa Fluor 647) Antibody (Abcam, ab150075) combining with Donkey Anti-Mouse IgG H and L (CF 568) Antibody (Biotium, 20802–500 µl). After incubation, cells were washed as above followed by another wash with PBS for 5 min. Post-fixation was performed with 3% paraformaldehyde/0.1% glutaraldehyde for 10 min without shaking. Then, cells were washed with PBS for three times, 5 min per wash, followed by washing with water for two times, 3 min per wash. Of note, DAPI and Hoechst were not allowed to dye DNA according to cSTORM protocol. cSTORM imaging buffer was freshly prepared as below. GLOX solution was compounded by mixing 100 µl of 70 mg/ml Glucose Oxidase (Sigma-Aldrich) diluted in Buffer A (10 mM Tris-HCl buffered at pH 8.0, 50 mM NaCl) with 25 µl of 17 mg/ml Catalase (Sigma-Aldrich) diluted in Buffer A. One mole per liter of Cysteamine (MEA) (Sigma-Aldrich) was compounded by diluting 77 mg of MEA into 1 ml 0.25 N HCl. On ice, cSTORM imaging buffer was compounded by mixing 7 µl of GLOX, 70 µl of 1M MEA, and 620 µl of Buffer B (50 mM Tris-HCl buffered at pH 8.0, 10 mM NaCl, 10% Glucose). Each well of Lab-Tek II chambered coverglass was added 700 µl of imaging buffer which was able to be used for 2 hr. Samples were imaged under a Nikon N-STORM super-resolution microscope equipped with a high-numerical-aperture (high-NA) 100 × oil immersion objective (Nikon CFI SR Apochromat TIRF 100 × oil, 1.49 NA), a high-sensitivity and high-resolution sCMOS camera (Hamamatsu Flash 4.0, 6.5 µm × 6.5 µm pixel size, and an 0.4 × relay lens to match the pixel size under STORM mode), and four lasers with excitation wavelengths of 405, 488, 561 and 647 nm. For cSTORM which we used here, 405 nm laser was used as activation laser. 488 nm, 561 nm and 647 nm lasers were used as reporter lasers. The lateral resolution of the cSTORM image is 20 nm and the axial resolution is 50 nm. The z position was maintained during the acquisition by a Nikon 'perfect focus system'. 20,000 to 25,000 frames were taken for each image. Single molecule localization was obtained by Gaussian fitting using the STORM plug-in of NIS-Elements Advanced Research software taking into account both drift and chromatic aberrations. For the quantitation of co-localization, cSTORM images were further analyzed with Imaris software (Version 9.2) (BITPLANE) by measuring the distance of spots-spots center. cSTORM-imaged protein molecules and complexes were transformed into small or large spots based on their diameter. The spots-spots co-localization was defined by the criterion of maximal distance of 10 nm. The complexes-spots co-localization was defined by the criterion of maximal distance of 100 nm. The percentages of co-localization were calculated for both total proteins-proteins co-localization, spots-spots co-localization and complexes-spots co-localization for each protein.

## CDK9 kinase assay

In vitro SUMOylation assay was performed for CDK9 as described above. Five groups were set: Group 1 (G1): CDK9 only; Group 2 (G2): CDK9 and SUMO4; Group 3 (G3): CDK9, SUMO4 and E1 (SAE1 and UBA2); Group 4 (G4): CDK9, SUMO4, E1 and E2 (UBC9); Group 5 (G5): CDK9, SUMO4, E1, E2 and E3 (TRIM28). The reaction was terminated by stop buffer. To initiate the CDK9 kinase assay, CDK9 substrate PDKtides and ATP were added into each samples according to the manufacturer's instruction (Promega). The reaction was incubated for 120 min at room temperature followed by ADP-Glo kinase assay (Promega). Briefly, ADP-Glo reagent was added into the reaction to deplete the remaining ATP. Samples were incubated at room temperature for 40 min. After ATP depletion, kinase detection reagent was added into samples to convert the ADP which was consumed during CDK9 kinase assay to ATP. This reaction was performed by incubating samples at room temperature for 30 min. Finally, the newly synthesized ATP was quantitated using luciferase/luciferin reaction. The luminescence generated during luciferase/luciferin reaction was recorded with integration time of 0.5 to 1 s. The relative light units were calculated by normalizing to untreated wild-type CDK9 group.

## Toxicity assay

TRIM28 in Hela cells and HIV-1-infected CD4$^+$ T cells was knocked down by siRNA targeting TRIM28. ShRNA and sgRNA lentiviruses targeting TRIM28 were used to knock down TRIM28 and knock out TRIM28 in J-Lat 10.6, respectively. The cytotoxicity assay was conducted by incubating Cell Counting Kit-8 (CCK-8) reagents (Dojindo, CK04) with wild type and TRIM28-deficient cells for 3 hr followed by measuring the absorbance at 450 nm using a microplate reader. The cell viability assay was conducted by measuring the percentage of amine-reactive fluorescent dye (BioLegend, 423113) non-permeant cells, which indicated the percentage of viable cells. Cell numbers were recorded every 2 days for both wild-type and TRIM28-deficient cells. The proliferation assay was conducted by staining live cells with CFSE (ThermoFisher, C34554). On Day 0, cells from each group were stained with CFSE. The percentage and mean fluorescence intensity (MFI) of CFSE-positive cells were analyzed by flow cytometry every 2 days.

## Virus out-growth assay

Resting CD4 +cells were isolated from HIV-1-infected individuals who underwent suppressive cART for at least 6 months with undetectable plasma HIV-1 RNA (less than 50 copies/ml) and high CD4$^+$ T cell count (at least 350 cells/mm$^3$) (Human Lymphocyte Separation Kit, TBDsciences; BD IMag Human CD4$^+$ T Lymphocyte Enrichment Set-DM, BD Biosciences). These CD4$^+$ T cells were nucleofected with siRNAs targeting negative control and TRIM28 respectively, and cultured in Super T Cell Medium (STCM) consisting of RPMI 1640 supplemented with 1% penicillin-streptomycin, 1% L-glutamine, 10% FBS, 100 U/ml IL-2, and 2% T-cell growth factor (TCGF) from the supernatants of mitogen-activated healthy PBMCs treated with 2 µg/ml PHA-M and 5 ng/ml PMA for 4 hr. Six hours post-transfection, supernatants were replaced with new culture medium. Twenty-four hours later, half of siNC-treated cells were separated and supplemented with 0.5 µg/ml PHA-M (Sigma-Aldrich). Half of siTRIM28-treated cells were separated and supplemented with 500 nM Vorinostat (SAHA) (Selleckchem). Another 24 hr later, supernatants from each group were changed with fresh culture medium to prevent the toxicity of the PHA-M or SAHA. Seventy-two hours post-transfection, cells were exposed to 20 Gy X-ray irradiation for 5 min and supplemented with PHA-activated healthy CD4 +T cells. The supernatants were collected and half-changed with fresh STCM every 3 days. Cell suspension was half-changed with PHA-stimulated healthy CD4$^+$ T cell suspension every 6 days. All the supernatants from each time points and each groups were measured for the presence of HIV-1 antigen with HIV-1 p24 ELISA kit (Abcam) according to the manufacturer's instruction by SkanIt SW for Microplate Readers (ThermoFisher).

## Genetic diversity analysis

The genetic diversity of HIV-1 quasispecies under different conditions was evaluated by sequencing the *envelope* V1-V3 region. HIV-1 RNAs from each group were reverse-transcribed by specific primer ES8B: 5'-CACTTCTCCAATTGTCCCTCA-3'. Two rounds of nested PCR were performed to amplify V1-V3 region with the following primer pairs: 1$^{st}$ round Nest PCR Forward Primer (E00): 5'-

TAGAAAGAGCAGAAGACAGTGGCAATGA-3', 1st round Nest PCR Reverse Primer (ES8B): 5'-CAC TTCTCCAATTGTCCCTCA-3'; 2nd round Nest PCR Forward Primer (E20): 5'-GGGCCACACATGCC TGTGTACCCACAG-3', 2nd round Nest PCR Reverse Primer (E115): 5'-AGAAAAATTCCCCTCCA-CAATTAA-3' (*Geng et al., 2016b*). For each PCR reaction, Phanta Max Super-Fidelity DNA Polymerase (Vazyme) was used to amplify the V1-V3 region of HIV-1 *envelope* in order to ensure the fidelity. The amplification error rate of Phanta Max is 53-fold lower than that of *Taq* and 6-fold lower than that of *Pfu* according to the manufacturer's instruction. After two rounds of nested PCR utilizing Phanta Max, the PCR products were proceeded to deoxyadenosine (A)-tailing at the 3'-end of the PCR products utilizing *Ex Taq* DNA polymerase (Takara) without thermal cycling as follows: 95°C, 5 min; 72°C, 30 min; 4°C hold. The A-tailed PCR products were TA-ligated into pMD-18T vector. To minimize the sampling bias, single genome amplification method was performed by obtaining 30 independent PCR products from each sample. At least 60 single clones were picked from each group and proceeded to Sanger sequencing. The sequences from each group were aligned using MUSCLE. The sequences with ambiguous positions were removed. The average genetic distance between one give clone and the relevant entire population were calculated by MEGA seven and represented as genetic diversity index. The Mann-Whitney *U*-test was performed to compare the genetic diversity indexes between different groups using Prism 5. The phylogenetic bootstrap consensus trees were generated for each samples using neighbor-joining method with 1000 bootstrap replications implemented in MEGA seven to depict the global landscape of HIV-1 diversity.

## RNA-Seq and ATAC-Seq

Freshly isolated CD4+ T cells were stimulated with PHA for 2 days or left untreated. Total RNAs from each group were extracted by TRIzol Reagent (ThermoFisher) according to the manufacturer's instruction. The quality of RNA samples were evaluated by Nanodrop 2000 (ThermoFisher) and Bio-Analyzer 2100 (Aglient). The RNA-Seq library were built with TruSeq Stranded mRNA Library Prep Kit (Illumina) and sequenced with HiSeq X Ten (Illumina) at BioMarker (Beijing, China) under the PE150 protocol. RNA-Seq reads were trimmed, filtered and quality-controlled by FastQC (Babraham Institute) tool. The reads were aligned to human reference genome NCBI build 38 (GRCh38) by Hisat2 (*Kim et al., 2015*), followed by calculating the reads per kilobase per million mapped reads (RPKM). Differentially expressed genes were filtered by DEGseq (Bioconductor) tool with log2FC of 1 and PvalueFDR cutoff of 0.05, and plotted as heatmap or volcanoplot by gplots (R Foundation).

TRIM28-defective (sgTRIM28) J-Lat 10.6 cell line was generated by CRISPR-CAS9 technique. ATAC-Seq was conducted with sgNT and sgTRIM28 J-Lat 10.6 cell lines, as well as siNC and siTRIM28 TZM-bl cell lines. The ATAC-Seq library was built with TruePrep DNA Library Prep Kit V2 (Vazyme) as previously described (*Buenrostro et al., 2013*). Briefly, approximately 30,000 cells were harvested, washed with ice-cold PBS, and lysed with 50 µl of ice-cold lysis buffer (10 mM Tris-HCl buffered at pH 7.4, 10 mM NaCl, 3 mM MgCl$_2$, 0.1% Igepal CA-630) for 10 min on ice. The lysates were centrifuged for 5 min at 500 G, 4°C. The supernatants were carefully removed. Transposition reaction mix, which consisted of 10 µl of 5 × TTBL, 5 µl of TTE Mix V50 and 35 µl of ddH$_2$O, was used to resuspend nuclei pellet and incubated at 37°C for 30 min. The transposed DNA was purified by VAHTS DNA Clean Beads (Vazyme) and PCR-amplified with the following mixture: 24 µl of purified DNA, 10 µl of 5 × TAB, 5 µl of PPM, 5 µl of N5 primer, 5 µl of N7 primer, and 1 µl of TAE. Thermal cycle was as follows: 72°C for 3 min; 98°C for 30 s; and thermocycling at 98°C for 15 s, 60°C for 30 s and 72°C for 3 min; following by 72°C 5 min. The amplified ATAC-Seq library was purified with VAHTS DNA Clean Beads and eluted with 30 µl ddH$_2$O. The library quality was evaluated by Qubit 3.0 Fluorometer (ThermoFisher) and BioAnalyzer 2100 (Aglient), and sequenced with HiSeq X Ten (Illumina) at BioMarker (Beijing, China) under the PE150 protocol. ATAC-Seq reads were trimmed, filtered and quality-controlled by FastQC tool. Then the reads were aligned to GRCh38 by Bowtie2 (*Langmead and Salzberg, 2012*), followed by rearranging with Samtools (*Li et al., 2009*). The reads were also separately aligned to HIV-1 reference genome K03455, M38432 (Version K03455.1) by Bowtie2, followed by rearranging with Samtools. Igvtools (Broad Institute) was used to visualize the tag peaks. Specific gene loci was amplified. Tag density from different groups was calculated by normalizing to the total mapped reads. The highest tag density was set as 100. Relative tag densities of two kilobases range centered HIV-1 5'LTR integration sites were calculated and compared with sgNT or siNC.

## Statistical analysis

Triplicates data were presented as mean ±SEM. A value of $p < 0.05$ was considered to be statistically significant and represented as asterisk (*). Value of $p < 0.01$ was considered to be more statistically significant and represented as double asterisks (**). Value of $p < 0.001$ was considered to be the most statistically significant and represented as triple asterisks (***). For the comparison of ChIP, the GFP-positive percentages and qPCR experiments, standard t test was used. For the comparison of genetic diversity index experiment, Mann-Whitney U-test was used. Statistical analyses were conducted with Prism 5 (GraphPad). The network analysis and clustering analysis were conducted with STRING and MCODE in Cytoscape (Cytoscape Consortium). Co-crystal structure of Cyclin T1 and CDK9 (PDB ID: 4EC8) were reconstituted in PyMOL (Schrödinger) (*Baumli et al., 2012*). Both ribbon models and surface models were used to present the structure.

## Acknowledgements

We thank the members of Zhang, Deng and Gao laboratories for good discussions. We also thank Dr. Juntao Gao from TsingHua University for suggestions on super resolution imaging processing. We also thank clinicians from department of infectious diseases of Guangzhou Eighth People's Hospital for recruiting study participants and collecting samples. We also honor and thank all the HIV-1-infected participants, whose supports impel us feel obligated to go on HIV-1 research.

## Additional information

### Funding

| Funder | Grant reference number | Author |
|---|---|---|
| National Special Research Program of China for Important Infectious Diseases | 2018ZX10302103 | Hui Zhang |
| National Special Research Program of China for Important Infectious Diseases | 2017ZX10202102 | Hui Zhang |
| National Natural Science Foundation of China | 81730060 | Hui Zhang |
| National Natural Science Foundation of China | 81561128007 | Hui Zhang |
| Joint-innovation Program in Healthcare for Special Scientific Research Projects of Guangzhou | 201803040002 | Hui Zhang |

The funders had no role in study design, data collection and interpretation, or the decision to submit the work for publication.

### Author contributions

Xiancai Ma, Conceptualization, Resources, Data curation, Formal analysis, Validation, Investigation, Visualization, Methodology, Writing—original draft, Writing—review and editing; Tao Yang, Data curation, Validation, Investigation, Methodology; Yuewen Luo, Liyang Wu, Yawen Jiang, Zheng Song, Ting Pan, Bingfeng Liu, Wanying Zhang, Liting Liang, Yuanjun Guan, Data curation, Methodology; Guangyan Liu, Fei Yu, Zhangping He, Jinyu Yang, Linghua Li, Weiping Cai, Xiaoping Tang, Resources, Methodology; Jun Liu, Software, Methodology; Xu Zhang, Methodology, Project administration; Song Gao, Resources, Software, Methodology; Kai Deng, Conceptualization, Resources, Methodology; Hui Zhang, Conceptualization, Supervision, Funding acquisition, Methodology, Writing—original draft, Project administration, Writing—review and editing

**Author ORCIDs**

Xiancai Ma ⓘ http://orcid.org/0000-0002-4934-4221

Song Gao ⓘ http://orcid.org/0000-0001-7427-6681

Hui Zhang ⓘ http://orcid.org/0000-0003-3620-610X

## Ethics

Human subjects: Chronically HIV-1-infected participants sampled by this study were recruited from Department of Infectious Diseases in Guangzhou 8th People's Hospital, Guangzhou. The Ethics Review Board of Sun Yat-Sen University and the Ethics Review Board of Guangzhou 8th People's Hospital approved this study. All the participants were given written informed consent with approval of the Ethics Committees. The enrollment of HIV-1-infected individuals was based on the criteria of prolonged suppression of plasma HIV-1 viremia on cART, which is undetectable plasma HIV-1 RNA levels (less than 50 copies/ml) for a minimum of six months, and having high CD4+ T cell count (at least 350 cells/mm3). Blood samples from healthy individuals were obtained from Guangzhou Blood Center. We did not have any interaction with the healthy individuals or protected information, and therefore no informed consent was required. The statement was also included in the Materials and Methods section.

## Decision letter and Author response

Decision letter https://doi.org/10.7554/eLife.42426.043

Author response https://doi.org/10.7554/eLife.42426.044

# Additional files

## Supplementary files

• Supplementary file 1. SiRNA library used to screen HIV-1 suppression and latency contributors. SiRNA library, which targeted several cellular pathways within the nucleus including chromatin binding, epigenetic modification, chromatin remodeling, ubiquitination, SUMOylation, and chromosome organization, was transfected into TZM-bl cells respectively. Both library and negative control siRNA were synthesized from RiboBio (Guangzhou, China).

DOI: https://doi.org/10.7554/eLife.42426.034

• Supplementary file 2. ChIP primers used to explore the enrichment of target proteins on HIV-1. Eight ChIP-qPCR primers targeting integrated HIV-1 reporter provirus in TZM-bl cell line were designed. G5: Cellular DNA and viral 5'LTR junction; A: Nucleosome 0 assembly site; B: Nucleosome free region; C: Nucleosome one assembly site; V5: Viral 5'LTR and gag leader sequence junction; L: Luciferase region; V3: Viral poly purine tract and 3'LTR junction; G3: Viral 3'LTR and cellular DNA junction. For ChIP-qPCR conducted in J-Lat 10.6, G5' represented cellular DNA and viral 5'LTR junction; E represented envelop; G3' represented viral 3'LTR and cellular DNA junction; A, B, C, V5 and V3 represented as in TZM-bl cell lines.

DOI: https://doi.org/10.7554/eLife.42426.035

• Supplementary file 3. SUMO mutants used in SUMO-MS and CDK9 mutants used to identify SUMOylation sites. The sequences of SUMO1-Q92R, SUMO2-Q88R and SUMO4-Q88R mutants, which mimicked yeast SUMO Smt3 to enable efficient identification of SUMO-acceptor lysines by MS, were represented below. Table also listed the major CDK9 mutants used in reversing mutation assay to identify SUMOylation sites on CDK9. All the sequences were verified by Sanger Sequencing to insure the accuracy.

DOI: https://doi.org/10.7554/eLife.42426.036

• Supplementary file 4. SUMOylated proteins at significance threshold below $10^{-7}$. Table showed 1,329 SUMOylated proteins identified in global site-specific SUMO-MS at significance threshold below $10^{-7}$.

DOI: https://doi.org/10.7554/eLife.42426.037

• Supplementary file 5. Subclusters clustered by MCODE analysis. Twelve highly interconnected functional subclusters were extracted from STRING network by MCODE analysis. Interconnectivity scores ranged from 14 to 96. Genes from each cluster were listed.

DOI: https://doi.org/10.7554/eLife.42426.038

• Supplementary file 6. Go analysis of SUMOylated proteins. Biological process analysis, molecular function analysis, cellular component analysis and protein class analysis were conducted for the identified SUMOylated proteins. Table showed gene numbers and percentages of each group.
DOI: https://doi.org/10.7554/eLife.42426.039

• Supplementary file 7. SUMOylated proteins at significance threshold below $10^{-8}$. Table showed 715 SUMOylated proteins identified in global site-specific SUMO-MS at significance threshold below $10^{-8}$.
DOI: https://doi.org/10.7554/eLife.42426.040

• Transparent reporting form
DOI: https://doi.org/10.7554/eLife.42426.041

## Data availability

All data generated or analysed during this study are included in the manuscript and supporting files.

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
