## [Decision Letter]

Thank you for submitting your article "TRIM28 Inhibits the Function of P-TEFb Complex by SUMOylating CDK9 and Leads to HIV-1 Latency" for consideration by *eLife*. Your article has been reviewed by three peer reviewers, and the evaluation has been overseen by a Reviewing Editor and Wenhui Li as the Senior Editor. The following individual involved in review of your submission has agreed to reveal his identity: Guangxia Gao (Reviewer #1).

Summary:

Your manuscript was evaluated by a dedicated group of reviewers, who have discussed the reviews with one another. The Reviewing Editor has drafted this decision to help you prepare a revised submission. We all agree that you have described an important body of work and wish to see your manuscript go forward.

Essential revisions:

There are two points that were highlighted in the comments of at least two reviewers:

1) Please provide evidence for the importance of SUMO4. Does SUMO4 disruption alter the chromatin status of the HIV DNA? Please provide evidence that SUMO4 is important in the primary cells of interest; minimally you should demonstrate that SUMO4 is expressed in the cell type of interest. Without this additional data we feel that you need to remove claims about the importance of SUMO4.

2) Please clarify whether or not the ATAC-Seq effect that you report is specific for the HIV LTR. Does this change in chromatin state occur with promoters more generally?

*Reviewer #1:*

It is well documented that TRIM28 (also known as KAP1) mediates transcriptional silencing of MLV in stem cells and endogenous retroviruses. In this manuscript, Ma et al. reported that the SUMO E3 activity of TRIM28 is important for the silencing of HIV-1 DNA in reporter cell lines (TZM-bl, J-Lat) and primary resting T cells from patients. The major findings are interesting and important and the biochemistry results of TRIM28-mediated Sumoylation of CDK9 are pretty clear and convincing. Some clarifications are required to further support the conclusions.

Major concerns:

1) As shown in Figure 1, TRIM28 is responsible for the repressive epigenetic marks deposited on HIV-1 DNA. Is the E3 activity of important for these repressive epigenetic modifications? The authors can knockdown SUMO4 (the major SUMO used by TRIM28), maybe also CDK9, and examine the repressive epigenetic marks on HIV-1 DNA (with ChIP antibodies H3K9me3, H3K27me3, H3ac, SETDB1, HP1, TRIM28). Given that the authors have all the antibodies and primers, this should be an easy experiment.

2) Discussion: paragraph two, the authors claimed that "TRIM28 was still able to enrich CDK9 in the presence of RNase (data not shown)". This is an important piece of data; it should be shown, at least in the supplementary information.

*Reviewer #2:*

Ma et al. reports a mechanism by which TRIM28 inhibits the reactivation of HIV-1. Their primary finding is the ability of TRIM28 to SUMOylate CDK9, reducing the activity of CDK9 required for a functional pTEF-b complex. The authors primarily demonstrate this interaction utilizing high-resolution microscopy, proteomics, and mutagenesis in cell line models. Authors also demonstrate this functionality is relevant in vivo by knocking down TRIM28 and showing an increase in HIV-1 reactivation in patient cells. Overall this is a very interesting study using multiple approaches to test the function of TRIM28 in HIV-1 latency, which is novel and highly significant in the field. This manuscript could be strengthened with the following revisions.

Major suggestions

1) Western blot: In Figures 2F, 3A, 3D, 3E, 4C, 4D, Figure 4—figure supplement 1I, J, Figure 7—figure supplement 1B, and Figure 9—figure supplement 1, the authors performed immunoblot for multiple proteins on a single Western blot. The authors should demonstrate anti-HA, anti-Flag and anti-GAPDH antibody blots separately. For example, when GAPDH (detected by a rabbit polyclonal antibody) is shown in the same blot, along with anti-HA and anti-Flag (tagging SUMO4, UBC9 and TRIM28), it is hard to tease out specific detection of each target.

2) ATACseq: In Figure 5, the authored showed ATACseq density near the HIV-1 integration site. The curve does not look continuous on the left panel. Authors should note whether the HIV-1 genome itself was analyzed for accessibility, and if so how they differentiated between their viral vector and the cell line genomes in regions of homology (LTRs, PSI, RRE, etc.). Authors should also include a supplemental figure indicating similar accessibility at ectopic loci (such as the promoter of the gene in which HIV-1 is integrated, and a housekeeping gene) between the two samples (sgNT vs sgTRIM28) to demonstrate comparable transposition between samples.

3) HIV-1 genetic diversification and viral outgrowth: In Figure 6, the authors analyzed the genetic diversity of HIV-1 reactivated upon different shRNA knockdown and stimulation. The phylogenetic analysis showed a diversity (>100 clones) of cell-associated HIV-1 RNA. Since the authors used TOPO cloning of PCR products instead of direct sequencing of the bulk PCR product, the so called "diversity" of HIV-1 RNA is reflecting PCR errors identified in TOPO cloning, not real HIV diversity. The authors should only focus on HIV-1 RNA levels (Figure 7A) and remove Figure 7B.

4) Viral outgrowth: In Figure 7C, all viral outgrowth culture p24 readouts have to be plotted in log scale, not linear scale. The "viral outgrowth assay" shown is mainly a yes-no viral outgrowth instead of a "quantitative" viral outgrowth measurement, as all outgrowths are positive. This does not test the hypothesis whether TRIM28 affects latency reversal (unless it's quantitative with limiting dilution). The authors should remove Figure 7C.

Reviewer #3:

In this manuscript, Ma et al. identified TRIM28 as a negative regulator of HIV transcription. Mechanistically, the authors suggest that TRIM28 post-translationally modify CDK9 by SUMO4 through its SUMO E3 ligase activity. This modification reduces both kinase activity and binding of CDK9 to cyclin T1, an essential partner of the elongation factor pTEF-b.

In spite of the elegant biochemical analysis, there are some concerns regarding whether the results completely demonstrate their conclusions regarding the role of SUMO4 in regulating pTEFb.

– All the experimental observations that SUMO4 can modify CDK9 are only evaluated in the context of over-expression systems. It will be important to address whether this modification happens in primary CD4 T cells, the main latent reservoir, under endogenous expression of TRIM28, CDK9 and SUMO4. It will be also important to address whether SUMO4 is expressed in CD4 T cells in their RNASeq data.

– Figure 3 is misleading. SUMO has a Flag epitope but not WB against FLAG is done in any of the IP membranes to ensure that the bands marked as SUMO-CDK9 are actually SUMOylated CDK9.

–The activity of CDK9 is also controlled by phosphorylation. Does SUMOylation affect CDK9 phosphorylation?

– Based on Figure 4, there is not a strong co-localization between endogenous CDK9 and TRIM28 in 293T cells, suggesting that the interaction proposed may be an artifact of the over-expression system.

– co-IP experiments shown in Figure 5E do not demonstrate that SUMOylation of CDK9 reduces binding to Cyclin T1. The figure seems mislabeled in the IP section and no reduction on binding is observed when UBC9/TRIM28/SUMO4 are co-transfected, invalidating their proposed working model in Figure 7D.

– SUMO4 can also strongly modify TRIM28 independent of CDK9. Does SUMO4 modification of TRIM28 modify its activity?

– ATAC-seq reveals a more accessible chromatin around the HIV LTR. Is this particular of the LTR or is it a global alteration of other promoters? This will be important when addressing targeting TRIM28 as potential LRA as its targeting may have multiple pleiotropic effects.

– It is important to note that reduction of TRIM28 levels both in transformed cell model of latency as well as cells isolated from aviremic participants does seem to reactivate latent HIV, however whether this is through SUMO4-mediated modification of CDK9 by TRIM28 is not fully supported by the experimental data. Furthermore, it will be important to address what it is the toxicity associated with targeting TRIM28 as well as specificity to the HIV promoter.

---

## [Author Response]

Essential revisions:There are two points that were highlighted in the comments of at least two reviewers:1) Please provide evidence for the importance of SUMO4. Does SUMO4 disruption alter the chromatin status of the HIV DNA? Please provide evidence that SUMO4 is important in the primary cells of interest; minimally you should demonstrate that SUMO4 is expressed in the cell type of interest. Without this additional data we feel that you need to remove claims about the importance of SUMO4.

We thank the editors and reviewers for bringing up the importance of SUMO4 and do apologize for missing this gap in our results, which we have now corrected. We would like to emphasize our revisions by the following points:

1) Within the section of “CDK9 Is SUMOylated by TRIM28”, we present a number of new chromatin immunoprecipitation data to indicate SUMO4 is the major HIV-1 latency contributor compared with the other SUMO paralogs. As we have shown in the newly-added fifteen figures in Figure 4D-M and Figure 4—figure supplement 1D-H, SUMO4 was enriched on HIV-1 LTR and mediated more than half of the enrichment of TRIM28 on HIV-1 LTR. This indicated that the enrichment of TRIM28 on HIV-1 LTR may be partly SUMOylation-dependent in addition to the Krüppel-associated box domain zinc fingers (KRAB-ZNFs)–dependent protein binding. Importantly, the absence of SUMO4 significantly reduced the enrichment of SETDB1, HP1α and HDAC1 on HIV-1 LTR, as well as the suppressive marks on HIV-1 LTR, such as H3K9me, H3K9me2 and H3K9me3. These data are consistent with previous findings that TRIM28 recruited SETDB1, HP1α and NuRD complex (the major component of which is HDAC1) in a SUMOylation-dependent manner. Correspondingly, the activation marks including H3K4me3 and H3K9acetyl were also increased upon SUMO4 knockdown. Intriguingly, we also noticed the decrease of H3K27me3 on HIV-1 LTR upon SUMO4 knockdown, which was compatible with our finding that the depletion of TRIM28 induced the decrease of H3K27me3 on HIV-1 promoter. Whether TRIM28 utilizes SUMO4 to influence the function of H3K27me3 mediators such as polycomb repressive complex 2 (PRC2) components EZH2 and SUZ12 is an interesting point, which deserves further investigation. However, the epigenetic roles of TRIM28 and SUMO4 are not the major points of our manuscript, which focuses on transcriptional regulation. Therefore, we have not profoundly extended on this area.

2) Within the section of “TRIM28 Suppresses HIV-1 Expression and Contributes to HIV-1 Latency” and “CDK9 Is SUMOylated by TRIM28”, we utilized four distinct siRNAs targeting the coding sequence and 3’UTR of TRIM28 or SUMO4 mRNAs to downregulate TRIM28 or SUMO4, which significantly upregulated HIV-1 promoter activity respectively, especially in combination with HIV-1 transactivator Tat. These results have been shown in four newly-added Figure 1—figure supplement 1A, Figure 1—figure supplement 1D and Figure 4A-B. To further demonstrate the importance of SUMO4 in primary CD4^+^ T cells, we firstly compared the expression of SUMO4 in different cells. We found that SUMO4 was ubiquitously expressed in several cell lines and primary CD4^+^ cells (newly-added Figure 4—figure supplement 1B). Besides, we also indicated the expression of SUMO4 in the volcanoplot of RNA-Seq data in CD4^+^ T cells which was shown in newly-added Figure 1—figure supplement 1G. The expression of SUMO4 mRNA was quantitated within unstimulated, PHA-stimulated and resting CD4^+^ T cells as we have done for the expression of TRIM28 mRNA (newly-added Figure 4—figure supplement 1C). Finally, we tested whether the depletion of SUMO4 could reactivate latent HIV-1 in resting CD4^+^ T cells isolated from HIV-1-infected individuals. The newly-added Figure 10—figure supplement 4 indicated that the depletion of SUMO4 reactivated substantial productions of HIV-1 RNAs which were even slightly higher than those activated by SAHA. The combination use of SUMO4 knockdown and SAHA addition could reactivate more HIV-1 RNAs than those reactivated by them separately. The result was consistent with that caused by TRIM28 depletion. Moreover, through immunoblotting the endogenous SUMO4, we confirmed that SUMOylation of cellular targets with SUMO4 are ubiquitous in primary CD4^+^ T cells, the result of which has been shown in newly-added Figure 5—figure supplement 1C. We have been trying very hard to monitor the endogenous SUMOylation of CDK9 in primary CD4^+^ T cells, Jurkat cell line, HeLa cell line and HEK293T cell line. However, we were unable to identify significant bands of SUMOylated CDK9. Instead, we only able to immunoblot a small portion of SUMOylated CDK9, which is in consistence with the other, previously reported SUMOylated substrates. Nevertheless, we conducted semi-endogenous SUMOylation assay. We overexpressed TRIM28, UBC9 and SUMO4 in primary CD4^+^ T cells, and immunoblotted the endogenous CDK9. The result showed that the endogenous CDK9 was also SUMOylated in the presence of exogenously expressed SUMOylation system components (newly-added Figure 5—figure supplement 1D-E).

2) Please clarify whether or not the ATAC-Seq effect that you report is specific for the HIV LTR. Does this change in chromatin state occur with promoters more generally?

TRIM28-mediated the increase of ATAC-Seq tag density is not specific for the HIV-1 LTR. The chromatin accessibilities of many TRIM28-regulated genes were also increased (newly-added Figure 8—figure supplement 1-2). This result is also consistent with the data shown in public database, which indicate that TRIM28 regulates lots of genes involved in cellular differentiation, DNA damage repairing, as well as the suppression of human cytomegalovirus (HCMV) and other human endogenous retroviruses (HERVs) in stem cells. The phenomenon that TRIM28 bound to and regulated HERVs was also found in human CD4^+^ T cells (Turelli et al., 2014, Genome Research, PMID: 24879559). Interestingly, we found that the tag densities of many corepressors of TRIM28, especially zinc finger proteins (ZNFs), were also significantly changed. This result is also consistent with a recently published paper showing that the knockout of TRIM28 induced the overexpression of several ZNFs (Tie et al., 2018, EMBO Reports, PMID: 30061100). Besides, we conducted several functional analysis and found that most genes which had upregulated ATAC-Seq density upon TRIM28 depletion were functional proteins with binding activity, catalytic activity, nucleic acid binding transcription factor activity and protein binding transcription factor activity. Few genes belonged to structural genes or housekeeping genes. More than forty percent of ATAC-Seq peaks lied in gene promoters. Forty-nine percent of ATAC-Seq peaks lied in distal intergenic regions that were enriched with HERVs and distal regulation elements. The increased accessibility enhanced the corresponding promoter activity, such as those transcription factor promoters.

It is notable that Figure 8A-B indicates that the HIV-1 promoter activity was inhibited by TRIM28 by SUMOylating CDK9. We are fully aware of the risk of pleiotropic effects caused by TRIM28 depletion. TRIM28 has long been identified as a multifunctional protein involving in transcriptional regulation, cellular differentiation and proliferation, DNA damage repair, viral suppression, and apoptosis. Also as we showed here, the depletion of TRIM28 could increase the chromatin accessibility of many functional genes. Some genes could even enhance the anti-HIV-1 activity. However, as far as we known, all of LRAs tested do not specifically target HIV-1. SAHA, the widely tested in pilot clinical trials to date, targets histone deacetylase (HDAC). JQ-1 targets the Bromodomain and Extra-Terminal (BET) family of bromodomain proteins. Disulfiram depletes the intracellular protein PTEN and consequently activates the Akt signaling pathway. Bryostatin-1, the PKC agonist, directly induces T cell activation. More LRAs and corresponding side effects have been well elucidated in a review paper published on the Annual Review of Medicine by Spivak and Planelles (PMID: 29099677). To test the toxicities associated with depleting TRIM28, we have conducted several toxicity experiments which included cytotoxicity assay, cell viability assay, cell number counting and cell proliferation assay in Jurkat cells, HeLa cells as well as resting CD4^+^ T cells isolated from aviremic participants. The newly-added Figure 10—figure supplement 1-2 indicates that the cytotoxicity, viability, cell number and cell proliferation abilities were not affected upon TRIM28 knockdown or knockout. Therefore, we propose that TRIM28 is a safe target for developing new LRAs.

Reviewer #1:[…] Major concerns:1) As shown in Figure 1, TRIM28 is responsible for the repressive epigenetic marks deposited on HIV-1 DNA. Is the E3 activity of TRIM28 important for these repressive epigenetic modifications? The authors can knockdown SUMO4 (the major SUMO used by TRIM28), maybe also CDK9, and examine the repressive epigenetic marks on HIV-1 DNA (with ChIP antibodies H3K9me3, H3K27me3, H3ac, SETDB1, HP1, TRIM28). Given that the authors have all the antibodies and primers, this should be an easy experiment.

We thank the reviewer for the insightful comment and agree that we should provide several lines of evidence to show the influence of SUMOylation on repressive epigenetic modifications. As the reviewer kindly suggested, we addressed this vital point in two major aspects as shown below.

1) We provided new evidence that the E3 activity of TRIM28 was important for the repressive epigenetic modifications. For a clear logic flow, we have summarized the results within the section of “TRIM28 SUMOylates Many Transcription Factors and Transferases” as Figure 2C-E. In this part, we have proved that both the intermolecular SUMOylation domain RING and intramolecular SUMOylation domain PHD of TRIM28 meditated the suppression of HIV-1 LTR. To test whether these two domains influenced the epigenetic status of HIV-1 LTR, firstly we knocked down the endogenous TRIM28 with siRNA targeting the 3’UTR of TRIM28 mRNA. Then we overexpressed the wild type TRIM28 construct and TRIM28 mutants without RING or PHD domain, respectively. The mRNAs expressed by these constructs were able to resist the degradation because the siRNA we used here targeting the 3’UTR of TRIM28 mRNA and the TRIM28 constructs we transfected here only expressed the coding sequences. Further, we investigated the epigenetic status of HIV-1 LTR among these different groups. We found that the absence of endogenous TRIM28 resulted in the decrease of suppressive epigenetic marks H3K9me3 and H3K27me3, as well as the increase of active epigenetic marks H3K9acetyl. The addition of exogenous wild type TRIM28 was able to rescue the suppressive epigenetic marks. However, the mutants without RING or PHD domain were only able to rescue the partial suppressive epigenetic marks, which indicated that both SUMOylation domains participated in the epigenetic regulation of HIV-1 LTR. To fully elucidate the detailed mechanisms behind, we conducted global site-specific SUMO-MS to identify the SUMOylation substrates as we have done within the section of “TRIM28 SUMOylates Many Transcription Factors and Transferases”.

2) We provided new evidence that SUMO4 increases the repressive epigenetic marks on HIV-1 LTR. We utilized several siRNAs targeting three coding regions and one 3’UTR region of TRIM28 or SUMO4 mRNA respectively and confirmed that SUMO4 suppressed the HIV-1 LTR activity (newly-added Figure 1—figure supplement 1A and Figure 4A). Then, we examined the suppressive epigenetic marks on HIV-1 LTR upon SUMO4 knockdown. We firstly examined the enrichment of TRIM28 on HIV-1 LTR when we knocked down SUMO4. We found that more than half of TRIM28 was lost from HIV-1 LTR upon SUMO4 knockdown, which indicated that the enrichment of TRIM28 on HIV-1 LTR may be partially SUMOylation-dependent apart from the Krüppel-associated box domain zinc fingers (KRAB-ZNFs)–dependent binding. H3K9me, H3K9me2 and H3K9me3 were significantly decreased on HIV-1 LTR in the absence of SUMO4, as well as the H3K9 methylation “writer” SETDB1 and “reader” HP1α. Moreover, we observed significant upregulation of H3K9acetyl and H3K4me3 and downregulation of HDAC1, which was consistent with previous reports that TRIM28 recruited SETDB1, HP1α and HDAC1 in a SUMOylation-dependent manner (Iyengar and Farnham, 2011). Besides, the H3K27me3 was also decreased on HIV-1 LTR upon SUMO4 knockdown. We suspected that some polycomb repressive complex 2 (PRC2) components such as EZH2 and SUZ12, the major “writers” of H3K27me3, may be SUMOylated by SUMO4, resulting in the enhancement of modifier function. However, we feel that the interactions between TRIM28 and SUMO4 with other epigenetic modifiers are beyond our scope. Therefore, we did not conduct further research on this area. The above results have been shown in several newly-added Figure 4D-M and Figure 4—figure supplement 1D-H. The reviewer also suggested that we might knock down CDK9 to examine the repressive epigenetic marks on HIV-1 DNA. However, we found that the knockdown of CDK9 was very toxic to the cells (data not shown). Besides, we feel that the SUMO4-mediated epigenetic marks change belongs to the epigenetic regulation of HIV-1 latency, while the SUMO4-mediated CDK9 SUMOylation belongs to the transcriptional regulation of HIV-1 latency. Similarly, the possible CDK9-meditated epigenetic modifications are also beyond the scope of our study.

2) Discussion: paragraph two, the authors claimed that "TRIM28 was still able to enrich CDK9 in the presence of RNase (data not shown)". This is an important piece of data; it should be shown, at least in the supplementary information.

We apologize for not showing it in the original submission. We have now shown it in newly-added Figure 7—figure supplement 1A, which indicates that TRIM28 was able to enrich CDK9 even in the presence of RNase.

Reviewer #2:[…] Major suggestions1) Western blot: In Figures 2F, 3A, 3D, 3E, 4C, 4D, Figure 4—figure supplement 1I, J, Figure 7—figure supplement 1B, and Figure 9—figure supplement 1, the authors performed immunoblot for multiple proteins on a single Western blot. The authors should demonstrate anti-HA, anti-Flag and anti-GAPDH antibody blots separately. For example, when GAPDH (detected by a rabbit polyclonal antibody) is shown in the same blot, along with anti-HA and anti-Flag (tagging SUMO4, UBC9 and TRIM28), it is hard to tease out specific detection of each target.

Our apologies for the confusion. We have separated each band of the target proteins in our revised manuscript.

2) ATACseq: In Figure 5, the authored showed ATACseq density near the HIV-1 integration site. The curve does not look continuous on the left panel. Authors should note whether the HIV-1 genome itself was analyzed for accessibility, and if so how they differentiated between their viral vector and the cell line genomes in regions of homology (LTRs, PSI, RRE,etc.). Authors should also include a supplemental figure indicating similar accessibility at ectopic loci (such as the promoter of the gene in which HIV-1 is integrated, and a housekeeping gene) between the two samples (sgNT vs sgTRIM28) to demonstrate comparable transposition between samples.

We apologize for the incomplete explanation of the analysis strategy for ATAC-Seq data, which we have carefully modified in the Materials and methods section of “RNA-Seq and ATAC-Seq”.

1) Yes, the HIV-1 genome itself was analyzed for accessibility. The reads were aligned to HIV-1 reference genome K03455, M38432 (Version K03455.1) by Bowtie2, followed by rearranging with Samtools. The human genome was aligned to human reference genome GRCh38. The integration sites of HIV-1 pseudotyped viruses in J-Lat 10.6 and TZM-bl have been identified by genome walking strategy. Thus, the read density centered HIV-1 5’LTR could be calculated and combined together. The density curves were indeed continuous. However, because we only showed the normalized tag densities of two kilobases range centered HIV-1 LTR, which was also one of the peak center of HIV-1, the other tag densities outside two kilobases have been truncated. LTR, PSI, and RRE sequences of integrated pseudotyped HIV-1 proviruses are different from those of human endogenous retrovirus (HERV) although they share very high homology. We still could distinguish them when we aligned the reads to HIV-1 reference genome or human reference genome.

2) The reviewer also kindly suggested us to include a supplemental figure indicating similar accessibility at ectopic loci. We now have made newly-added Figure 8—figure supplement 2C-F to show the accessibilities at the promoters of the genes which the pseudotyped HIV-1 proviruses are integrated as well as at the promoters of housekeeping gene GAPDH. The results showed that the chromatin accessibilities of these genes were similar between wild type cells or TRIM28 depletion cells.

3) HIV-1 genetic diversification and viral outgrowth: In Figure 6, the authors analyzed the genetic diversity of HIV-1 reactivated upon different shRNA knockdown and stimulation. The phylogenetic analysis showed a diversity (>100 clones) of cell-associated HIV-1 RNA. Since the authors used TOPO cloning of PCR products instead of direct sequencing of the bulk PCR product, the so called "diversity" of HIV-1 RNA is reflecting PCR errors identified in TOPO cloning, not real HIV diversity. The authors should only focus on HIV-1 RNA levels (Figure 7A) and remove Figure 7B.

We thank the reviewer for the suggestion. However, we still think that the HIV-1 genetic diversity assay is vital for supporting our hypothesis that TRIM28 could be the target for developing dual functional LRAs releasing both epigenetic and transcriptional restrictions. We hope to keep Figure 7B (Figure 10B in revised manuscript).

The reviewer concerns that the PCR errors which were introduced by TOPO cloning might overestimate the HIV-1 diversity. However, the PCR/cloning method which we used was not TOPO cloning. We apologize for having omitted some essential experimental description, which we have added in the revised manuscript. The PCR/cloning method, which we used to ligate the PCR products, was a little different from the standard TOPO cloning which used *Taq* DNA polymerase to amplify DNA fragments. For each PCR reaction, we firstly used Phanta Max Super-Fidelity DNA Polymerase (Vazyme) to amplify the V1-V3 region of HIV-1 *envelope* in order to ensure the fidelity. The amplification error rate of Phanta Max is 53-fold lower than that of *Taq* and 6-fold lower than that of *Pfu* according to the manufacturer’s instruction. After two rounds of nested PCR utilizing Phanta Max, the PCR products were proceeded to deoxyadenosine (A)-tailing at the 3'-end of the PCR products utilizing *Ex Taq* DNA polymerase (Takara) without thermal cycling as follows: 95℃, 5 min; 72℃, 30 min; 4℃ hold. The A-tailed PCR products were TA-ligated into pMD-18T vector. To minimize the sampling bias, we also obtained 30 independent PCR products for each sample. We also picked at least 60 single clones for each PCR products to detect low frequency mutations. Although some groups used single genome sequencing (SGS) method to analyze the HIV-1 genetic diversity, we found that we could hardly get enough PCR amplicons when we serially diluted the cDNA due to the low viral cDNA. The HIV-1 genetic diversities in infected individuals might be underestimated if we used SGS. We also carefully referred to papers comparing these two methods. Based on the paper of Michael R. Jordan et al., 2010, Journal of Virological Methods (PMID: 20451557), both PCR/cloning and SGS measures intra-patient HIV-1 genetic diversity similarly. The intrapopulation average pairwise distance (APD), the sampling bias and the systematic position specific bias, which are measured by each method, show no differences.

The reason we suggested to keep Figure 7B is that the HIV-1 genetic diversity assay is a method firstly described by our group to assess the quality of LRAs and corresponding cellular targets (Geng et al., 2016). We propose that the viral production only is unable to reflect the true size of the HIV-1 latent reservoir. Although some LRAs such as PMA/ionomycin could activate substantial HIV-1 RNA, the genetic diversities of these activated viral RNAs were quite low. On contrary, an attenuated Tat Protein, Tat-R5M4 which was screened out by our lab, could reactivate more genetically-diversified HIV-1 than PMA/ionomycin, although the amount of reactivated viral RNA was lower than that reactivated by PMA/ionomycin. Thus, we have proposed that different LRAs reactivate different amounts of genetically-diversified HIV-1 (Author response image 1).

**Author response image 1. respfig1:** Attenuated Tat Protein Tat-R5M4 can reactivate more genetically-diversified HIV-1. (Geng et al., 2016, *Molecular Therapy*, PMID: 27434587).

The underlying rationale is that there are viral quasispecies within an HIV-1-infected individual. After HIV-1 replicates for months or years, the genetically-diversified viruses could persistently convert into integrated proviral DNA in the CD4^+^ T cells and some of them become latent viral reservoir. For such a genetically-diversified viral reservoir, some LRAs such as PMA/ionomycin could reactivate more HIV-1 RNA production but with lower genetic diversity, indicating that only a few integrated proviruses are activated and the viral RNA are generated from these small amount of integrated viral DNA merely with a high transcriptional efficiency. Some LRAs such as Tat-R5M4 or SAHA can reactivate more genetically-diversified HIV-1, even though with a lower amount. Given that the major purpose of LRA is to expose more latently-infected CD4 cells to immunosurveillance, it is quite important that proviruses at more integration sites in more latently-infected cells are activated. Therefore, we believe that genetic diversity of activated viral RNA would be an important biomarker to access the quality of LRAs (Author response image 2). Downregulation of TRIM28 result in the activation of more genetically-diversified RNA, suggesting that it can activate more integrated proviruses and merits being an ideal target for LRA development.

**Author response image 2. respfig2:** Different LRAs reactivate genetically-diversified HIV-1 at different integration sites.

4) Viral outgrowth: In Figure 7C, all viral outgrowth culture p24 readouts have to be plotted in log scale, not linear scale. The "viral outgrowth assay" shown is mainly a yes-no viral outgrowth instead of a "quantitative" viral outgrowth measurement, as all outgrowths are positive. This does not test the hypothesis whether TRIM28 affects latency reversal (unless it's quantitative with limiting dilution). The authors should remove Figure 7C.

We thank the reviewer for the suggestion. We have modified Figure 7C (Figure 10C in revised manuscript) by showing the viral outgrowth culture p24 readouts in log scale instead of linear scale. The reviewer also suggested to remove Figure 7C. However, we hope to keep this figure, which is also very important to support our hypothesis. Our purpose to conduct this experiment is not to measure the size of viral latent reservoir but determine the viruses we reactivated are the replication-competent rather than the dead viruses. The accumulating production of p24 indicated the ongoing HIV-1 replication. The time course study would measure the kinetics of viral replication and therefore reflect the viral infectivity, which is much better than the yes-no viral outgrowth. Our methodology to determine replication-competent HIV-1 between different experimental conditions has been well-established and repeated in several of our published papers (Huang et al., 2007; Li et al., 2016; Geng et al., 2016; Liu et al., 2016) and many works from other HIV-1 latency labs as well. Besides, although standard QVOA could be better to test our hypothesis, we are greatly limited by the shortage of large samples (more than 180 mL of blood for a single experimental group) from the study participants. Moreover, viral reservoir measured with VOA merely stands for a quite small amount of latent viral reservoir (Ho et al., 2013).

Reviewer #3:In this manuscript, Ma et al. identified TRIM28 as a negative regulator of HIV transcription. […] In spite of the elegant biochemical analysis, there are some concerns regarding whether the results completely demonstrate their conclusions regarding the role of SUMO4 in regulating pTEFb.

We thank the reviewer for supporting this study. We have addressed each comment according to the reviewer’s suggestion. Especially, we further elucidated the role of SUMO4 on HIV-1 latency.

– All the experimental observations that SUMO4 can modify CDK9 are only evaluated in the context of over-expression systems. It will be important to address whether this modification happens in primary CD4 T cells, the main latent reservoir, under endogenous expression of TRIM28, CDK9 and SUMO4. It will be also important to address whether SUMO4 is expressed in CD4 T cells in their RNASeq data.

We thank the reviewer for pointing out these absences. As we have explained in the Discussion, the percentage of SUMOylated CDK9 is only a small proportion, less than 5%. The phenomenon is also observed for most of the previously identified SUMOylation targets. Through immunoblotting the endogenous SUMO4, we confirmed that SUMOylation of cellular targets with SUMO4 are ubiquitous in primary CD4^+^ T cells, the result of which has been shown in newly-added Figure 5—figure supplement 1C. We have been trying very hard to monitor the endogenous SUMOylation of CDK9 in primary CD4^+^ T cells, Jurkat cell line, HeLa cell line and HEK293T cell line. However, we were unable to identify significant bands of SUMOylated CDK9. Instead, we only able to immunoblot a small portion of SUMOylated CDK9, which is in consistence with the other, previously reported SUMOylated substrates (newly-added Figure 5—figure supplement 1D). Nevertheless, we conducted semi-endogenous SUMOylation assay. We overexpressed TRIM28, UBC9 and SUMO4 in primary CD4^+^ T cells, and immunoblotted the endogenous CDK9. The result showed that the endogenous CDK9 was also SUMOylated in the presence of exogenously expressed SUMOylation system components, the result of which has been shown in newly-added Figure 5—figure supplement 1E.

To confirm that SUMO4 is expressed in the cells we used, we compared the expression of SUMO4 in different cells to show that SUMO4 is ubiquitously expressed in several cell lines and primary cells (newly-added Figure 4—figure supplement 1B). We also indicated the expression of SUMO4 in the volcanoplot of RNA-Seq data which we showed in Figure 1—figure supplement 1G. Besides, we quantitated the expression of SUMO4 mRNA within unstimulated, PHA-stimulated and memory CD4^+^ T cells as we have conducted for the expression of TRIM28 mRNA, the result of which has been shown in newly-added Figure 4—figure supplement 1C.

– Figure 3 is misleading. SUMO has a Flag epitope but not WB against FLAG is done in any of the IP membranes to ensure that the bands marked as SUMO-CDK9 are actually SUMOylated CDK9.

Our apologies for the confusion. All the IP membranes were actually immunoblotted with both antibodies against HA and Flag. We missed the statement of “IB: Flag” on the left side of the figures. The SUMO-CDK9 represented the results of SUMOylated CDK9 with both epitopes. In order to present the results more precisely, we have separated the anti-HA and anti-Flag blots in our revised manuscript.

–The activity of CDK9 is also controlled by phosphorylation. Does SUMOylation affect CDK9 phosphorylation?

This is an interesting point raised by the reviewer. Although we did not conduct systematic experiments to elucidate the possibility of the effect of SUMOylation on CDK9 phosphorylation, we expect that SUMOylation could affect CDK9 phosphorylation. In our effort to locate all the SUMOylation sites on CDK9, we surprisingly found that several lysines on CDK9 were SUMOylated. Among them, multiple SUMOylation sites: Lys274, Lys276, Lys280, Lys325 and Lys345 were adjacent to CDK9 C-terminal autophosphorylation sites which have been reported to be required for high-affinity binding of Tat–P-TEFb to TAR RNA (Baumli et al., 2008; Garber et al., 2000). SUMOylation may decrease the binding ability by preventing the neighboring phosphorylation. However, in our further effort to identify which sites were indeed SUMOylated by TRIM28, we found that only the Lys44, Lys56 and Lys68 residues were specifically SUMOylated by TRIM28. The SUMOylation sites adjacent to CDK9 C-terminal autophosphorylation sites were not SUMOylated by TRIM28. Other CDK9 SUMOylation E3 ligases may exist to mediate their SUMOylation. Because other E3 ligases are out of our research scope, we have not conduct further experiments to study the effect of SUMOylation on CDK9 phosphorylation. The reviewer’s question is really helpful for the comprehensiveness of our work. Thus, we have added the following statements: “Among them, multiple SUMOylation sites were adjacent to CDK9 C-terminal autophosphorylation sites which have been reported to be required for high-affinity binding of Tat–P-TEFb to TAR RNA (Baumli et al., 2008; Garber et al., 2000). SUMOylation may decrease the binding ability by preventing the neighboring phosphorylation.”

– Based on Figure 4, there is not a strong co-localization between endogenous CDK9 and TRIM28 in 293T cells, suggesting that the interaction proposed may be an artifact of the over-expression system.

We apologize for not explaining the imaging figures clearly. The results we presented in Figure 4A and 4B (Figure 6A-B in revised manuscript) are the cSTORM images of endogenous TRIM28 with endogenous SUMO4 and CDK9. We did not overexpress any of them. The single molecule localization is obtained by Gaussian fitting. The co-localization of cSTORM is measured by the center of two points, which is slightly different from confocal, which measures the overlapping of two different protein clusters. From the amplified view, 3D-cSTORM and related videos, we easily found that dotted SUMO4 and CDK9 proteins were enriched by TRIM28 and shaped big spots. In Figure 4E (Figure 7C in revised manuscript) which was SIM image of overexpressed system, we used exogenously expressed GFP-tagged wild type TRIM28, GFP-tagged TRIM28 mutant and RFP-tagged CDK9 to investigate the co-localization due to that we have not constructed the endogenously expressed TRIM28 mutant so far. We still can easily notice the significant co-localization of GFP-TRIM28 and RFP-CDK9. We have conducted two statistical analyses, shown in the newly-added Figure 6E and Figure 7D in our revised manuscript. The quantitation strategies were elucidated within the method of “SIM and STORM imaging”. (Figure 4 has been made two newly-added Figure 6 and Figure 7.)

– co-IP experiments shown in Figure 5E do not demonstrate that SUMOylation of CDK9 reduces binding to Cyclin T1. The figure seems mislabeled in the IP section and no reduction on binding is observed when UBC9/TRIM28/SUMO4 are co-transfected, invalidating their proposed working model in Figure 7D.

We apologize for mislabeling and the lack of proper explanation. We have specifically labeled “SUMO-CDK9” which represented SUMOylated CDK9 in the upper part of Figure 5E (Figure 8E in revised manuscript). From the IP section of the result, we can easily notice the significant reduction of SUMOylated CDK9, which was not enriched by Cyclin T1. Instead, the wild type CDK9 was able to bind to Cyclin T1 was unchanged. Because the system we used here was the exogenous overexpression, it is very difficult to observe the reduction of wild type CDK9, although some CDK9 has been SUMOylated.

– SUMO4 can also strongly modify TRIM28 independent of CDK9. Does SUMO4 modification of TRIM28 modify its activity?

This is an interesting question. We did find that the function of TRIM28 was influenced by SUMO4 modification of TRIM28. The PHD domain of TRIM28 can mediate the intramolecular SUMOylation of TRIM28. The RING domain of TRIM28 is not only able to mediate the intermolecular SUMOylation of other proteins (Figure 3D), but also be able to mediate the intramolecular SUMOylation of TRIM28 (Figure 7—figure supplement 1B). As we have shown in Figure 7—figure supplement 1B, TRIM28 was strongly self-SUMOylated in the presence of SUMO4 and UBC9.

When we knocked down SUMO4 in the TZM-bl cell line, we found that more than half of TRIM28 was lost from HIV-1 LTR, which indicated that the enrichment of TRIM28 on HIV-1 LTR was partially SUMOylation-dependent, which is apart from the Krüppel-associated box domain zinc fingers (KRAB-ZNFs)–dependent binding. Some other HIV-1 LTR binding proteins may harbor SUMO-interacting motifs (SIM) mediate the enrichment of SUMOylated TRIM28. The result mentioned above has been shown in newly-added Figure 4D.

Furthermore, we found that the H3K9 methylation “writer” SETDB1 and “reader” HP1α were significantly decreased on HIV-1 LTR in the absence of SUMO4. Moreover, we observed a significant upregulation of H3K9acetyl and H3K4me3 and a downregulation of HDAC1. These results were in consistence with previous reports that TRIM28 recruited SETDB1, HP1α and HDAC1 in a SUMOylation-dependent manner. The result mentioned above has been shown in newly-added Figure 4E-M.

– ATAC-seq reveals a more accessible chromatin around the HIV LTR. Is this particular of the LTR or is it a global alteration of other promoters? This will be important when addressing targeting TRIM28 as potential LRA as its’ targeting may have multiple pleiotropic effects.

We thank the reviewer for the vital concern. The specificity of TRIM28 to HIV-1 LTR is not our study priority, however, the concern of which is very important when addressing targeting TRIM28 as potential LRA. The phenomenon that TRIM28 contributes to HIV-1 latency and the mechanism that TRIM28 SUMOylates CDK9 to mediate transcriptional control are two key findings of our study and we wish that TRIM28 could be a candidate target to develop LRAs. As we have carefully addressed in Essential revision 2 to editors and reviewers, we would like to comprehensively elucidate the question from three points as shown below:

1) TRIM28-mediated the increase of ATAC-Seq tag density is not specific for the HIV-1 LTR. The chromatin accessibilities of many TRIM28-regulated genes were also increased. The newly-added Figure 8—figure supplement 1-2 show the above result. This result was also consistent with the data shown in public database, which indicate that TRIM28 regulated lots of genes involved in cellular differentiation, DNA damage repairing, as well as the suppression of human cytomegalovirus (HCMV) and other human endogenous retroviruses (HERVs) in stem cells. The phenomenon that TRIM28 bound to and regulated HERVs was also found in human CD4^+^ T cells (Turelli et al., 2014, Genome Research, PMID: 24879559). Interestingly, we found that the tag densities of many corepressors of TRIM28, especially zinc finger proteins (ZNFs), were also significantly changed, the result of which was also consistent with a recently published paper showing that the knockout of TRIM28 induced the overexpression of several ZNFs (Tie et al., 2018, EMBO Reports, PMID 30061100). They proposed a model that the depletion of TRIM28 could reactivate some HERVs and ZNFs.

2) Besides, we conducted several functional analysis and found that most genes which had upregulated ATAC-Seq density upon TRIM28 depletion were functional proteins with binding activity, catalytic activity, nucleic acid binding transcription factor activity and protein binding transcription factor activity. Few genes belonged to structural genes or housekeeping genes. More than forty percent of ATAC-Seq peaks lied in gene promoters. Forty-nine percent of ATAC-Seq peaks lied in distal intergenic regions that were enriched with HERVs and distal regulation elements. The increased accessibility enhanced the corresponding promoter activity, such as those transcription factor promoters. Thus, these transcription factors promoted the HIV-1 reactivation by enhancing transcription.

3) The result we showed in Figure 5A-B (Figure 8A-B in revised manuscript) is to prove that the HIV-1 promoter activity was inhibited by TRIM28 through SUMOylating CDK9. We were fully aware of the risk of pleiotropic effects caused by TRIM28 depletion. TRIM28 has long been identified as a multifunctional protein involving in transcriptional regulation, cellular differentiation and proliferation, DNA damage repair, viral suppression, and apoptosis. Also as we showed here, the depletion of TRIM28 could increase the chromatin accessibility of many functional genes. Some genes could enhance the anti-HIV-1 activity. However, as far as we have known, all of LRAs tested so far do not specifically target HIV-1. SAHA, the widely tested in pilot clinical trials to date, targets histone deacetylase (HDAC). JQ-1 targets the Bromodomain and Extra-Terminal (BET) family of bromodomain proteins. Disulfiram depletes the intracellular protein PTEN, resulting in activating the Akt signaling pathway. Bryostatin-1, the PKC agonist, directly induces T cell activation. More LRAs and corresponding side effects have been well elucidated in a paper published on the Annual Review of Medicine by Spivak and Planelles (PMID: 29099677).

– It is important to note that reduction of TRIM28 levels both in transformed cell model of latency as well as cells isolated from aviremic participants does seem to reactivate latent HIV, however whether this is through SUMO4-mediated modification of CDK9 by TRIM28 is not fully supported by the experimental data. Furthermore, it will be important to address what it is the toxicity associated with targeting TRIM28 as well as specificity to the HIV promoter.

Thank you for pointing out these points of confusion in the original manuscript. If we understand correctly, this comment addressed three different issues. The first issue is the concern of the contribution of the SUMO4-mediated modification of CDK9 by TRIM28 on HIV-1 latency. The second issue is to address the toxicity associated with targeting TRIM28. The third issue is the specificity of targeting TRIM28 to the HIV-1 promoter.

1) We firstly would like to address that SUMO4-mediated modification of CDK9 by TRIM28 is one of the mechanisms used by TRIM28 to contribute to HIV-1 latency in cells isolated from aviremic participants. As we have carefully elucidated in Essential revision 1 to editors and reviewers, within the section of “TRIM28 Suppresses HIV-1 Expression and Contributes to HIV-1 Latency”, we utilized four distinct siRNAs targeting the coding sequence and 3’UTR of TRIM28 and SUMO4 mRNAs to downregulate TRIM28 and SUMO4, which significantly upregulated HIV-1 promoter activity, especially in combination with HIV-1 transactivator Tat. These results have been shown in four newly-added Figure 1—figure supplement 1A, Figure 1—figure supplement 1D and Figure 4A-B. To further demonstrate the importance of SUMO4 in primary CD4^+^ T cells, we firstly compared the expression of SUMO4 in different cells. We found that SUMO4 was ubiquitously overexpressed in several cell lines and primary CD4^+^ cells (newly-added Figure 4—figure supplement 1B). Besides, we also indicated the expression of SUMO4 in the volcanoplot of RNA-Seq data in CD4^+^ T cells shown in newly-added Figure 1—figure supplement 1G. The expression of SUMO4 mRNA was quantitated within unstimulated, PHA-stimulated and memory CD4^+^ T cells as we have done for the expression of TRIM28 mRNA (newly-added Figure 4—figure supplement 1C). Finally, we tested whether the depletion of SUMO4 could reactivate latent HIV-1 in resting CD4^+^ T cells isolated from HIV-1-infected individuals. The newly-added Figure 10—figure supplement 4 indicated that the depletion of SUMO4 reactivated substantial productions of HIV-1 RNAs which were even slightly higher than those activated by SAHA. The combination use of SUMO4 knockdown and SAHA addition could reactivate more HIV-1 RNAs than those reactivated by them separately. The result was consistent with that caused by TRIM28 depletion. Moreover, through immunoblotting the endogenous SUMO4, we confirmed that SUMOylation of cellular targets with SUMO4 are ubiquitous in primary CD4^+^ T cells, the result of which has been shown in the newly-added Figure 5—figure supplement 1C. We have been trying very hard to monitor the endogenous SUMOylation of CDK9 in primary CD4^+^ T cells, Jurkat cell line, HeLa cell line and HEK293T cell line. However, we were unable to identify significant bands of SUMOylated CDK9. Instead, we only able to immunoblot a small portion of SUMOylated CDK9, which is in consistence with the other, previously reported SUMOylated substrates (newly-added Figure 5—figure supplement 1D). Nevertheless, we conducted semi-endogenous SUMOylation assay. We overexpressed TRIM28, UBC9 and SUMO4 in primary CD4^+^ T cells, and immunoblotted the endogenous CDK9. The result showed that the endogenous CDK9 was also SUMOylated in the presence of exogenously expressed SUMOylation system components, the result of which has been shown in newly-added Figure 5—figure supplement 1E.

2) To address the possible toxicities associated with targeting TRIM28, we conducted several experiments including cytotoxicity assay, cell viability assay, cell number counting and cell proliferation assay. We used siRNAs to knock down TRIM28 in HeLa cells and the resting CD4^+^ T cells isolated from aviremic participants. We also used shRNA constructs to knock down TRIM28 and sgRNA constructs to knock out TRIM28 in Jurkat cells. The cytotoxicity was measured by comparing the amounts of dehydrogenases between wild type cells and TRIM28 knockdown or knockout cells. The cell viability was measured by comparing the percentages of live cells between wild type cells and TRIM28 deficiency cells. The cell numbers were measured by counting cells every days upon TRIM28 knockdown or knockout. The cell proliferation abilities were measured by CFSE staining. The results have been shown in newly-added Figure 10—figure supplement 1-2. We found that upon TRIM28 knockdown, the cytotoxicity, viability, cell number, and cell proliferation abilities were not influenced compared with wild type cells. However, previous reports found that targeting HDACs with SAHA had some toxicity to cell viability, although the toxicities might come from the side effects of LRAs. Our finding here provided another safe target to develop new LRAs, which was TRIM28.

3) The reviewer’s third concern that the specificity of targeting TRIM28 to the HIV-1 promoter has been explained in the ATAC-Seq data interpretation section. In summary, HIV-1 LTR was not the only target of TRIM28. All of the reported targets of LRAs were not specific to HIV-1 LTR only as well. We have conducted several toxicities experiments to prove that targeting TRIM28 is non-toxic. We do provide another novel target to develop LRA, although other target may also be influenced. We will carefully examine the other toxicity and carcinogenic potency associated with TRIM28 deficiency when we develop LRAs.